# Sketchy Moment Matching: Toward Fast and Provable Data Selection for Finetuning

**Yijun Dong**[*]
Courant Institute
New York University
yd1319@nyu.edu

**Hoang Phan**[*]
Center of Data Science
New York University
hvp2011@nyu.edu

**Xiang Pan**[*]
Center of Data Science
New York University
xiangpan@nyu.edu

**Qi Lei**
Center of Data Science
New York University
ql518@nyu.edu

## Abstract

We revisit data selection in a modern context of finetuning from a fundamental perspective. Extending the classical wisdom of variance minimization in low dimensions to high-dimensional finetuning, our generalization analysis unveils the importance of additionally reducing bias induced by low-rank approximation. Inspired by the variance-bias tradeoff in high dimensions from the theory, we introduce **Sk**etchy **M**oment **M**atching (SkMM), a scalable data selection scheme with two stages. (i) First, the bias is controlled using gradient sketching that explores the finetuning parameter space for an informative low-dimensional subspace $\mathcal{S}$; (ii) then the variance is reduced over $\mathcal{S}$ via moment matching between the original and selected datasets. Theoretically, we show that gradient sketching is fast and provably accurate: selecting $n$ samples by reducing variance over $\mathcal{S}$ preserves the fast-rate generalization $O(\dim(\mathcal{S})/n)$, independent of the parameter dimension. Empirically, we concretize the variance-bias balance via synthetic experiments and demonstrate the effectiveness of SkMM for finetuning in real vision tasks.

## 1 Introduction

As the data volume and training cost explode with the unprecedented model performance, the long-standing problem of data selection [1, 2] is getting increasing attention in the modern context of deep learning from various perspectives, including data pruning [3, 4], coreset selection [1, 5, 6, 7, 8, 9], and data filtering [2, 10, 11, 12, 13]. A common goal shared by these perspectives is to train a model from scratch on less data to learn high-quality representations and achieve competitive generalization. However, empirical observations also suggest the limitation of data removal during pre-training: a seemingly inevitable tradeoff between less computation and higher-quality representations [14, 15]. While existing works on data selection have a dominating focus on the training-from-scratch setting, the sensitivity of representation learning to data and the growing availability of powerful pre-trained models calls for attention to a less studied [16] but equally important problem: data selection for finetuning.

In the simplest finetuning setting—linear probing on low-dimensional representations[2], data selection falls in the classical frames of coreset selection for linear regression [17, 18, 19, 20, 21, 22, 23, 24] and optimal experimental design [25, 26, 27, 28, 29] where the generalization gap can be

---

[*]Equal contribution.

[2]Throughout this work, we refer to "low-dimension" as the setting where the number of finetuning parameters $r$ is smaller than the selected downstream sample size $n$, while "high-dimension" refers to the opposite, $r > n$.

38th Conference on Neural Information Processing Systems (NeurIPS 2024).

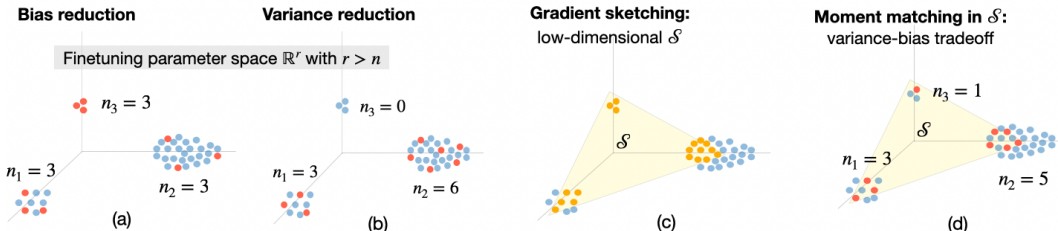

Figure 1: Controlling variance-bias tradeoff in data selection for high-dimensional finetuning via gradient sketching + moment matching (SkMM). Consider a toy dataset with $N$ samples (in blue) whose finetuning gradients lie in a high-dimensional parameter space $\mathbb{R}^r$ (visualized in 3D) with a low intrinsic dimension (*e.g.*, three clusters). The goal is to select $n = n_1 + n_2 + n_3 < r$ samples for finetuning. (a) **Bias reduction** focuses on minimizing the low-rank approximation error, resulting in uniform selection across clusters regardless of their variance. (b) **Variance reduction**[3] places more emphasis on high-variance clusters and could lead to large bias by missing low-variance ones. (c) **Gradient sketching** efficiently finds a low-dimensional subspace $\mathcal{S}$ (where $\dim(\mathcal{S}) < n$) with small bias. (d) **Moment matching** in $\mathcal{S}$ controls the variance within the low-bias subspace, leading to a variance-bias balance with fast-rate generalization $O(\dim(\mathcal{S})/n)$.

reduced by selecting data that minimize the associated variance. However, for high-dimensional finetuning, variance minimization alone is insufficient to characterize the generalization due to the overparametrized nature of modern architectures. Even for linear probing, when the parameter dimension $r$ is higher than the sample size $n$, the selected data necessarily fails to capture a subspace of the parameter space with dimension at least $r - n$, leading to errors in addition to variance. Nevertheless, the prevailing empirical and theoretical evidence [14, 30] on the ubiquitous intrinsic low-dimensional structures of high-dimensional data/model motivates a natural question:

*Can the low intrinsic dimension be leveraged in data selection for high-dimensional finetuning?*

**Low intrinsic dimension leads to variance-bias tradeoff in data selection.** We provide a positive answer to this question through a variance-bias tradeoff perspective. Intuitively, we consider a low-dimensional subspace $\mathcal{S}$ in the finetuning parameter space where the model learns the necessary knowledge for the downstream task. The generalization gap can be controlled by simultaneously reducing *the bias (redundant information) by "exploring" the finetuning parameter space to find a suitable $\mathcal{S}$* and *the variance by "exploiting" the useful knowledge in $\mathcal{S}$*.

Given the high-dimensional nature of the finetuning parameter space, direct search for such suitable subspace $\mathcal{S}$ is computationally infeasible in general. This leads to a follow-up question:

*How to explore the intrinsic low-dimensional structure efficiently for data selection?*

We propose a two-stage solution—*Sketchy Moment Matching (SkMM)*: (i) dimensionality reduction via gradient sketching to efficiently explore the finetuning parameter space, and (ii) variance control via moment matching to exploit useful knowledge in the low-dimensional subspace.

**Gradient sketching finds a good low-dimensional subspace fast and provably.** First, we construct a low-dimensional parameter subspace $\mathcal{S}$ by sketching the model gradients. Sketching [17, 31] is a well-established dimensionality reduction tool known for affordable and accurate low-rank approximations [32, 33]. In deep learning, sketching recently extends its empirical applications to scalable estimations of influence functions for data selection [16, 34]. We make a first step toward the theoretical guarantee of gradient sketching for data selection: *gradient sketching efficiently finds a low-dimensional subspace $\mathcal{S}$ with small bias such that selecting $n$ samples by reducing variance over $\mathcal{S}$ is sufficient to preserve the fast-rate generalization $O(\dim(\mathcal{S})/n)$, linear in the low intrinsic dimension $\dim(\mathcal{S})$ while independent of the high parameter dimension $r$.*

**Moment matching in low dimension selects data that control the variance.** Second, we select data that reduce variance in the low-dimensional subspace $\mathcal{S}$ via moment matching. The variance of data selection is characterized by matching between the sketched gradient moments of the original

---

[3] Most classical variance reduction methods are tailored only for low-dimensional settings with $n > r$. Here, we intuitively refer to "variance reduction" as the direct extension of low-dimensional variance reduction to high dimensions, without careful control of the bias (vide Section 4.1 for some concretizations).

and selected datasets, $\widetilde{\boldsymbol{\Sigma}}, \widetilde{\boldsymbol{\Sigma}}_S$, formally $\mathrm{tr}(\widetilde{\boldsymbol{\Sigma}}\widetilde{\boldsymbol{\Sigma}}_S^\dagger)$. This objective involves optimizing over the inversions of (potentially) ill-conditioned matrices, leading to a challenging discrete optimization problem [28, 35]. Under a common heuristic assumption that $\widetilde{\boldsymbol{\Sigma}}, \widetilde{\boldsymbol{\Sigma}}_S$ commute [36, 37], we introduce a continuous relaxation with a quadratic objective and linear constraints that is numerically stable (free of pseudoinverse) and can be efficiently optimized via projected gradient descent.

The contributions of this work are summarized as follows:

- We provide a rigorous generalization analysis on data selection for finetuning, illustrating the critical role of dimensionality by unveiling the variance-bias tradeoff in high dimensions.

- We show that gradient sketching provably finds a low-dimensional parameter subspace $\mathcal{S}$ with small bias, reducing variance over which preserves the fast-rate generalization $O(\dim(\mathcal{S})/n)$. Techniques used in analyzing gradient sketching for data selection are agnostic to the selection method or the finetuning setting and could be of independent interest.

- We introduce SkMM, a scalable two-stage data selection method for finetuning that simultaneously "explores" the high-dimensional parameter space via gradient sketching and "exploits" the information in the low-dimensional subspace via moment matching.

## 1.1 Related Works

**Coreset selection and low-rank approximations.** From the variance-bias tradeoff perspective, data selection for high-dimensional finetuning can be viewed as a combination of (i) variance reduction in coreset selection for linear regression [17, 18, 19, 21, 23, 24] with low-dimensional features, and (ii) bias reduction via sample-wise low-rank approximation for high-dimensional matrices [38, 39, 40, 41, 42, 43, 44, 45, 46].

**Gradient sketching.** Gradient sketching [17, 32] based on Johnson-Lindenstrauss transforms (JLTs) [31, 47] has achieved impressive recent successes in efficient data selection [16] and attribution [34]. Despite the empirical success, theoretical understanding of the effect of gradient sketching on generalization remains limited. We make a first step toward this in the context of data selection leveraging existing theories on sketching (vide Remark 3.1 and Appendix C).

**Moment matching and optimal experimental design.** Moment matching is an intuitive idea for selecting low-dimensional data (*i.e.*, overdetermined with coreset size $n$ larger than data/representation dimension $r$), bearing various objectives like the A/V-optimality [28, 29] from optimal experimental design (OED) [25, 26, 27]. While classical OED studies the overdetermined scenario with $n \geq r$, efforts have been made to extend the notion of V-optimality beyond the overdetermined setting [48, 49]. Nevertheless, these works focus on the general overparametrized setting without considering potential special structures in data. In the context of data selection, this can lead to pessimistic sample complexity, especially for learning problems with low-intrinsic dimensions.

For multimodal contrastive learning, recent works [12, 13] illustrated the effectiveness of moment matching via tailored data selection criteria for CLIP [50]. Distinct from our setting of general finetuning in both low and high dimensions, these works focus on data filtering (with $n > r$) for pretraining from scratch.

**(Unsupervised) data selection.** In this work, we focus on unsupervised data selection that instead of relying on labels[4], leverages the geometry of the feature space and aims to select samples that are spread out, with a broad spectrum of concretizations including herding [51, 52], k-center greedy [53], leverage score sampling [24, 54, 55], adaptive sampling [44, 56], and volume sampling [39, 41].

An inspiring recent work [57] investigates the generalization of weakly supervised data selection via independent sampling in the low- ($n \to \infty$ with fixed $r$) and high-dimensional ($n, r \to \infty$ with $n/r \to$ constant) asymptotics. Instead of the asymptotic regime, we consider a realistic setting with finite $n$ and $r$, without specific assumptions on the data/feature distribution other than the low intrinsic dimension. Along this line, (weakly) supervised data selection commonly make choices based on the uncertainty [58, 59, 60] or sensitivity of the loss to samples (*e.g.*, influence function [4, 61, 62], sensitivity scores [5, 63, 64, 65], and heuristics based on losses and their gradients [66, 67, 68, 69]).

---

[4]From the theory perspective, data selection for finetuning is less sensitive to labels compared to training from scratch, especially given suitable pre-trained models with reasonable zero-shot accuracy (*e.g.*, Assumption 2.2).

## 1.2 Notations

Given any $n \in \mathbb{Z}_+$, we denote $[n] = \{1, \cdots, n\}$. Let $\mathbf{e}_n$ be the $n$-th canonical basis of the conformable dimension; $\mathbf{I}_n$ be the $n \times n$ identity matrix; and $\mathbf{0}_n, \mathbf{1}_n \in \mathbb{R}^n$ being vectors with all entries equal to zero and one, respectively. Let $\mathbb{S}^{n-1} := \{\mathbf{x} \in \mathbb{R}^n | \|\mathbf{x}\|_2 = 1\}$ be the unit sphere in $\mathbb{R}^n$, and $\Delta_n := \{\mathbf{p} \in [0,1]^b \mid \|\mathbf{p}\|_1 = 1\}$ be the dimension-$n$ probability simplex. We adapt the standard asymptotic notations: for any functions $f, g : \mathbb{R}_+ \to \mathbb{R}_+$, we write $f = O(g)$ or $f \lesssim g$ if there exists some constant $C > 0$ such that $f(x) \leq Cg(x)$ for all $x \in \mathbb{R}_+$; $f = \Omega(g)$ or $f \gtrsim g$ if $g = O(f)$; $f \asymp g$ if $f = O(g)$ and $f = \Omega(g)$. For any matrix $\mathbf{A} \in \mathbb{R}^{n \times d}$, let $s_1(\mathbf{A}) \geq \cdots \geq s_{\text{rank}(\mathbf{A})}(\mathbf{A}) \geq 0$ be the singular values; and $\mathbf{A}^\dagger$ be the Moore-Penrose pseudoinverse. Additionally for any $k \leq \text{rank}(\mathbf{A})$, let $\langle \mathbf{A} \rangle_k = \text{argmin}_{\mathbf{B}: \text{rank}(\mathbf{B}) \leq k} \|\mathbf{A} - \mathbf{B}\|_F$ be the optimal rank-$k$ approximation of $\mathbf{A}$ (characterized by the rank-$k$ truncated SVD). For any symmetric matrices $\mathbf{A}, \mathbf{B} \in \mathbb{R}^{d \times d}$, we write $\mathbf{A} \succcurlyeq \mathbf{B}$ or $\mathbf{A} - \mathbf{B} \succcurlyeq 0$ if $\mathbf{A} - \mathbf{B}$ is positive semidefinite.

# 2 Data Selection for Finetuning

Given a data space $\mathcal{X} \subseteq \mathbb{R}^d$ and a label space $\mathcal{Y} \subseteq \mathbb{R}$, let $\mathcal{D} = \{(\mathbf{x}_i, y_i) \in \mathcal{X} \times \mathcal{Y} \mid i \in [N]\}$ be a large dataset, with matrix form $(\mathbf{X}, \mathbf{y}) \in \mathbb{R}^{N \times d} \times \mathbb{R}^N$, for some downstream task where the performance is measured by a loss function $\ell : \mathcal{Y} \times \mathcal{Y} \to \mathbb{R}_{\geq 0}$.

**Finetuning.** Let $\mathcal{F}$ be a class of prediction functions where each $f = h \circ \phi \in \mathcal{F}$ can be expressed as the composition of an expressive representation function $\phi$ and a prediction head $h$. We consider a pre-trained model $\phi$ that yields high-quality representations for some downstream tasks on $\mathcal{D}$ and denote $\mathcal{F}_{|\phi} \subseteq \mathcal{F}$ as the class of finetuned models based on $\phi$. Assume that for every $(\mathbf{x}_i, y_i) \in \mathcal{D}$, $y_i \sim P(y \mid \mathbf{x}_i)$ *i.i.d.* such that there exists $f_*^\phi \in \mathcal{F}_{|\phi}$ with respect to $\phi$ satisfying (i) $\mathbb{E}[y_i \mid \phi(\mathbf{x}_i)] = f_*^\phi(\mathbf{x}_i)$, and (ii) $\mathbb{V}[y_i \mid \phi(\mathbf{x}_i)] \leq \sigma^2$ for some $\sigma > 0$ (which will be formalized later in respective settings).

**Data selection.** Instead of finetuning on the entire dataset $\mathcal{D}$, we aim to select a small coreset $\mathcal{D}_S \subseteq \mathcal{D}$ of size $n \ll N$ where the generalization is close. Precisely, let $\mathcal{D}_S$ be indexed by $S \subset [N]$ and denoted as $(\mathbf{X}_S, \mathbf{y}_S) \in \mathbb{R}^{n \times d} \times \mathbb{R}^n$. With $\mathcal{L}_{\mathcal{D}_S}(f) = \frac{1}{n} \sum_{(\mathbf{x},y) \in \mathcal{D}_S} \ell(f(\mathbf{x}), y)$ and a regularization $\mathcal{R} : \mathcal{F}_{|\phi} \to \mathbb{R}_{\geq 0}$ associated with a hyperparameter $\alpha \geq 0$, we want $f_S = \text{argmin}_{f \in \mathcal{F}_{|\phi}} \mathcal{L}_{\mathcal{D}_S}(f) + \alpha \cdot \mathcal{R}(f)$ to provide a low excess risk over $\mathcal{D}$: $\text{ER}(f_S) := \frac{1}{N} \sum_{i=1}^N \ell(f_S(\mathbf{x}_i), f_*^\phi(\mathbf{x}_i))$.

## 2.1 Low-dimensional Linear Probing: Variance Minimization

Warming up with linear probing, we concretize the general assumption on the ground truth (*i.e.*, $\mathbb{E}[y_i \mid \phi(\mathbf{x}_i)] = f_*^\phi(\mathbf{x}_i) = \phi(\mathbf{x}_i)^\top \boldsymbol{\theta}_*$ and $\mathbb{V}[y_i \mid \phi(\mathbf{x}_i)] \leq \sigma^2$) as follows:

**Assumption 2.1** (linear probing ground truth). *Assume* $\mathbf{y} = \phi(\mathbf{X})\boldsymbol{\theta}_* + \mathbf{z}$ *for some* $\boldsymbol{\theta}_* \in \mathbb{R}^r$ *where* $\mathbf{z} = [z_1, \cdots, z_N]^\top \in \mathbb{R}^N$ *consists of* i.i.d. *entries with* $\mathbb{E}[\mathbf{z}] = \boldsymbol{0}_N$ *and* $\mathbb{E}[\mathbf{z}\mathbf{z}^\top] \preccurlyeq \sigma^2 \mathbf{I}_N$.

Consider the pre-trained representations $\phi(\mathbf{X}) \in \mathbb{R}^{N \times r}$ and $\phi(\mathbf{X}_S) \in \mathbb{R}^{n \times r}$ with respective moments $\boldsymbol{\Sigma}^\phi := \frac{1}{N} \phi(\mathbf{X})^\top \phi(\mathbf{X})$ and $\boldsymbol{\Sigma}_S^\phi := \frac{1}{n} \phi(\mathbf{X}_S)^\top \phi(\mathbf{X}_S)$. For low-dimensional linear probing with $r \leq n$ (*s.t.* $\text{rank}(\boldsymbol{\Sigma}_S^\phi) = r$), the linear regression $\boldsymbol{\theta}_S = \text{argmin}_{\boldsymbol{\theta}} \frac{1}{n} \|\phi(\mathbf{X}_S)\boldsymbol{\theta} - \mathbf{y}_S\|_2^2$ has a unique solution with excess risk $\text{ER}(\boldsymbol{\theta}_S) = \|\boldsymbol{\theta}_S - \boldsymbol{\theta}_*\|_{\boldsymbol{\Sigma}^\phi}^2$[5] controlled by $\boldsymbol{\Sigma}^\phi$ and $\boldsymbol{\Sigma}_S^\phi$, analogous to the V-optimality criterion [28, 29] in optimal experimental design:

$$\mathbb{E}[\text{ER}(\boldsymbol{\theta}_S)] \leq \frac{\sigma^2}{n} \text{tr}(\boldsymbol{\Sigma}^\phi (\boldsymbol{\Sigma}_S^\phi)^{-1}), \tag{1}$$

If $\mathcal{D}_S$ satisfies $\boldsymbol{\Sigma}^\phi \preccurlyeq c_S \boldsymbol{\Sigma}_S^\phi$ for some $c_S \geq \frac{n}{N}$, then $\mathbb{E}[\text{ER}(\boldsymbol{\theta}_S)] \leq c_S \frac{\sigma^2 r}{n}$ (proof in Appendix B.1), where $c_S$ characterizes the variance controlled by $\mathcal{D}_S$, *i.e.*, smaller $c_S$ implies lower variance.

Despite its simplicity, uniform sampling is often observed in practice to serve as a strong baseline for data selection [1], especially when $n$ is large. In the low-dimensional linear probing scenario, (1) provides a theoretical justification for such effectiveness of uniform sampling:

---

[5]For any $\mathbf{u} \in \mathbb{R}^r$, $\|\mathbf{u}\|_{\boldsymbol{\Sigma}^\phi} := \sqrt{\mathbf{u}^\top \boldsymbol{\Sigma}^\phi \mathbf{u}}$ is the seminorm associated with $\boldsymbol{\Sigma}^\phi \succcurlyeq 0$.

**Proposition 2.1** (Uniform sampling for low-dimensional linear probing (Appendix B.2)). *Assume there exists (i) $B_\phi > 0$ such that $\|\phi(\mathbf{x})\|_2 \leq B_\phi \ \forall \ \mathbf{x} \in \mathcal{D}$; and (ii) $\gamma > 0$ with $\mathbf{\Sigma}^\phi \succcurlyeq \gamma \mathbf{I}_r$. For $S$ sampled uniformly (with replacement) over $\mathcal{D}$, with probability at least $1 - \delta$ over $S$, $\mathbf{\Sigma}^\phi \preccurlyeq c_S \mathbf{\Sigma}^\phi_S$ for any $c_S > 1$ if $n \gtrsim \frac{B_\phi^4}{\gamma^2} \cdot \frac{r + \log(1/\delta)}{(1 - 1/c_S)^2}$.*

That is, for linear probing with sufficiently low dimension $r \ll n$, under mild regularity assumptions on data, uniform sampling enjoys a near-optimal generalization $O(r/n)$.

## 2.2 High-dimension Finetuning with Low Intrinsic Dimension: Variance-Bias Tradeoff

Extending the analysis to general finetuning, we consider a set of $r$ finetuning parameters $\boldsymbol{\theta} \in \mathbb{R}^r$[6] (potentially with $r \gg n$) over a pre-trained model $\phi$ (*e.g.*, $\boldsymbol{\theta}$ can be the parameters of the last layer (*i.e.*, linear probing), last few layers, the entire network, or the LoRA [70] matrices).

Let $\mathcal{F}_{|\phi} = \left\{ f^\phi(\cdot; \boldsymbol{\theta}) : \mathcal{X} \to \mathbb{R} \mid \boldsymbol{\theta} \in \mathbb{R}^r \right\}$ be the finetuning function class. Without loss of generality, we assume zero initialization of $\boldsymbol{\theta}$ such that $f^\phi(\cdot; \mathbf{0}_r)$ corresponds to the pre-trained model. Analogous to the assumption in [16], under locality constraint on $\boldsymbol{\theta}$ (*e.g.*, $\|\boldsymbol{\theta}\|_2 < 1$), the dynamics of finetuning falls in the kernel regime [71] where $f^\phi$ can be approximated by its first-order Taylor expansion: $f^\phi(\mathbf{x}; \boldsymbol{\theta}) \approx f^\phi(\mathbf{x}; \mathbf{0}_r) + \nabla_{\boldsymbol{\theta}} f^\phi(\mathbf{x}; \mathbf{0}_r)^\top \boldsymbol{\theta}$. Then, we formalize the ground truth as follows:

**Assumption 2.2** (Finetuning ground truth). *Given the pre-trained $\phi$, there exists a bounded ground truth $\boldsymbol{\theta}_* \in \mathbb{R}^r$ with $\|\boldsymbol{\theta}_*\|_2 < 1$ such that for all $(\mathbf{x}, y) \in \mathcal{D}$, (i) $\mathbb{E}[y \mid \phi(\mathbf{x})] = f^\phi_*(\mathbf{x}) = f^\phi(\mathbf{x}; \boldsymbol{\theta}_*)$, and (ii) $\mathbb{V}[y \mid \phi(\mathbf{x})] \leq \sigma^2$ for some $\sigma > 0$.*

Intuitively, Assumption 2.2 implies that the pre-trained model $f^\phi(\cdot; \mathbf{0}_r)$ has a reasonable zero-shot performance. Given any $S \subset [N]$ with $|S| = n$, let $f^\phi(\mathbf{X}_S; \boldsymbol{\theta}) \in \mathbb{R}^n$ and $\nabla_{\boldsymbol{\theta}} f^\phi(\mathbf{X}_S; \boldsymbol{\theta}) \in \mathbb{R}^{n \times r}$ be the evaluation of $f^\phi(\mathbf{x}; \boldsymbol{\theta})$ and its Jacobian over $\mathbf{X}_S$ at $\boldsymbol{\theta}$. We observe that with $\mathbf{z} := \mathbf{y} - f^\phi(\mathbf{X}; \boldsymbol{\theta}_*)$, Assumption 2.2 implies $\mathbf{y} - f^\phi(\mathbf{X}; \mathbf{0}_r) \approx \mathbf{G}\boldsymbol{\theta}_* + \mathbf{z}$ where $\mathbb{E}[\mathbf{z}] = \mathbf{0}_N$ and $\mathbb{E}[\mathbf{z}\mathbf{z}^\top] \preccurlyeq \sigma^2 \mathbf{I}_N$; while $\mathbf{G} := \nabla_{\boldsymbol{\theta}} f^\phi(\mathbf{X}; \mathbf{0}_r) \in \mathbb{R}^{N \times r}$ is the Jacobian over $\mathcal{D}$ at initialization.

Then in the kernel regime [71], the finetuning objective $\min_{\boldsymbol{\theta} \in \mathbb{R}^r} \frac{1}{n} \left\| f^\phi(\mathbf{X}_S; \boldsymbol{\theta}) - \mathbf{y}_S \right\|_2^2 + \alpha \|\boldsymbol{\theta}\|_2^2$ can be well approximated by a ridge regression problem:

$$\boldsymbol{\theta}_S = \operatorname*{argmin}_{\boldsymbol{\theta} \in \mathbb{R}^r} \frac{1}{n} \left\| \nabla_{\boldsymbol{\theta}} f^\phi(\mathbf{X}_S; \mathbf{0}_r) \boldsymbol{\theta} - \left(\mathbf{y}_S - f^\phi(\mathbf{X}_S; \mathbf{0}_r)\right) \right\|_2^2 + \alpha \|\boldsymbol{\theta}\|_2^2. \tag{2}$$

Recall $\mathbf{G} := \nabla_{\boldsymbol{\theta}} f^\phi(\mathbf{X}; \mathbf{0}_r) \in \mathbb{R}^{N \times r}$ and $\mathbf{G}_S := \nabla_{\boldsymbol{\theta}} f^\phi(\mathbf{X}_S; \mathbf{0}_r) \in \mathbb{R}^{n \times r}$. With the moments $\mathbf{\Sigma}^\phi = \frac{1}{N} \mathbf{G}^\top \mathbf{G}$ and $\mathbf{\Sigma}^\phi_S = \frac{1}{n} \mathbf{G}_S^\top \mathbf{G}_S$, the excess risk $\mathrm{ER}(\boldsymbol{\theta}_S) = \|\boldsymbol{\theta}_S - \boldsymbol{\theta}_*\|_{\mathbf{\Sigma}^\phi}^2$ satisfies[7]:

**Theorem 2.2** (Main result I: variance-bias tradeoff (Appendix B.3)). *Given $S$, let $\mathbf{P}_\mathcal{S} \in \mathbb{R}^{r \times r}$ be an orthogonal projector onto some subspace $\mathcal{S} \subseteq \mathrm{Range}(\mathbf{\Sigma}^\phi_S)$, and $\mathbf{P}_\mathcal{S}^\perp = \mathbf{I}_r - \mathbf{P}_\mathcal{S}$ be its orthogonal complement. Under Assumption 2.1, there exists an $\alpha > 0$ such that (2) satisfies $\mathbb{E}[\mathrm{ER}(\boldsymbol{\theta}_S)] \leq$ **variance + bias** with (i) **variance** $= \frac{2\sigma^2}{n} \mathrm{tr}(\mathbf{\Sigma}^\phi(\mathbf{P}_\mathcal{S}\mathbf{\Sigma}^\phi_S\mathbf{P}_\mathcal{S})^\dagger)$ and (ii) **bias** $= 2\,\mathrm{tr}\left(\mathbf{\Sigma}^\phi\mathbf{P}_\mathcal{S}^\perp\right) \|\boldsymbol{\theta}_*\|_2^2$.*

Specifically, the variance-bias tradeoff is controlled by the unknown $\mathcal{S}$: expanding $\mathcal{S}$ leads to higher variance but lower bias. Reducing the generalization gap involves finding a suitable $\mathcal{S}$ in the high-dimensional parameter space, a computationally challenging problem addressed in Section 3.1.

It is worth highlighting that Theorem 2.2 encapsulates both the low- and high-dimensional finetuning. For low-dimensional linear probing, (1) is a special case of Theorem 2.2 (up to constants) with $\mathbf{P}_\mathcal{S} = \mathbf{I}_r$. While in high dimension, an intrinsic low-dimensional structure (*e.g.*, Assumption 2.3) is necessary for the effectiveness of data selection[8].

---

[6]Notice that $r$ is the dimension of the finetuning parameter space. For linear probing in Section 2.1, $r$ is the same as the pre-trained representation dimension; but for general finetuning, $r$ can be much larger.

[7]Notice that for linear probing, $f^\phi(\mathbf{x}; \boldsymbol{\theta}) = f^\phi(\mathbf{x}; \mathbf{0}_r) + \nabla_{\boldsymbol{\theta}} f^\phi(\mathbf{x}; \mathbf{0}_r)^\top \boldsymbol{\theta}$ with $\nabla_{\boldsymbol{\theta}} f^\phi(\mathbf{x}; \mathbf{0}_r) = \phi(\mathbf{x})$. Therefore, the finetuning objective can be exactly formulated as (2), and the excess risk of high-dimensional linear probing satisfies Theorem 2.2 with $\mathbf{\Sigma}^\phi := \frac{1}{N} \phi(\mathbf{X})^\top \phi(\mathbf{X})$ and $\mathbf{\Sigma}^\phi_S := \frac{1}{n} \phi(\mathbf{X}_S)^\top \phi(\mathbf{X}_S)$.

[8]Otherwise, if all directions of $\mathrm{Range}(\mathbf{\Sigma}^\phi)$ are equally important, with $n \ll r$, $\boldsymbol{\theta}_S$ learned from $\mathcal{D}_S$ necessarily fails to capture the orthogonal complement of $\mathrm{Range}(\mathbf{\Sigma}^\phi_S)$ and therefore $\mathbb{E}[\mathrm{ER}(\boldsymbol{\theta}_S)] \gtrsim r - n$.

**Assumption 2.3** (Low intrinsic dimension). *Consider the second moment $\boldsymbol{\Sigma}^\phi \succcurlyeq 0$ over $\mathcal{D}$ with $N$ samples. Let $\bar{r} := \min\{t \in [r] \mid \operatorname{tr}\left(\boldsymbol{\Sigma}^\phi - \langle\boldsymbol{\Sigma}^\phi\rangle_t\right) \leq \operatorname{tr}\left(\boldsymbol{\Sigma}^\phi\right)/N\}$ be the intrinsic dimension. Assume that $\boldsymbol{\Sigma}^\phi$ has a low intrinsic dimension: $\bar{r} \ll \min\{N, r\}$.*

When the high-dimensional finetuning parameter space has a low intrinsic dimension $\bar{r} \ll \min\{N, r\}$, Theorem 2.2 can be further concretized with suitable $\mathcal{D}_S$ and associated $\mathcal{S}$:

**Corollary 2.3** (Exploitation + exploration (Appendix B.3)). *Under the same setting as Theorem 2.2 and Assumption 2.3, if $S$ satisfies for some subspace $\mathcal{S} \subseteq \operatorname{Range}(\boldsymbol{\Sigma}_S^\phi)$ with $\operatorname{rank}(\mathbf{P}_\mathcal{S}) \asymp \bar{r}$ and $c_S \geq \frac{n}{N}$ that (i) $\mathbf{P}_\mathcal{S}(c_S\boldsymbol{\Sigma}_S^\phi - \boldsymbol{\Sigma}^\phi)\mathbf{P}_\mathcal{S} \succcurlyeq 0$ and (ii) $\operatorname{tr}(\boldsymbol{\Sigma}^\phi\mathbf{P}_\mathcal{S}^\perp) \leq \frac{N}{n}\operatorname{tr}(\boldsymbol{\Sigma}^\phi - \langle\boldsymbol{\Sigma}^\phi\rangle_{\bar{r}})$, then[9]*

$$\mathbb{E}\left[\operatorname{ER}\left(\boldsymbol{\theta}_S\right)\right] \leq \textbf{\textit{variance}} + \textbf{\textit{bias}} \lesssim \frac{1}{n}\left(c_S\sigma^2\bar{r} + \operatorname{tr}\left(\boldsymbol{\Sigma}^\phi\right)\|\boldsymbol{\theta}_*\|_2^2\right). \tag{3}$$

In particular, with $c_S, \sigma \lesssim 1$, $\|\boldsymbol{\theta}_*\|_2^2 < 1$, and $\operatorname{tr}(\boldsymbol{\Sigma}^\phi) \asymp \bar{r}$ (depending only on the low intrinsic dimension), the generalization achieves a fast rate $O(\bar{r}/n)$, independent of $r \gg \bar{r}$.

In (3), (i) **bias** is reduced by exploring the parameter space for an $\mathcal{S}$ with small low-rank approximation error $\operatorname{tr}(\boldsymbol{\Sigma}^\phi\mathbf{P}_\mathcal{S}^\perp) \leq \frac{1}{n}\operatorname{tr}(\boldsymbol{\Sigma}^\phi)$; while (ii) **variance** is reduced by exploiting information in $\mathcal{S}$ through moment matching, $\mathbf{P}_\mathcal{S}(c_S\boldsymbol{\Sigma}_S^\phi - \boldsymbol{\Sigma}^\phi)\mathbf{P}_\mathcal{S} \succcurlyeq 0$, where smaller $c_S$ means better exploitation.

## 3 Sketchy Moment Matching

A gap between Corollary 2.3 and practice is *how to find a suitable $\mathcal{S}$ efficiently in the high-dimensional parameter space*. In this section, we introduce a simple scalable algorithm for constructing $\mathcal{S}$ and $\mathcal{D}_S$ that satisfies the exploration and exploitation conditions in Corollary 2.3.

### 3.1 Find Low Intrinsic Dimension via Gradient Sketching

For high-dimensional finetuning with $r \gg n$, a critical limit of Theorem 2.2 and Corollary 2.3 is that the large moment matrices $\boldsymbol{\Sigma}^\phi, \boldsymbol{\Sigma}_S^\phi$ are not invertible, storable, or even directly computable, due to the prohibitive cost. As a remedy, sketching [17, 32] via Johnson-Lindenstrauss transforms [31] is a classical dimensionality reduction strategy that gets increasing recent attention for gradient approximation in large-scale machine learning problems [16, 34][10].

**Remark 3.1** (Gradient sketching). *In the high-dimensional setting with $r \gg n$, to reduce the dimensionality of the gradients $\mathbf{G} = \nabla_{\boldsymbol{\theta}}f^\phi\left(\mathbf{X};\boldsymbol{0}_r\right) \in \mathbb{R}^{N \times r}$ with a low intrinsic dimension $\bar{r} \ll \min\{N, r\}$ (Assumption 2.3), we draw a Johnson-Lindenstrauss transform [31] (JLT, formally in Definition C.1) $\boldsymbol{\Gamma} \in \mathbb{R}^{r \times m}$ that projects the dimension-$r$ gradients to a lower dimension $m \asymp \bar{r} \ll r$: $\widetilde{\mathbf{G}} = \mathbf{G}\boldsymbol{\Gamma} \in \mathbb{R}^{N \times m}$. One of the most common constructions of JLT is the* Gaussian embedding *(i.e., a Gaussian random matrix with* i.i.d. *entries $\boldsymbol{\Gamma}_{ij} \sim \mathcal{N}(0, 1/m)$ discussed in Lemma C.3, vide Remark C.1 for a brief overview of various (fast) JLTs and their efficiency).*

While sketching is known for preserving Euclidean distances [31] and providing accurate low-rank approximations [17, 32, 33], *whether gradient sketching can convert Theorem 2.2 to an efficiently computable form without compromising the generalization guarantee?* We answer this question affirmatively with the following theorem.

**Theorem 3.1** (Main result II: gradient sketching (formally in Theorem C.1)). *Under Assumption 2.2 and 2.3 with a low intrinsic dimension $\bar{r} \ll \min\{N, r\}$, draw a Gaussian embedding $\boldsymbol{\Gamma} \in \mathbb{R}^{r \times m}$ (Lemma C.3) with $m \geq 11\bar{r}$. Let $\widetilde{\boldsymbol{\Sigma}}^\phi := \boldsymbol{\Gamma}^\top\boldsymbol{\Sigma}^\phi\boldsymbol{\Gamma}$ and $\widetilde{\boldsymbol{\Sigma}}_S^\phi := \boldsymbol{\Gamma}^\top\boldsymbol{\Sigma}_S^\phi\boldsymbol{\Gamma}$ be the sketched gradient moments. For any $\mathcal{D}_S$ with $n > m$ samples such that $\operatorname{rank}(\boldsymbol{\Sigma}_S^\phi) = n$, and the $\lceil 1.1\bar{r}\rceil$-th largest eigenvalue $s_{\lceil 1.1\bar{r}\rceil}(\widetilde{\boldsymbol{\Sigma}}_S^\phi) \geq \gamma_S$ for some $\gamma_S > 0$, with probability at least $0.9$ over $\boldsymbol{\Gamma}$, there exists $\alpha > 0$ where (2) satisfies $\mathbb{E}\left[\operatorname{ER}\left(\boldsymbol{\theta}_S\right)\right] \lesssim \textbf{\textit{variance}} + \textbf{\textit{sketching error}} + \textbf{\textit{bias}}$ with (i) $\textbf{\textit{variance}} = \frac{\sigma^2}{n}\operatorname{tr}(\widetilde{\boldsymbol{\Sigma}}^\phi(\widetilde{\boldsymbol{\Sigma}}_S^\phi)^\dagger)$, (ii) $\textbf{\textit{sketching error}} = \frac{\sigma^2}{n}\frac{1}{m\gamma_S}\|\widetilde{\boldsymbol{\Sigma}}^\phi(\widetilde{\boldsymbol{\Sigma}}_S^\phi)^\dagger\|_2\operatorname{tr}(\boldsymbol{\Sigma}^\phi)$, and (iii) $\textbf{\textit{bias}} = \frac{1}{n}\|\widetilde{\boldsymbol{\Sigma}}^\phi(\widetilde{\boldsymbol{\Sigma}}_S^\phi)^\dagger\|_2\operatorname{tr}(\boldsymbol{\Sigma}^\phi)\|\boldsymbol{\theta}_*\|_2^2$.*

---

[9]We note that in contrast to the classical slow rate $O(1/\sqrt{n})$ in low dimension (when $n > r$), ridge regression on $\mathcal{D}_S$ in the high-dimensional finetuning (with $n < r$) achieves a fast rate $O(1/n)$. This is granted by the low-rankness of $\boldsymbol{\Sigma}_S^\phi$, which enables a more fine-grained analysis of the regularization (vide Appendix B.3).

[10]We highlight a key nuance here: for fast influence function approximation in [16, 34], the gradient of the *loss function* is sketched, whereas in our setting, we sketch the gradient of the *pre-trained model* $f^\phi(\mathbf{x};\boldsymbol{0}_r)$.

*If $S$ further satisfies $\widetilde{\boldsymbol{\Sigma}}^\phi \preccurlyeq c_S \widetilde{\boldsymbol{\Sigma}}_S^\phi$ for some $c_S \geq \frac{n}{N}$, with $m = \max\{\sqrt{\mathrm{tr}\,(\boldsymbol{\Sigma}^\phi)/\gamma_S}, 11\overline{r}\}$,*

$$\mathbb{E}\left[\mathrm{ER}\left(\boldsymbol{\theta}_S\right)\right] \lesssim \textbf{\textit{variance}} + \textbf{\textit{sketching error}} + \textbf{\textit{bias}} \lesssim \frac{c_S}{n}\left(\sigma^2 m + \mathrm{tr}\left(\boldsymbol{\Sigma}^\phi\right)\|\boldsymbol{\theta}_*\|_2^2\right). \quad (4)$$

Comparing (4) with (3), we observe that by controlling the variance with $\widetilde{\boldsymbol{\Sigma}}^\phi \preccurlyeq c_S \widetilde{\boldsymbol{\Sigma}}_S^\phi$ in low dimension $m \asymp \overline{r} \ll r$, gradient sketching preserves the fast-rate generalization $O(m/n) = O(\overline{r}/n)$ up to constants. That is, gradient sketching implicitly finds a random subspace $\mathcal{S} \subseteq \mathrm{Range}(\boldsymbol{\Sigma}_S^\phi)$ (vide (9)) that satisfies the exploration assumption in Corollary 2.3. Meanwhile, the choice of sketching size $m$ balances the tradeoff between **variance** and **sketching error**: a larger $m$ reduces the sketching error at the cost of higher variance. Such tradeoff is optimized at $m = \sqrt{\mathrm{tr}\,(\boldsymbol{\Sigma}^\phi)/\gamma_S}$.

## 3.2 Control Variance via Moment Matching

Given the intrinsic low-dimensional structure with small bias in Section 3.1, Theorem 3.1 connects generalization to the variance controlled by the matching between $\widetilde{\boldsymbol{\Sigma}}^\phi$ and $\widetilde{\boldsymbol{\Sigma}}_S^\phi$. Specifically, when the selected data $\mathcal{D}_S$ satisfies $\widetilde{\boldsymbol{\Sigma}}^\phi \preccurlyeq c_S \widetilde{\boldsymbol{\Sigma}}_S^\phi$ for some $c_S \geq \frac{n}{N}$, we have $\mathrm{tr}(\widetilde{\boldsymbol{\Sigma}}^\phi (\widetilde{\boldsymbol{\Sigma}}_S^\phi)^\dagger) \leq c_S m$ and $\|\widetilde{\boldsymbol{\Sigma}}^\phi (\widetilde{\boldsymbol{\Sigma}}_S^\phi)^\dagger\|_2 \leq c_S$ upper bounded, leading to the fast-rate generalization in (4).

---

**Algorithm 3.1** Sketchy Moment Matching (SkMM)

---

1: **Input:** $f^\phi\left(\cdot; \mathbf{0}_r\right), n \ll N, m < n, c_S \in [\frac{n}{N}, 1]$.
2: Draw a (fast) Johnson-Lindenstrauss transform $\boldsymbol{\Gamma} \in \mathbb{R}^{r \times m}$ (Remark 3.1).
3: Compute gradient sketching $\widetilde{\mathbf{G}} = \nabla_{\boldsymbol{\theta}} f^\phi\left(\mathbf{X}; \mathbf{0}_r\right)\boldsymbol{\Gamma} \in \mathbb{R}^{N \times m}$. (Remark 3.4)
4: Compute the spectral decomposition of $\widetilde{\boldsymbol{\Sigma}}^\phi = \frac{1}{N}\widetilde{\mathbf{G}}^\top \widetilde{\mathbf{G}} \succcurlyeq 0$: $\widetilde{\boldsymbol{\Sigma}}^\phi = \mathbf{V}\boldsymbol{\Lambda}\mathbf{V}^\top$ where
   (a) $\mathbf{V} = [\mathbf{v}_1, \cdots, \mathbf{v}_m] \in \mathbb{R}^{m \times m}$ consists of the orthonormal eigenvectors, and
   (b) $\boldsymbol{\Lambda} = \mathrm{diag}\left(\lambda_1, \cdots, \lambda_m\right)$ contains descending eigenvalues $\lambda_1 \geq \cdots \geq \lambda_m \geq 0$.
5: Initialize $\mathbf{s} = [s_1, \cdots, s_N]$ with $s_i = \frac{1}{n}$ on $n$ uniformly sampled $i$'s and $s_i = 0$ elsewhere.
6: Let $\mathrm{diag}(\mathbf{s}) \in \mathbb{R}^{N \times N}$ be a diagonal matrix with $\mathbf{s}$ on diagonal. Optimizing:

$$\min_{\mathbf{s} \in \Delta_N} \min_{\boldsymbol{\gamma} = [\gamma_1, \cdots, \gamma_m] \in \mathbb{R}^m} \sum_{j=1}^m \left(\mathbf{v}_j^\top \widetilde{\mathbf{G}}^\top \mathrm{diag}\left(\mathbf{s}\right)\widetilde{\mathbf{G}}\mathbf{v}_j - \gamma_j \cdot \lambda_j\right)^2 \quad (5)$$

$$\textit{s.t.} \quad 0 \leq s_i \leq 1/n \ \forall\, i \in [N], \quad \gamma_j \geq 1/c_S \ \forall\, j \in [m].$$

7: **Output:** $S \subset [N]$ by sampling $n$ data from $\mathbf{s} \in \Delta_N$ without replacement.

---

While directly minimizing $\mathrm{tr}(\widetilde{\boldsymbol{\Sigma}}^\phi (\widetilde{\boldsymbol{\Sigma}}_S^\phi)^\dagger)$ involves integer programming and pseudoinverse, causing hard and numerically unstable optimization, $\widetilde{\boldsymbol{\Sigma}}^\phi \preccurlyeq c_S \widetilde{\boldsymbol{\Sigma}}_S^\phi$ has a straightforward relaxation (vide Remark 3.2), leading to the simple and stable moment matching objective (5) in Algorithm 3.1.

**Remark 3.2** (Relaxing $\widetilde{\boldsymbol{\Sigma}}^\phi \preccurlyeq c_S \widetilde{\boldsymbol{\Sigma}}_S^\phi$ to (5))**.** *Given the spectral decomposition $\widetilde{\boldsymbol{\Sigma}}^\phi = \mathbf{V}\boldsymbol{\Lambda}\mathbf{V}^\top$, $\widetilde{\boldsymbol{\Sigma}}^\phi \preccurlyeq c_S \widetilde{\boldsymbol{\Sigma}}_S^\phi$ can be rewritten as $\mathbf{V}^\top (\frac{1}{n}\widetilde{\mathbf{G}}_S^\top \widetilde{\mathbf{G}}_S)\mathbf{V} \succcurlyeq \frac{1}{c_S}\boldsymbol{\Lambda}$, and (5) is a relaxation: (i) instead of enforcing $\widetilde{\boldsymbol{\Sigma}}^\phi \preccurlyeq c_S \widetilde{\boldsymbol{\Sigma}}_S^\phi$ strictly, constraints are only imposed on the diagonal[11]: $\mathbf{v}_j^\top (\frac{1}{n}\widetilde{\mathbf{G}}_S^\top \widetilde{\mathbf{G}}_S)\mathbf{v}_j \geq \lambda_j/c_S$, $j \in [m]$; and (ii) the selection of $S$ is relaxed to a weight vector $\mathbf{s} \in \Delta_N$ with linear constraints $0 \leq s_i \leq 1/n$. Free of integer constraints and pseudoinverse, the quadratic data selection objective with linear constraints in (5) can be solved efficiently and stably via projected gradient descent.*

Alternative to the moment matching heuristic in Remark 3.2, variance reduction by controlling $\widetilde{\boldsymbol{\Sigma}}^\phi (\widetilde{\boldsymbol{\Sigma}}_S^\phi)^\dagger$ in the low-dimensional subspace can be realized via various methods, including leverage score sampling [18, 19, 72, 73, 74] and V-optimal experimental design [28, 29]. We provide brief discussions on these alternatives in Appendix A.2.

**Remark 3.3** ($c_S$ controls strength of moment matching)**.** *In Algorithm 3.1, smaller $c_S$ enforces $\widetilde{\boldsymbol{\Sigma}}_S^\phi$ to exploit more information in $\widetilde{\boldsymbol{\Sigma}}^\phi$, bringing lower variance and better generalization. While the lower bound $c_S \geq \frac{n}{N}$ could be tight (vide Remark B.1), in practice, the smallest feasible $c_S$ depends on the data distribution and tends to be larger (e.g., $c_S \approx 1$ in the experiments).*

---

[11] This is equivalent to assuming that $\widetilde{\boldsymbol{\Sigma}}^\phi, \widetilde{\boldsymbol{\Sigma}}_S^\phi$ commute. While such assumption does not hold in general, it is a valuable heuristic whose effectiveness has been demonstrated in various domains [36, 37].

**Remark 3.4** (Computational efficiency of SkMM). *SkMM is efficient in both memory and computation. Consider the two stages in Algorithm 3.1: (i) Gradient sketching can be computed in parallel with input-sparsity time and on the fly without storing the (potentially) high-dimensional gradients (vide Remark C.1). (ii) After gradient sketching, variance reduction via moment matching happens in the low dimension $m$, with a low memory footprint $O(Nm)$, taking $O(m^3)$ for the spectral decomposition and $O(Nm)$ per iteration for optimizing the moment matching objective* (5).

## 4 Experiments

### 4.1 Synthetic High-dimensional Linear Probing

To ground the theoretical insight on variance-bias tradeoff in high-dimensional finetuning, we simulate linear probing with a synthetic underdetermined ridge regression problem[12].

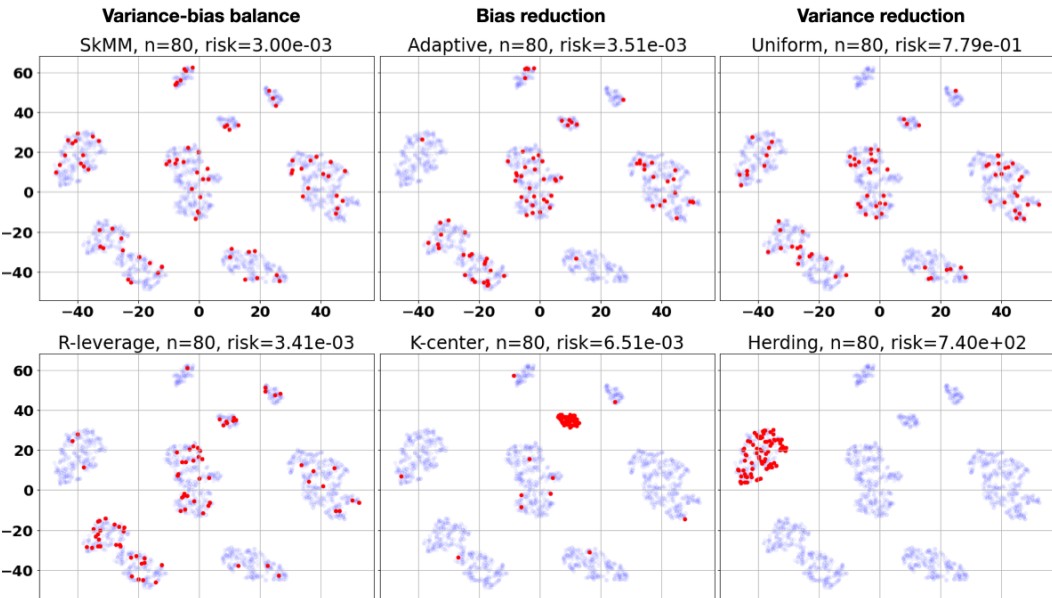

Figure 2: Selecting $n = 80$ data (colored in red) from the GMM dataset. Intuitively, a coreset $\mathcal{D}_S$ with low bias contains at least one sample per cluster; whereas a low-variance $\mathcal{D}_S$ selects more data from clusters with larger variance. We recall from Theorem 2.2 that the variance-bias balance is essential for good generalization.

**Setup.** We consider a set of $N = 2000$ samples with high-dimensional pre-trained representations $\phi(\mathbf{X}) \in \mathbb{R}^{N \times r}$, $r = 2400$, modeled by a Gaussian mixture model (GMM) consisting of $\bar{r} = 8$ well-separated clusters, each with random sizes and variances (vide Figure 2). Samples within each cluster share the same randomly generated label. We solve the ridge regression problem (2) over the selected coreset of $n$ samples with hyperparameter $\alpha$ tuning. The empirical risk is evaluated over the full dataset $\mathcal{L}_\mathcal{D}(\boldsymbol{\theta}_S) = \frac{1}{N} \|\phi(\mathbf{X})\boldsymbol{\theta}_S - \mathbf{y}\|_2^2$ (vide Appendix D.1 for implementation details).

**Data selection.** For SkMM (Algorithm 3.1), we use a sketching dimension $m = 4\bar{r} = 32$ and set $c_S = 0.999$. We optimize (5) via Adam [75] with constraint projection under learning rate $10^{-7}$ for $10^4$ iterations and sample $S$ from $\mathbf{s} \in \Delta_N$ with the lowest objective value.

We compare SkMM to representative unsupervised data selection methods for regression, including uniform, leverage score [18, 19, 72, 73, 74], adaptive sampling [44, 56], herding [51, 52], and k-center greedy [53]. Specifically, (i) `SkMM`, truncated leverage score (`T-leverage`), and ridge leverage score sampling (`R-leverage`) can be viewed as different ways of variance-bias balancing; (ii) adaptive sampling (`Adaptive`) and k-center greedy (`K-center`) focus on bias reduction (*i.e.*, providing good low-rank approximation/clustering for $\phi(\mathbf{X})$); while (iii) `Herding` and uniform sampling (`Uniform`) reduce variance (vide Appendix D.2 for baseline details).

We observe from Figure 2 and Table 1 that balancing the variance-bias tradeoff is crucial for the generalization of data selection in high dimensions. In particular, SkMM achieves the best empirical

---

[12]Our experiment code is available at https://github.com/Xiang-Pan/sketchy_moment_matching

Table 1: Empirical risk $\mathcal{L}_{\mathcal{D}}(\boldsymbol{\theta}_S)$ on the GMM dataset at various $n$, under the same hyperparameter tuning where ridge regression over the full dataset $\mathcal{D}$ with $N = 2000$ samples achieves $\mathcal{L}_{\mathcal{D}}(\boldsymbol{\theta}_{[N]}) =$ **2.95e-3**. For methods involving sampling, results are reported over 8 random seeds.

| $n$ | 48 | 64 | 80 | 120 | 400 | 800 | 1600 |
|---|---|---|---|---|---|---|---|
| Herding | 7.40e+2 | 7.40e+2 | 7.40e+2 | 7.40e+2 | 7.38e+2 | 1.17e+2 | **2.95e-3** |
| Uniform | $(1.14 \pm 2.71)$e-1 | $(1.01 \pm 2.75)$e-1 | $(3.44 \pm 0.29)$e-3 | $(3.13 \pm 0.14)$e-3 | $(2.99 \pm 0.03)$e-3 | $\mathbf{(2.96 \pm 0.01)}$e-3 | $\mathbf{(2.95 \pm 0.00)}$e-3 |
| K-center | $(1.23 \pm 0.40)$e-2 | $(9.53 \pm 0.60)$e-2 | $(1.12 \pm 0.45)$e-2 | $(2.73 \pm 1.81)$e-2 | $(5.93 \pm 4.80)$e-2 | $(1.18 \pm 0.64)$e-1 | $(1.13 \pm 0.70)$e+0 |
| Adaptive | $(3.81 \pm 0.65)$e-2 | $(3.79 \pm 1.37)$e-3 | $(4.83 \pm 1.90)$e-3 | $(4.03 \pm 1.35)$e-3 | $(3.40 \pm 0.67)$e-3 | $(7.34 \pm 3.97)$e-3 | $(3.19 \pm 0.16)$e-3 |
| T-leverage | $(0.99 \pm 1.65)$e-2 | $(3.63 \pm 0.49)$e-3 | $(3.30 \pm 0.30)$e-3 | $(3.24 \pm 0.14)$e-3 | $\mathbf{(2.98 \pm 0.01)}$e-3 | $\mathbf{(2.96 \pm 0.01)}$e-3 | $\mathbf{(2.95 \pm 0.00)}$e-3 |
| R-leverage | $(4.08 \pm 1.58)$e-3 | $(3.48 \pm 0.43)$e-3 | $(3.25 \pm 0.31)$e-3 | $(3.09 \pm 0.06)$e-3 | $(3.00 \pm 0.02)$e-3 | $(2.97 \pm 0.01)$e-3 | $\mathbf{(2.95 \pm 0.00)}$e-3 |
| SkMM | $\mathbf{(3.54 \pm 0.51)}$**e-3** | $\mathbf{(3.31 \pm 0.15)}$**e-3** | $\mathbf{(3.12 \pm 0.07)}$**e-3** | $\mathbf{(3.07 \pm 0.08)}$**e-3** | $\mathbf{(2.98 \pm 0.02)}$**e-3** | $\mathbf{(2.96 \pm 0.01)}$**e-3** | $\mathbf{(2.95 \pm 0.00)}$**e-3** |

risk across different coreset sizes $n$, especially when $n$ is small. While as $n/N \to 1$, uniform sampling provides a strong baseline, coinciding with common empirical observations [1].

## 4.2 Experiments on Regression Tasks

We further validate the effectiveness of SkMM on UTKFace [76], a real-world regression dataset for age estimation. We finetune a randomly initialized classification head on top of the feature representation of CLIP [50] with Adam [75] and learning rate $10^{-1}$. We also retain those baselines from the above synthetic setup in this experiment.

Table 2: Mean Absolute Error (the lower the better) on UTKFace with a linear regressor trained on top of frozen features from a pre-trained CLIP (ViT-B/32). We use the **bold** font to indicate the best method for each coreset size.

| Method | 100 | 200 | 500 | 1000 | 2000 | 3000 |
|---|---|---|---|---|---|---|
| Uniform Sampling | $10.55 \pm 3.09$ | $8.94 \pm 3.48$ | $6.09 \pm 0.42$ | $4.70 \pm 0.23$ | $3.92 \pm 0.16$ | $3.68 \pm 0.15$ |
| Adaptive | $6.02 \pm 0.53$ | $4.75 \pm 0.14$ | $4.40 \pm 0.14$ | N/A | N/A | N/A |
| Greedy | $10.40 \pm 1.21$ | $7.56 \pm 0.18$ | $6.43 \pm 0.09$ | $5.51 \pm 0.19$ | $4.87 \pm 0.03$ | $4.37 \pm 0.08$ |
| Herding | $17.57 \pm 0.01$ | $13.41 \pm 0.01$ | $8.47 \pm 0.01$ | $5.79 \pm 0.01$ | $4.19 \pm 0.01$ | $3.53 \pm 0.01$ |
| R-leverage | $\mathbf{5.44 \pm 0.01}$ | $4.79 \pm 0.02$ | $4.36 \pm 0.01$ | $\mathbf{3.86 \pm 0.01}$ | $3.61 \pm 0.01$ | $3.53 \pm 0.04$ |
| SkMM | $5.57 \pm 0.38$ | $\mathbf{4.70 \pm 0.05}$ | $\mathbf{4.23 \pm 0.20}$ | $4.02 \pm 0.11$ | $\mathbf{3.54 \pm 0.19}$ | $\mathbf{3.25 \pm 0.10}$ |

The results for linear probing are provided in Table 2, where our method remarkably outperforms comparative baselines on UTKFace. For every coreset size, SkMM improves the performance of CLIP compared to uniform sampling. Especially for small coreset size $n = 100, 200$, it achieves a Mean Absolute Error reduction of approximately $50\%$.

## 4.3 Experiments on Image Classification Tasks

While our analysis focuses on data selection for finetuning regression models, a natural question is whether the idea of SkMM applies to broader scopes. To answer this, we extend our empirical investigation to classification. In particular, we consider an imbalanced classification task: Stanford-Cars [77] with 196 classes, 8144 training samples, and 8041 testing samples where the classes are highly imbalanced with training sample sizes ranging from 24 to 68.

**Finetuning.** We consider two common ways of finetuning: (i) linear probing (LP) over the last layer and (ii) funetuning (FT) over the last few layers, covering both the low- (*i.e.*, $n \geq r$ for LP) and high-dimensional (*i.e.*, $r > n$ for FT) settings. For LP, we learn the last layer over the embeddings from a CLIP-pretrained ViT-B/32 [50] with a learning rate of $10^{-1}$. For FT[13], we finetuning the last two layers of an ImageNet-pretrained ResNet18 [84] with a learning rate of $10^{-2}$. In both settings, we optimize via Adam for 50 epochs. Due to space limit constraints, detailed results for fine-tuning are deferred to the appendix.

**Data selection.** For SkMM-LP, the gradients (of the last layer) are given by the pretrained features from CLIP. For SkMM-FT, the gradients (of the last two layers) are calculated based on a random classification head. We tune the sketching dimension $m \in \{32, 64, 128, 256, 512\}$ and the lower bound for slackness variables $c_S \in \{0.6, 0.7, 0.8, 0.9\}$. Within suitable ranges, smaller $m$ and larger $c_S$ lead to better performance in the low data regime. Intuitively, smaller $m$ encourages variance reduction in a more compressed subspace, and larger $c_s$ leads to easier optimization.

---

[13]We notice that finetuning the last few layers of strong pretrained models like CLIP can distort the features and hurt the performance, as studied in [83]. Therefore, we stay with a weaker pretrained model for finetuning.

Table 3: Accuracy and F1 score (%) of LP over CLIP on StanfordCars

| | $n$ | 2000 | 2500 | 3000 | 3500 | 4000 |
|---|---|---|---|---|---|---|
| Uniform Sampling | Acc | $67.63 \pm 0.17$ | $70.59 \pm 0.19$ | $72.49 \pm 0.19$ | $74.16 \pm 0.22$ | $75.40 \pm 0.16$ |
| | F1 | $64.54 \pm 0.18$ | $67.79 \pm 0.23$ | $70.00 \pm 0.20$ | $71.77 \pm 0.23$ | $73.14 \pm 0.12$ |
| Herding [51] | Acc | $67.22 \pm 0.16$ | $71.02 \pm 0.13$ | $73.17 \pm 0.22$ | $74.64 \pm 0.18$ | $75.71 \pm 0.29$ |
| | F1 | $64.07 \pm 0.23$ | $68.28 \pm 0.15$ | $70.64 \pm 0.28$ | $72.22 \pm 0.26$ | $73.26 \pm 0.39$ |
| Contextual Diversity [78] | Acc | $67.64 \pm 0.13$ | $70.82 \pm 0.23$ | $72.66 \pm 0.12$ | $74.46 \pm 0.17$ | $75.77 \pm 0.12$ |
| | F1 | $64.51 \pm 0.17$ | $68.18 \pm 0.25$ | $70.05 \pm 0.11$ | $72.13 \pm 0.15$ | $73.35 \pm 0.07$ |
| Glister [79] | Acc | $67.60 \pm 0.24$ | $70.85 \pm 0.27$ | $73.07 \pm 0.26$ | $74.63 \pm 0.21$ | $76.00 \pm 0.20$ |
| | F1 | $64.50 \pm 0.34$ | $68.07 \pm 0.38$ | $70.47 \pm 0.35$ | $72.18 \pm 0.25$ | $73.69 \pm 0.24$ |
| GraNd [66] | Acc | $67.27 \pm 0.07$ | $70.38 \pm 0.07$ | $72.56 \pm 0.05$ | $74.67 \pm 0.06$ | $75.77 \pm 0.12$ |
| | F1 | $64.04 \pm 0.09$ | $67.48 \pm 0.09$ | $69.81 \pm 0.08$ | $72.13 \pm 0.05$ | $73.44 \pm 0.13$ |
| Forgetting [80] | Acc | $67.59 \pm 0.10$ | $70.99 \pm 0.05$ | $72.54 \pm 0.07$ | $74.81 \pm 0.05$ | $75.74 \pm 0.01$ |
| | F1 | $64.85 \pm 0.13$ | $68.53 \pm 0.07$ | $70.30 \pm 0.05$ | $72.59 \pm 0.04$ | $73.74 \pm 0.02$ |
| DeepFool [81] | Acc | $67.77 \pm 0.29$ | $70.73 \pm 0.22$ | $73.24 \pm 0.22$ | $74.57 \pm 0.23$ | $75.71 \pm 0.15$ |
| | F1 | $64.16 \pm 0.68$ | $68.49 \pm 0.53$ | $70.93 \pm 0.32$ | $72.44 \pm 0.27$ | $73.79 \pm 0.15$ |
| Entropy [82] | Acc | $67.95 \pm 0.11$ | $71.00 \pm 0.10$ | $73.28 \pm 0.10$ | $75.02 \pm 0.08$ | $75.82 \pm 0.06$ |
| | F1 | $64.55 \pm 0.10$ | $67.95 \pm 0.12$ | $70.68 \pm 0.12$ | $72.46 \pm 0.12$ | $73.29 \pm 0.04$ |
| Margin [82] | Acc | $67.53 \pm 0.14$ | $71.19 \pm 0.09$ | $73.09 \pm 0.14$ | $74.66 \pm 0.11$ | $75.57 \pm 0.13$ |
| | F1 | $64.16 \pm 0.15$ | $68.33 \pm 0.14$ | $70.37 \pm 0.17$ | $72.03 \pm 0.11$ | $73.14 \pm 0.20$ |
| Least Confidence [82] | Acc | $67.68 \pm 0.11$ | $70.99 \pm 0.14$ | $73.04 \pm 0.05$ | $74.65 \pm 0.09$ | $75.58 \pm 0.08$ |
| | F1 | $64.09 \pm 0.20$ | $68.03 \pm 0.20$ | $70.30 \pm 0.07$ | $72.02 \pm 0.10$ | $73.15 \pm 0.12$ |
| SkMM-LP | Acc | $\mathbf{68.27 \pm 0.03}$ | $\mathbf{71.53 \pm 0.05}$ | $\mathbf{73.61 \pm 0.02}$ | $\mathbf{75.12 \pm 0.01}$ | $\mathbf{76.34 \pm 0.02}$ |
| | F1 | $\mathbf{65.29 \pm 0.03}$ | $\mathbf{68.75 \pm 0.06}$ | $\mathbf{71.14 \pm 0.03}$ | $\mathbf{72.64 \pm 0.02}$ | $\mathbf{74.02 \pm 0.10}$ |

We compare SkMM to various unsupervised and (weakly) supervised data selection methods for classification, including uniform sampling, herding [51], Contextual Diversity [78], Glister [79], GraNd [66], Forgetting [80], DeepFool [81], as well as three uncertainty-based methods, Entropy, Margin, and Least Confidence [82].

**Observations.** We first observe that for both LP (Table 3) and FT (Table 4), SkMM achieves competitive finetuning accuracy on StanfordCars. Since SkMM is an unsupervised process agnostic of true class sizes, the appealing performance of SkMM on the imbalanced StanfordCars dataset echoes the ability of SkMM to handle data selection among clusters of various sizes through variance-bias balance (*cf.* synthetic experiments in Figure 2). Meanwhile, for LP in the low-dimensional setting (Table 3), uniform sampling provides a surprisingly strong baseline. This coincides with the theoretical insight from Proposition 2.1 and the empirical observations in [1].

## 5    Discussion, Limitations, and Future Directions

We investigated data selection for finetuning in both low and high dimensions from a theoretical perspective. Beyond variance reduction in low dimension, our analysis revealed the *variance-bias tradeoff in data selection for high-dimensional finetuning with low intrinsic dimension $\bar{r}$*, balancing which led to a *fast-rate generalization $O(\bar{r}/n)$*. For efficient control of such variance-bias tradeoff in practice, we introduced SkMM that first explores the high-dimensional parameter space via *gradient sketching* and then exploits the resulting low-dimensional subspace via *moment matching*. Theoretically, we showed that *the low-dimensional subspace from gradient sketching preserves the fast-rate generalization*. Moreover, we ground the theoretical insight on balancing the variance-bias tradeoff via synthetic experiments, while demonstrating the effectiveness of SkMM for finetuning real vision tasks.

In this work, we focus only on moment matching via optimization inspired by the analysis for variance reduction after gradient sketching. Nevertheless, there is a remarkable variety of existing low-dimensional data selection strategies (*e.g.*, via greedy selection or sampling) that could potentially be extended to high dimensions leveraging sketching as an efficient pre-processing step. In linear algebra, sketching has been widely studied for accelerating, as well as stabilizing, large-scale low-rank approximations and linear solvers. However, the intuitions and theories there may or may not be directly applicable to the statistical learning regime. In light of the high-dimensional nature of deep learning where sketching brings an effective remedy, we hope that providing a rigorous generalization analysis for sketching in data selection would make a step toward bridging the classical wisdom of sketching and the analogous challenges in modern learning problems.

## Acknowledgments

The authors wish to thank Yunzhen Feng, Julia Kempe, and Christopher Musco for insightful discussions. QL was partially supported by the NYU Research Catalyst Prize and the Department of Energy under ASCR Award DE-SC0024721. YD was supported by the NYU Courant Instructorship.

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

# A  Additional Discussions

## A.1  Additional Notations

Given any matrix $\mathbf{A} \in \mathbb{R}^{n \times d}$, along with indices $i \in [n]$, $j \in [d]$, $I \subseteq [n]$, and $J \subseteq [d]$, let $[\mathbf{A}]_{i,j}$ be the $(i,j)$-th entry of $\mathbf{A}$, $[\mathbf{A}]_i$ be the $i$-th row (or the $i$-th entry if $\mathbf{A} \in \mathbb{R}^n$ is a vector), and $[\mathbf{A}]_{:,j}$ be the $j$-th column; $\mathbf{A}_I = [\mathbf{A}]_{I,:}$ consists of rows in $\mathbf{A}$ indexed by $I$; and let $\mathbf{A}_{I,J} = [\mathbf{A}]_{I,J}$ be the submatrix of $\mathbf{A}$ with rows indexed by $I$ and columns indexed by $J$.

## A.2  Alternatives to Moment Matching Heuristic in Remark 3.2

In addition to the moment matching heuristic in Remark 3.2, variance in the resulting low-dimensional subspace from gradient sketching can be controlled by $\widetilde{\boldsymbol{\Sigma}}^\phi (\widetilde{\boldsymbol{\Sigma}}_S^\phi)^\dagger$ via alternative methods like leverage score sampling and V-optimal experimental design.

**Remark A.1** (Leverage score sampling). *Leverage score sampling [18, 19, 72, 73, 74] provides arguably one of the most intuitive ways for selecting data based on $\widetilde{\mathbf{G}} \in \mathbb{R}^{N \times m}$. In particular, [17, Theorem 17] implies that for a coreset of size at least $n = \Omega(m \log(m/\delta)\epsilon^{-2})$ drawn i.i.d. with replacement via leverage score sampling over $\widetilde{\mathbf{G}}$, $c_S \leq (1 + \epsilon) \frac{m}{\tau_S N}$ with probability at least $1 - \delta$, where $\tau_S \in [0, 1]$ is the minimum leverage score of $\widetilde{\mathbf{G}}$ over the coreset $S$.[14] Such dependence on $\tau_S$ can render the upper bound of $c_S$ vacuous when $\tau_S \to 0$.*

*Nevertheless, when $\tau_S$ is reasonably large, leverage score sampling based on $\widetilde{\mathbf{G}}$ can be computed more efficiently than SkMM in $O(Nm^2)$ time and can provide good control over $c_S$. While both SkMM and leverage score sampling can facilitate variance reduction in the low-dimensional subspace, SkMM provides better empirical performance (cf. Section 4.1) at a slightly higher cost in the low intrinsic dimension $m$ (vide Remark 3.4) as it is tailored for optimizing moment matching.*

**Remark A.2** (V-optimal experimental design). *Variance in the low-dimensional subspace can also be controlled by applying the V-optimal experimental design methods [28, 29] on $\widetilde{\mathbf{G}} \in \mathbb{R}^{N \times m}$. For example, [29] provides a polynomial-time algorithm to find a $(1 + \epsilon)$-estimation of the V-optimal design for $\widetilde{\mathbf{G}}$ with a coreset of size at least $n = \Omega(m\epsilon^{-2})$; and $c_S$ is effectively controlled by the V-optimality criterion $\mathrm{tr}(\widetilde{\boldsymbol{\Sigma}}^\phi (\widetilde{\boldsymbol{\Sigma}}_S^\phi)^\dagger)$.*

*While such V-optimal design methods can provide good control over $c_S$ with nearly optimal sample complexity, they are computationally more expensive than SkMM (or leverage score sampling) and tend to suffer from numerical instability issues in practice. For example, the algorithm in [29] consists of two stages: (i) solving a continuous relaxation of the original discrete optimization problem posed by V-optimality, and (ii) rounding the continuous solution via regret minimization. While the cost of rounding is negligible, solving the continuous relaxation of V-optimality (in contrast to leveraging fast and stable heuristics like the one in SkMM, cf. Remark 3.2) is challenging, both in terms of computational complexity and numerical stability.*

# B  Proofs for Section 2.1

## B.1  Proofs of (1)

*Proof of* (1) *and beyond.* Under the assumption $\mathrm{rank}\,(\phi\,(\mathbf{X}_S)) = r$, both $\phi\,(\mathbf{X}_S)$, $\phi\,(\mathbf{X})$ have full column rank. Therefore $\phi\,(\mathbf{X}_S)^\dagger \phi\,(\mathbf{X}_S) = \phi\,(\mathbf{X})^\dagger \phi\,(\mathbf{X}) = \mathbf{I}_r$, and $\boldsymbol{\theta}_S = \phi\,(\mathbf{X}_S)^\dagger \mathbf{y}_S$. Then, since $\mathbf{y} = \phi\,(\mathbf{X})\,\boldsymbol{\theta}_* + \mathbf{z}$ and $\mathbf{y}_S = \phi\,(\mathbf{X}_S)\,\boldsymbol{\theta}_* + \mathbf{z}_S$, we have

$$\boldsymbol{\theta}_S - \boldsymbol{\theta}_* = \phi\,(\mathbf{X}_S)^\dagger \mathbf{y}_S - \boldsymbol{\theta}_* = \left( \phi\,(\mathbf{X}_S)^\dagger \phi\,(\mathbf{X}_S)\,\boldsymbol{\theta}_* - \boldsymbol{\theta}_* \right) + \phi\,(\mathbf{X}_S)^\dagger \mathbf{z}_S = \phi\,(\mathbf{X}_S)^\dagger \mathbf{z}_S,$$

---

[14]Notice that $\tau_S$ appears because samples in $S$ are equally weighted in the data selection setting, in contrast to the standard leverage score sampling where samples are weighted by the respective sampling probabilities.

which leads to

$$
\begin{aligned}
\mathbb{E}\left[\mathrm{ER}\left(\boldsymbol{\theta}_S\right)\right] &= \mathbb{E}\left[\frac{1}{N}\left\|\phi\left(\mathbf{X}\right)\left(\boldsymbol{\theta}_S - \boldsymbol{\theta}_*\right)\right\|_2^2\right] \\
&= \mathrm{tr}\left(\left(\frac{1}{N}\phi\left(\mathbf{X}\right)^\top \phi\left(\mathbf{X}\right)\right)\phi\left(\mathbf{X}_S\right)^\dagger \mathbb{E}\left[\mathbf{z}_S\mathbf{z}_S^\top\right]\left(\phi\left(\mathbf{X}_S\right)^\dagger\right)^\top\right) \\
&= \sigma^2\,\mathrm{tr}\left(\left(\frac{1}{N}\phi\left(\mathbf{X}\right)^\top \phi\left(\mathbf{X}\right)\right)\left(\phi\left(\mathbf{X}_S\right)^\top \phi\left(\mathbf{X}_S\right)\right)^{-1}\right) \\
&= \frac{\sigma^2}{n}\,\mathrm{tr}\left(\boldsymbol{\Sigma}^\phi\left(\boldsymbol{\Sigma}_S^\phi\right)^{-1}\right).
\end{aligned}
$$

Now we explain the necessity of assuming $c_S \geq n/N$ for $\boldsymbol{\Sigma}^\phi \preccurlyeq c_S\boldsymbol{\Sigma}_S^\phi$:

**Remark B.1** (Lower bound of $c_S$). *Since $\phi\left(\mathbf{X}\right)^\top \phi\left(\mathbf{X}\right) \succcurlyeq \phi\left(\mathbf{X}_S\right)^\top \phi\left(\mathbf{X}_S\right)$, we observe that $N\boldsymbol{\Sigma}^\phi \succcurlyeq n\boldsymbol{\Sigma}_S^\phi$, which implies $\boldsymbol{\Sigma}^\phi \succcurlyeq \frac{n}{N}\boldsymbol{\Sigma}_S^\phi$. Therefore, $\boldsymbol{\Sigma}^\phi \preccurlyeq c_S\boldsymbol{\Sigma}_S^\phi$ is only possible when $c_S \geq n/N$. Notice that this lower bound of $c_S$ is tight, e.g. when $\widetilde{\mathbf{G}}$ consists of $N - n$ rows of zeros.*

**Low-dimensional linear probing with moment matching.** Recall from (1) that $\mathbb{E}\left[\mathrm{ER}\left(\boldsymbol{\theta}_S\right)\right] = \frac{\sigma^2}{n}\,\mathrm{tr}\left(\boldsymbol{\Sigma}^\phi\left(\boldsymbol{\Sigma}_S^\phi\right)^{-1}\right)$. Further assuming a suitable selection of $\mathcal{D}_S$ with $\boldsymbol{\Sigma}^\phi \preccurlyeq c_S\boldsymbol{\Sigma}_S^\phi$, we have

$$
\mathrm{tr}\left(\boldsymbol{\Sigma}^\phi\left(\boldsymbol{\Sigma}_S^\phi\right)^{-1}\right) \leq c_S\,\mathrm{tr}\left(\mathbf{I}_r\right) = c_S r
$$

and therefore, $\mathbb{E}\left[\mathrm{ER}\left(\boldsymbol{\theta}_S\right)\right] \leq c_S\frac{\sigma^2 r}{n}$. $\qquad\square$

## B.2 Proof of Proposition 2.1

*Proof of Proposition 2.1.* Let $\widehat{\boldsymbol{\Sigma}}_S^\phi := \left(\boldsymbol{\Sigma}^\phi\right)^{-1/2}\boldsymbol{\Sigma}_S^\phi\left(\boldsymbol{\Sigma}^\phi\right)^{-1/2}$. The goal of $\boldsymbol{\Sigma}^\phi \preccurlyeq c_S\boldsymbol{\Sigma}_S^\phi$ can be re-expressed as $c_S\widehat{\boldsymbol{\Sigma}}_S^\phi \succcurlyeq \mathbf{I}_r$, or equivalently when $c_S > 1$, $\left\|\widehat{\boldsymbol{\Sigma}}_S^\phi - \mathbf{I}_r\right\|_2 \leq 1 - \frac{1}{c_S}$. With uniform sampling, since

$$
\mathbb{E}_S\left[\boldsymbol{\Sigma}_S^\phi\right] = \mathbb{E}_S\left[\frac{1}{n}\sum_{\mathbf{x}\in S}\phi\left(\mathbf{X}\right)\phi\left(\mathbf{X}\right)^\top\right] = \mathbb{E}_\mathbf{x}\left[\phi\left(\mathbf{X}\right)\phi\left(\mathbf{X}\right)^\top\right] = \boldsymbol{\Sigma}^\phi,
$$

we have $\mathbb{E}_S\left[\widehat{\boldsymbol{\Sigma}}_S^\phi\right] = \mathbf{I}_r$. For any fixed unit vector $\mathbf{z} \in \mathbb{S}^{r-1}$, let $Z_i := \mathbf{z}^\top\left(\boldsymbol{\Sigma}^\phi\right)^{-1/2}\phi\left(\mathbf{x}_i\right)$ be random variables with randomness on $i \in [N]$. Since $\|\phi(\mathbf{x})\|_2 \leq B_\phi\ \forall\ \mathbf{x} \in \mathcal{D}$ and $\boldsymbol{\Sigma}^\phi \succcurlyeq \gamma\mathbf{I}_r$, we observe that

$$
|Z_i| \leq \left\|\left(\boldsymbol{\Sigma}^\phi\right)^{-1/2}\phi\left(\mathbf{x}_i\right)\right\|_2 \leq \frac{B_\phi}{\sqrt{\gamma}}\quad \forall\ i \in [N]
$$

is bounded. Therefore, $Z_i$ is $\left(\frac{B_\phi^2}{\gamma}\right)$-subGaussian, and $\left(Z_i^2 - \mathbb{E}\left[Z_i^2\right]\right) = \mathbf{z}^\top\left(\widehat{\boldsymbol{\Sigma}}_S^\phi - \mathbf{I}_r\right)\mathbf{z}$ is $\left(16\frac{B_\phi^2}{\gamma}\right)$-subexponential. Then, by Bernstein's inequality [85, Theorem 2.8.2][86, Section 2.1.3], for any $0 < \epsilon_1 \leq 16B_\phi^2/\gamma$,

$$
\mathbf{P}\left[\mathbf{z}^\top\left(\widehat{\boldsymbol{\Sigma}}_S^\phi - \mathbf{I}_r\right)\mathbf{z} \geq \epsilon_1\right] \leq \exp\left(-\frac{n}{2}\cdot\frac{\epsilon_1^2\gamma^2}{16^2 B_\phi^4}\right). \tag{6}
$$

By recalling that $\left\|\widehat{\boldsymbol{\Sigma}}_S^\phi - \mathbf{I}_r\right\|_2 = \max_{\mathbf{u}\in\mathbb{S}^{r-1}}\mathbf{u}^\top\left(\widehat{\boldsymbol{\Sigma}}_S^\phi - \mathbf{I}_r\right)\mathbf{u}$, Equation (6) for a fixed $\mathbf{z} \in \mathbb{S}^{r-1}$ can be extended to the entire unit sphere $\mathbb{S}^{r-1}$ through an $\epsilon$-net argument as follows. Recall that for

any $\epsilon_2 > 0$, there exists an $\epsilon_2$-net $\mathcal{U} \subset \mathbb{S}^{r-1}$ such that $|\mathcal{U}| \leq \left(1 + \frac{2}{\epsilon_2}\right)^r$. Then, by the union bound,

$$\mathbb{P}\left[\max_{\mathbf{u} \in \mathcal{U}} \mathbf{u}^\top \left(\widehat{\mathbf{\Sigma}}_S^\phi - \mathbf{I}_r\right) \mathbf{u} > \epsilon_1\right] \leq \left(1 + \frac{2}{\epsilon_2}\right)^r \exp\left(-\frac{n}{2} \cdot \frac{\epsilon_1^2 \gamma^2}{16^2 B_\phi^4}\right)$$

$$= \exp\left(r \log\left(1 + \frac{2}{\epsilon_2}\right) - \frac{n}{2} \cdot \frac{\epsilon_1^2 \gamma^2}{16^2 B_\phi^4}\right).$$

That is, with probability at least $1 - \delta$, $\max_{\mathbf{u} \in \mathcal{U}} \mathbf{u}^\top \left(\widehat{\mathbf{\Sigma}}_S^\phi - \mathbf{I}_r\right) \mathbf{u} \leq \epsilon_1$ when

$$n \geq \frac{512 B_\phi^4}{\gamma^2 \epsilon_1^2} \left(r \log\left(1 + \frac{2}{\epsilon_2}\right) + \log\left(\frac{1}{\delta}\right)\right).$$

By the construction of the $\epsilon_2$-net $\mathcal{U}$, for all $\mathbf{v} \in \mathbb{S}^{r-1}$, there exists $\mathbf{u} \in \mathcal{U}$ such that $\|\mathbf{u} - \mathbf{v}\|_2 \leq \epsilon_2$. Therefore, for any $\mathbf{v} \in \mathbb{S}^{r-1}$, we have

$$\mathbf{v}^\top \left(\widehat{\mathbf{\Sigma}}_S^\phi - \mathbf{I}_r\right) \mathbf{v}$$

$$= \mathbf{u}^\top \left(\widehat{\mathbf{\Sigma}}_S^\phi - \mathbf{I}_r\right) \mathbf{u} + (\mathbf{v} - \mathbf{u})^\top \left(\widehat{\mathbf{\Sigma}}_S^\phi - \mathbf{I}_r\right) (\mathbf{v} - \mathbf{u}) + 2 (\mathbf{v} - \mathbf{u})^\top \left(\widehat{\mathbf{\Sigma}}_S^\phi - \mathbf{I}_r\right) \mathbf{u}$$

$$\leq \epsilon_1 + \left\|\widehat{\mathbf{\Sigma}}_S^\phi - \mathbf{I}_r\right\|_2 \left(\epsilon_2^2 + 2\epsilon_2\right),$$

which implies $\left\|\widehat{\mathbf{\Sigma}}_S^\phi - \mathbf{I}_r\right\|_2 \leq \frac{\epsilon_1}{2 - (1 + \epsilon_2)^2}$. By taking $\epsilon_2$ as a small constant (*e.g.*, $\epsilon_2 = \sqrt{3/2} - 1$), we have $\left\|\widehat{\mathbf{\Sigma}}_S^\phi - \mathbf{I}_r\right\|_2 \leq 1 - \frac{1}{c_S}$ when

$$n \gtrsim \frac{B_\phi^4}{\gamma^2} \cdot \frac{r + \log(1/\delta)}{(1 - 1/c_S)^2}.$$

$\qquad\qquad\qquad\qquad\qquad\qquad\qquad\qquad\qquad\qquad\qquad\qquad\qquad\qquad\qquad\qquad\qquad\qquad\qquad\quad\square$

### B.3 Proof of Theorem 2.2

*Proof of Theorem 2.2.* With $\mathbf{\Sigma}^\phi = \frac{1}{N} \mathbf{G}^\top \mathbf{G}$, we have

$$\mathbb{E}\left[\text{ER}\left(\boldsymbol{\theta}_S\right)\right] = \mathbb{E}\left[\frac{1}{N} \|\mathbf{G}\left(\boldsymbol{\theta}_S - \boldsymbol{\theta}_*\right)\|_2^2\right] = \mathbb{E}\left[\|\boldsymbol{\theta}_S - \boldsymbol{\theta}_*\|_{\mathbf{\Sigma}^\phi}^2\right],$$

Observing that by the optimality of $\boldsymbol{\theta}_S$, we have

$$\frac{2}{n} \mathbf{G}_S^\top \left(\mathbf{G}_S \boldsymbol{\theta}_S - \mathbf{y}_S\right) + 2\alpha \boldsymbol{\theta}_S = \mathbf{0}_r.$$

Recalling that $\mathbf{\Sigma}_S^\phi := \frac{1}{n} \mathbf{G}_S^\top \mathbf{G}_S$, this implies

$$\boldsymbol{\theta}_S = \left(\frac{1}{n} \mathbf{G}_S^\top \mathbf{G}_S + \alpha \mathbf{I}_r\right)^{-1} \frac{1}{n} \mathbf{G}_S^\top \mathbf{y}_S$$

$$= \frac{1}{n} \left(\mathbf{\Sigma}_S^\phi + \alpha \mathbf{I}_r\right)^{-1} \mathbf{G}_S^\top \left(\mathbf{G}_S \boldsymbol{\theta}_* + \mathbf{z}_S\right)$$

$$= \left(\mathbf{\Sigma}_S^\phi + \alpha \mathbf{I}_r\right)^{-1} \mathbf{\Sigma}_S^\phi \boldsymbol{\theta}_* + \frac{1}{n} \left(\mathbf{\Sigma}_S^\phi + \alpha \mathbf{I}_r\right)^{-1} \mathbf{G}_S^\top \mathbf{z}_S.$$

Therefore, with $\mathbb{E}_{\mathbf{z}}\left[\mathbf{z}\right] = \mathbf{0}_N$, $\mathbb{E}\left[\text{ER}\left(\boldsymbol{\theta}_S\right)\right]$ can be decomposed the bias term and variance terms as follows:

$$\mathbb{E}\left[\text{ER}\left(\boldsymbol{\theta}_S\right)\right] = \mathbb{E}\left[\|\boldsymbol{\theta}_S - \boldsymbol{\theta}_*\|_{\mathbf{\Sigma}^\phi}^2\right]$$

$$= \mathbb{E}_{\mathbf{z}}\left[\left\|\left(\left(\mathbf{\Sigma}_S^\phi + \alpha \mathbf{I}_r\right)^{-1} \mathbf{\Sigma}_S^\phi - \mathbf{I}_r\right) \boldsymbol{\theta}_* + \frac{1}{n} \left(\mathbf{\Sigma}_S^\phi + \alpha \mathbf{I}_r\right)^{-1} \mathbf{G}_S^\top \mathbf{z}_S\right\|_{\mathbf{\Sigma}^\phi}^2\right]$$

$$= \underbrace{\left\|\left(\left(\mathbf{\Sigma}_S^\phi + \alpha \mathbf{I}_r\right)^{-1} \mathbf{\Sigma}_S^\phi - \mathbf{I}_r\right) \boldsymbol{\theta}_*\right\|_{\mathbf{\Sigma}^\phi}^2}_{\text{Bias}} + \underbrace{\mathbb{E}_{\mathbf{z}}\left[\left\|\frac{1}{n} \left(\mathbf{\Sigma}_S^\phi + \alpha \mathbf{I}_r\right)^{-1} \mathbf{G}_S^\top \mathbf{z}_S\right\|_{\mathbf{\Sigma}^\phi}^2\right]}_{\text{Variance}}.$$

Since $\mathbb{E}_{\mathbf{z}}\left[\mathbf{z}_S \mathbf{z}_S^\top\right] \preccurlyeq \sigma^2 \mathbf{I}_n$, the variance term can be bounded as

$$\text{Variance} \leq \frac{\sigma^2}{n} \operatorname{tr}\left(\left(\mathbf{\Sigma}_S^\phi + \alpha \mathbf{I}_r\right)^{-1} \mathbf{\Sigma}^\phi \left(\mathbf{\Sigma}_S^\phi + \alpha \mathbf{I}_r\right)^{-1} \mathbf{\Sigma}_S^\phi\right)$$
$$\leq \frac{\sigma^2}{n} \operatorname{tr}\left(\mathbf{\Sigma}^\phi \left(\mathbf{\Sigma}_S^\phi + \alpha \mathbf{I}_r\right)^{-1}\right),$$

where the second inequality follows from the fact that $\left\|\left(\mathbf{\Sigma}_S^\phi + \alpha \mathbf{I}_r\right)^{-1} \mathbf{\Sigma}_S^\phi\right\|_2 \leq 1$.

Recall that $\mathbf{P}_S \in \mathbb{R}^{r \times r}$ is an orthogonal projector onto any subspace $S$ of $\operatorname{Range}\left(\mathbf{\Sigma}_S^\phi\right)$, and $\mathbf{P}_S^\perp = \mathbf{I}_r - \mathbf{P}_S$ is the orthogonal projector onto its orthogonal complement. By observing that $\mathbf{\Sigma}_S^\phi + \alpha \mathbf{I}_r \succcurlyeq \mathbf{P}_S \mathbf{\Sigma}_S^\phi \mathbf{P}_S + \alpha \mathbf{P}_S^\perp$, since $\operatorname{Range}(\mathbf{P}_S) \perp \operatorname{Range}(\mathbf{P}_S^\perp)$, we have

$$\left(\mathbf{\Sigma}_S^\phi + \alpha \mathbf{I}_r\right)^{-1} \preccurlyeq \left(\mathbf{P}_S \mathbf{\Sigma}_S^\phi \mathbf{P}_S\right)^\dagger + \frac{1}{\alpha} \mathbf{P}_S^\perp.$$

Therefore,

$$\text{Variance} \leq \frac{\sigma^2}{n} \left(\operatorname{tr}\left(\mathbf{\Sigma}^\phi \left(\mathbf{P}_S \mathbf{\Sigma}_S^\phi \mathbf{P}_S\right)^\dagger\right) + \frac{1}{\alpha} \operatorname{tr}\left(\mathbf{\Sigma}^\phi \mathbf{P}_S^\perp\right)\right).$$

For the bias part, we first observe that

$$\mathbf{I}_r - \left(\mathbf{\Sigma}_S^\phi + \alpha \mathbf{I}_r\right)^{-1} \mathbf{\Sigma}_S^\phi = \alpha \left(\mathbf{\Sigma}_S^\phi + \alpha \mathbf{I}_r\right)^{-1}.$$

Therefore,

$$\text{Bias} = \left\|\left(\left(\mathbf{\Sigma}_S^\phi + \alpha \mathbf{I}_r\right)^{-1} \mathbf{\Sigma}_S^\phi - \mathbf{I}_r\right) \boldsymbol{\theta}_*\right\|_{\mathbf{\Sigma}^\phi}^2 = \left\|\alpha \left(\mathbf{\Sigma}_S^\phi + \alpha \mathbf{I}_r\right)^{-1} \boldsymbol{\theta}_*\right\|_{\mathbf{\Sigma}^\phi}^2$$
$$= \alpha^2 \operatorname{tr}\left(\left(\mathbf{\Sigma}_S^\phi + \alpha \mathbf{I}_r\right)^{-1} \mathbf{\Sigma}^\phi \left(\mathbf{\Sigma}_S^\phi + \alpha \mathbf{I}_r\right)^{-1} \boldsymbol{\theta}_* \boldsymbol{\theta}_*^\top\right)$$
$$\leq \alpha^2 \operatorname{tr}\left(\mathbf{\Sigma}^\phi \left(\mathbf{\Sigma}_S^\phi + \alpha \mathbf{I}_r\right)^{-2}\right) \|\boldsymbol{\theta}_*\|_2^2.$$

Since $\left(\mathbf{\Sigma}_S^\phi + \alpha \mathbf{I}_r\right)^2 \succcurlyeq \left(\mathbf{P}_S \mathbf{\Sigma}_S^\phi \mathbf{P}_S + \alpha \mathbf{I}_r\right)^2 \succcurlyeq 2\alpha \cdot \mathbf{P}_S \mathbf{\Sigma}_S^\phi \mathbf{P}_S + \alpha^2 \mathbf{P}_S^\perp$, we have

$$\left(\mathbf{\Sigma}_S^\phi + \alpha \mathbf{I}_r\right)^{-2} \preccurlyeq \frac{1}{2\alpha} \left(\mathbf{P}_S \mathbf{\Sigma}_S^\phi \mathbf{P}_S\right)^\dagger + \frac{1}{\alpha^2} \mathbf{P}_S^\perp,$$

and thus

$$\text{Bias} \leq \left(\frac{\alpha}{2} \operatorname{tr}\left(\mathbf{\Sigma}^\phi \left(\mathbf{P}_S \mathbf{\Sigma}_S^\phi \mathbf{P}_S\right)^\dagger\right) + \operatorname{tr}\left(\mathbf{\Sigma}^\phi \mathbf{P}_S^\perp\right)\right) \|\boldsymbol{\theta}_*\|_2^2.$$

Combining the bias and variance terms, we have

$$\mathbb{E}\left[\text{ER}\left(\boldsymbol{\theta}_S\right)\right] \leq \frac{\sigma^2}{n} \left(\operatorname{tr}\left(\mathbf{\Sigma}^\phi \left(\mathbf{P}_S \mathbf{\Sigma}_S^\phi \mathbf{P}_S\right)^\dagger\right) + \frac{1}{\alpha} \operatorname{tr}\left(\mathbf{\Sigma}^\phi \mathbf{P}_S^\perp\right)\right)$$
$$+ \left(\frac{\alpha}{2} \operatorname{tr}\left(\mathbf{\Sigma}^\phi \left(\mathbf{P}_S \mathbf{\Sigma}_S^\phi \mathbf{P}_S\right)^\dagger\right) + \operatorname{tr}\left(\mathbf{\Sigma}^\phi \mathbf{P}_S^\perp\right)\right) \|\boldsymbol{\theta}_*\|_2^2$$
$$\leq \frac{\sigma^2}{n} \operatorname{tr}\left(\mathbf{\Sigma}^\phi \left(\mathbf{P}_S \mathbf{\Sigma}_S^\phi \mathbf{P}_S\right)^\dagger\right) + \operatorname{tr}\left(\mathbf{\Sigma}^\phi \mathbf{P}_S^\perp\right) \|\boldsymbol{\theta}_*\|_2^2$$
$$+ \frac{1}{\alpha} \cdot \frac{\sigma^2}{n} \operatorname{tr}\left(\mathbf{\Sigma}^\phi \mathbf{P}_S^\perp\right) + \alpha \cdot \frac{\|\boldsymbol{\theta}_*\|_2^2}{2} \operatorname{tr}\left(\mathbf{\Sigma}^\phi \left(\mathbf{P}_S \mathbf{\Sigma}_S^\phi \mathbf{P}_S\right)^\dagger\right).$$

By taking $\alpha_* = \sqrt{\frac{\sigma^2}{n} \operatorname{tr}\left(\boldsymbol{\Sigma}^\phi \mathbf{P}_{\bar{\mathcal{S}}}^\perp\right) \Big/ \left(\frac{\|\boldsymbol{\theta}_*\|_2^2}{2} \operatorname{tr}\left(\boldsymbol{\Sigma}^\phi \left(\mathbf{P}_{\mathcal{S}} \boldsymbol{\Sigma}_{\mathcal{S}}^\phi \mathbf{P}_{\mathcal{S}}\right)^\dagger\right)\right)}$, we have

$$\frac{1}{\alpha_*} \cdot \frac{\sigma^2}{n} \operatorname{tr}\left(\boldsymbol{\Sigma}^\phi \mathbf{P}_{\bar{\mathcal{S}}}^\perp\right) + \alpha_* \cdot \frac{\|\boldsymbol{\theta}_*\|_2^2}{2} \operatorname{tr}\left(\boldsymbol{\Sigma}^\phi \left(\mathbf{P}_{\mathcal{S}} \boldsymbol{\Sigma}_{\mathcal{S}}^\phi \mathbf{P}_{\mathcal{S}}\right)^\dagger\right)$$

$$\leq 2\sqrt{\frac{\sigma^2}{n} \operatorname{tr}\left(\boldsymbol{\Sigma}^\phi \mathbf{P}_{\bar{\mathcal{S}}}^\perp\right) \cdot \frac{\|\boldsymbol{\theta}_*\|_2^2}{2} \operatorname{tr}\left(\boldsymbol{\Sigma}^\phi \left(\mathbf{P}_{\mathcal{S}} \boldsymbol{\Sigma}_{\mathcal{S}}^\phi \mathbf{P}_{\mathcal{S}}\right)^\dagger\right)}$$

$$\leq \frac{1}{\sqrt{2}} \left(\frac{\sigma^2}{n} \operatorname{tr}\left(\boldsymbol{\Sigma}^\phi \left(\mathbf{P}_{\mathcal{S}} \boldsymbol{\Sigma}_{\mathcal{S}}^\phi \mathbf{P}_{\mathcal{S}}\right)^\dagger\right) + \operatorname{tr}\left(\boldsymbol{\Sigma}^\phi \mathbf{P}_{\bar{\mathcal{S}}}^\perp\right) \|\boldsymbol{\theta}_*\|_2^2\right).$$

Therefore overall, we have

$$\mathbb{E}\left[\operatorname{ER}\left(\boldsymbol{\theta}_S\right)\right] \leq \frac{2\sigma^2}{n} \operatorname{tr}\left(\boldsymbol{\Sigma}^\phi \left(\mathbf{P}_{\mathcal{S}} \boldsymbol{\Sigma}_{\mathcal{S}}^\phi \mathbf{P}_{\mathcal{S}}\right)^\dagger\right) + 2 \operatorname{tr}\left(\boldsymbol{\Sigma}^\phi \mathbf{P}_{\bar{\mathcal{S}}}^\perp\right) \|\boldsymbol{\theta}_*\|_2^2.$$

$\square$

*Proof of Corollary 2.3.* Given $\mathbf{P}_{\mathcal{S}}(c_S \boldsymbol{\Sigma}_{\mathcal{S}}^\phi - \boldsymbol{\Sigma}^\phi)\mathbf{P}_{\mathcal{S}} \succeq 0$ and $\operatorname{rank}(\mathbf{P}_{\mathcal{S}}) \asymp \bar{r}$, the variance term is asymptotically upper bounded by

$$\text{variance} = \frac{2\sigma^2}{n} \operatorname{tr}\left(\boldsymbol{\Sigma}^\phi \left(\mathbf{P}_{\mathcal{S}} \boldsymbol{\Sigma}_{\mathcal{S}}^\phi \mathbf{P}_{\mathcal{S}}\right)^\dagger\right) \lesssim \frac{\sigma^2}{n} \cdot c_S \bar{r}.$$

Meanwhile, given $\operatorname{tr}(\boldsymbol{\Sigma}^\phi \mathbf{P}_{\bar{\mathcal{S}}}^\perp) \leq \frac{N}{n} \operatorname{tr}(\boldsymbol{\Sigma}^\phi - \langle \boldsymbol{\Sigma}^\phi \rangle_{\bar{r}})$ and $\operatorname{tr}(\boldsymbol{\Sigma}^\phi - \langle \boldsymbol{\Sigma}^\phi \rangle_{\bar{r}}) \leq \operatorname{tr}\left(\boldsymbol{\Sigma}^\phi\right)/N$, the bias term can be asymptotically upper bounded by

$$\text{bias} = 2 \operatorname{tr}\left(\boldsymbol{\Sigma}^\phi \mathbf{P}_{\bar{\mathcal{S}}}^\perp\right) \|\boldsymbol{\theta}_*\|_2^2 \leq \frac{2}{n} \operatorname{tr}\left(\boldsymbol{\Sigma}^\phi\right) \|\boldsymbol{\theta}_*\|_2^2.$$

The result follows from Theorem 2.2 by combining the variance and bias terms. $\square$

## C Proofs for Section 3.1

### C.1 Formal Statement and Proof of Theorem 3.1

**Theorem C.1** (Formal version of Theorem 3.1). *Under Assumption 2.2 and 2.3 with a small intrinsic dimension $\bar{r} \ll \min\{N, r\}$, for any $\delta \in (0,1)$, draw a Gaussian random matrix $\boldsymbol{\Gamma} \in \mathbb{R}^{r \times m}$ with i.i.d. entries from $\mathcal{N}(0, 1/m)$ where $m \asymp k/\delta$ for some $k \geq 1.1\bar{r}$. Let $\widetilde{\boldsymbol{\Sigma}}^\phi := \boldsymbol{\Gamma}^\top \boldsymbol{\Sigma}^\phi \boldsymbol{\Gamma}$ and $\widetilde{\boldsymbol{\Sigma}}_S^\phi := \boldsymbol{\Gamma}^\top \boldsymbol{\Sigma}_S^\phi \boldsymbol{\Gamma}$ be the sketched gradient moments. For any $S \subseteq [N]$ with $n > m$ samples such that (i) $\operatorname{rank}(\boldsymbol{\Sigma}_S^\phi) = n$, and (ii) the $k$-th largest eigenvalue $s_k(\widetilde{\boldsymbol{\Sigma}}_S^\phi) \geq \gamma_S$ for some $\gamma_S > 0$, with probability at least $1 - \delta$ over $\boldsymbol{\Gamma}$, there exists $\alpha > 0$ where (2) satisfies*

$$
\begin{aligned}
\mathbb{E}\left[\operatorname{ER}\left(\boldsymbol{\theta}_S\right)\right] \lesssim\ & \frac{\sigma^2}{n} \operatorname{tr}\left(\widetilde{\boldsymbol{\Sigma}}^\phi \langle \widetilde{\boldsymbol{\Sigma}}_S^\phi \rangle_k^\dagger\right) && (\textbf{\textit{variance}}) \\
&+ \frac{\sigma^2}{n} \frac{1}{m\gamma_S} \left\| \widetilde{\boldsymbol{\Sigma}}^\phi \langle \widetilde{\boldsymbol{\Sigma}}_S^\phi \rangle_k^\dagger \right\|_2 \operatorname{tr}\left(\boldsymbol{\Sigma}^\phi\right) && (\textbf{\textit{sketching error}}) \quad (7) \\
&+ \frac{1}{n} \left\| \widetilde{\boldsymbol{\Sigma}}^\phi \langle \widetilde{\boldsymbol{\Sigma}}_S^\phi \rangle_k^\dagger \right\|_2 \operatorname{tr}\left(\boldsymbol{\Sigma}^\phi\right) \|\boldsymbol{\theta}_*\|_2^2 && (\textbf{\textit{bias}}).
\end{aligned}
$$

*If $S$ further satisfies $\widetilde{\boldsymbol{\Sigma}}^\phi \preceq c_S \widetilde{\boldsymbol{\Sigma}}_S^\phi$ for some $c_S \geq \frac{n}{N}$, taking $m = \max\{\sqrt{\operatorname{tr}\left(\boldsymbol{\Sigma}^\phi\right)/\gamma_S}, 1.1\bar{r}/\delta\}$ leads to*

$$\mathbb{E}\left[\operatorname{ER}\left(\boldsymbol{\theta}_S\right)\right] \lesssim \textbf{\textit{variance}} + \textbf{\textit{sketching error}} + \textbf{\textit{bias}} \lesssim \frac{c_S}{n}\left(\sigma^2 m + \operatorname{tr}\left(\boldsymbol{\Sigma}^\phi\right) \|\boldsymbol{\theta}_*\|_2^2\right). \quad (8)$$

We start by introducing some helpful notations for the proofs. Let $\mathbf{G} := \nabla_{\boldsymbol{\theta}} f^\phi\left(\mathbf{X}; \mathbf{0}_r\right) \in \mathbb{R}^{N \times r}$ and $\mathbf{G}_S = [\mathbf{G}]_S \in \mathbb{R}^{n \times r}$ be the original gradients of $\mathcal{D}$ and $\mathcal{D}_S$, respectively. Recall that $\boldsymbol{\Sigma}^\phi = \mathbf{G}^\top \mathbf{G}/N$ and $\boldsymbol{\Sigma}_S^\phi = \mathbf{G}_S^\top \mathbf{G}_S/n$ are the corresponding second moments.

We consider a Johnson-Lindenstrauss transform (JLT) [31] $\boldsymbol{\Gamma} \in \mathbb{R}^{r \times m}$ as follows:

**Definition C.1** (JLT [18] (adapting [17, Definition 3])). *For any $\epsilon > 0$, $\delta \in (0, 1)$, and $n \in \mathbb{N}$, a random matrix $\mathbf{\Gamma} \in \mathbb{R}^{r \times m}$ is a $(\epsilon, \delta, k)$-Johnson-Lindenstrauss transform ($(\epsilon, \delta, k)$-JLT) if for any $\mathbf{U} \in \mathbb{R}^{r \times k}$ consisting of $k$ orthonormal columns in $\mathbb{R}^r$, with probability at least $1 - \delta$,*

$$\left\| \mathbf{I}_k - \mathbf{U}^\top \mathbf{\Gamma} \mathbf{\Gamma}^\top \mathbf{U} \right\|_2 \le \epsilon.$$

**Definition C.2** (JL second moment property [87] (adapting [17, Definition 12])). *For any $\epsilon > 0$, $\delta \in (0, 1)$, a random matrix $\mathbf{\Gamma} \in \mathbb{R}^{r \times m}$ satisfies the $(\epsilon, \delta)$-JL second moment property if*

$$\mathbb{E}\left[ \left( \left\| \mathbf{\Gamma}^\top \mathbf{u} \right\|_2^2 - 1 \right)^2 \right] \le \epsilon^2 \delta \quad \forall \, \mathbf{u} \in \mathbb{S}^{r-1}.$$

**Lemma C.2** (Approximated matrix-matrix multiplication [87] (adapting [17, Theorem 13])). *Given $\epsilon > 0$, $\delta \in (0, 1/2)$, and a random matrix $\mathbf{\Gamma} \in \mathbb{R}^{r \times m}$ satisfying the $(\epsilon, \delta)$-JL second moment property (Definition C.2), for any matrices $\mathbf{A}, \mathbf{B}$ each with $r$ rows,*

$$\Pr\left[ \left\| \mathbf{A}^\top \mathbf{\Gamma} \mathbf{\Gamma}^\top \mathbf{B} - \mathbf{A}^\top \mathbf{B} \right\|_F > 3\epsilon \left\| \mathbf{A} \right\|_F \left\| \mathbf{B} \right\|_F \right] \le \delta.$$

One of the most classical constructions of a JLT with JL second moment property is the Gaussian embedding:

**Lemma C.3** (Gaussian embedding [17, Theorem 6]). *For any $\epsilon > 0$, $\delta \in (0, 1)$, a Gaussian random matrix $\mathbf{\Gamma} \in \mathbb{R}^{r \times m}$ with i.i.d. entries $\mathbf{\Gamma}_{ij} \sim \mathcal{N}(0, 1/m)$ (i) is a $(\epsilon, \delta, k)$-JLT if $m \gtrsim (k + \log(1/\delta)) \epsilon^{-2}$; and (ii) satisfies the $(\epsilon, \delta)$-JL second moment property if $m \gtrsim \epsilon^{-2} \delta^{-1}$.*

*Proof of Lemma C.3.* The $(\epsilon, \delta, k)$-JLT condition follows directly from [17, Theorem 6].

To show the $(\epsilon, \delta)$-JL second moment property, we observe that for any $\mathbf{u} \in \mathbb{S}^{r-1}$, $\left\| \mathbf{\Gamma}^\top \mathbf{u} \right\|_2^2 = \mathbf{u}^\top \mathbf{\Gamma} \mathbf{\Gamma}^\top \mathbf{u}$ is an average of $m$ independent $\chi^2$ random variables with mean 1 and variance 2, we have $\mathbb{E}\left[ \left\| \mathbf{\Gamma}^\top \mathbf{u} \right\|_2^2 \right] = 1$ and its variance is $\mathbb{E}\left[ \left( \left\| \mathbf{\Gamma}^\top \mathbf{u} \right\|_2^2 - 1 \right)^2 \right] = 2/m$. Therefore, $m \gtrsim \epsilon^{-2} \delta^{-1}$ leads to the $(\epsilon, \delta)$-JL second moment property. $\qquad\square$

**Remark C.1** ((Fast) Johnson-Lindenstrauss transforms). *While we mainly focus on the Gaussian embedding in the analysis for simplicity, there is a rich spectrum of JLTs with the JL second moment property [88, 89, 90, 91], some of which enjoy remarkably better efficiency than the Gaussian embedding without compromising accuracy empirically. We refer interested readers to [17, 32, 33] for in-depth reviews on different JLTs and their applications, while briefly synopsizing two common choices and their efficiency as follows.*

(a) *Subgaussian embedding [88] is a random matrix $\mathbf{\Gamma} \in \mathbb{R}^{r \times m}$ with i.i.d. entries from a zero-mean subgaussian distribution with variance $1/m$. Common choices include the Rademacher distribution and Gaussian distribution (i.e., Gaussian embedding).*

   *Applying subgaussian embeddings to an $N \times r$ matrix $\mathbf{A}$ with $\mathrm{nnz}(\mathbf{A}) \le Nr$ nonzero entries takes $O(\mathrm{nnz}(\mathbf{A})m) \le O(Nrm)$ time, while the involved matrix-matrix multiplication can be computed distributedly in parallel leveraging the efficiency of Level 3 BLAS [92]. In practice, generating and applying Rademacher random matrices tend to be slightly faster than Gaussian embeddings due to the simple discrete support.*

(b) *Sparse sign matrix [89, 93] is a sparse random matrix $\mathbf{\Gamma} = \sqrt{\frac{r}{\xi}} \left[ \boldsymbol{\gamma}_1, \cdots, \boldsymbol{\gamma}_r \right]^\top \in \mathbb{R}^{r \times m}$ ($\xi \in \mathbb{N}$) with i.i.d. rows $\boldsymbol{\gamma}_j \in \mathbb{R}^m$ each consisting of $\xi$ non-zero entries at uniformly random coordinates filled with Rademacher random variables. When $\xi = 1$, $\mathbf{\Gamma}$ is known as CountSketch [94] and requires as many as $m = O(k^2)$ columns to satisfy the JLT property with constant distortion. Increasing the sparsity slightly, [95] showed that $m = O(k \log k)$ is sufficient for constant-distortion JLT when $\xi = O(\log k)$. In practice, [96] suggested that a small constant sparsity $\xi \ge 8$ is usually enough for many applications like low-rank approximations.*

   *The sparse sign matrix can be applied to an $N \times r$ matrix $\mathbf{A}$ with $\mathrm{nnz}(\mathbf{A})$ nonzero entries in $O(\mathrm{nnz}(\mathbf{A})\xi) \le O(Nr\xi)$ time, independent of the sketching size $m$. With careful implementation, sketching via sparse sign matrices can be significantly faster than the subgaussian embeddings in practice [33, 43].*

Let $\widetilde{\mathbf{G}} := \mathbf{G}\boldsymbol{\Gamma} \in \mathbb{R}^{N \times m}$ and $\widetilde{\mathbf{G}}_S = \mathbf{G}_S\boldsymbol{\Gamma} \in \mathbb{R}^{n \times m}$ be the sketched gradients such that

$$\widetilde{\boldsymbol{\Sigma}}^\phi := \boldsymbol{\Gamma}^\top \boldsymbol{\Sigma}^\phi \boldsymbol{\Gamma} = \widetilde{\mathbf{G}}^\top \widetilde{\mathbf{G}}/N \in \mathbb{R}^{m \times m}, \quad \widetilde{\boldsymbol{\Sigma}}_S^\phi := \boldsymbol{\Gamma}^\top \boldsymbol{\Sigma}_S^\phi \boldsymbol{\Gamma} = \widetilde{\mathbf{G}}_S^\top \widetilde{\mathbf{G}}_S/n \in \mathbb{R}^{m \times m}.$$

In particular, for a Gaussian embedding $\boldsymbol{\Gamma}$, when $\mathrm{rank}(\boldsymbol{\Sigma}_S^\phi) = \mathrm{rank}(\mathbf{G}_S) = n$, $\mathrm{rank}(\widetilde{\boldsymbol{\Sigma}}_S^\phi) = \mathrm{rank}(\widetilde{\mathbf{G}}_S) = m$ almost surely.

Recall the low intrinsic dimension $\bar{r}$ from Assumption 2.3. For any $k \in \mathbb{N}$ with $1.1\bar{r} \le k < m$, let $\mathbf{P}_S \in \mathbb{R}^{r \times r}$ be an orthogonal projector onto a dimension-$k$ subspace $\mathcal{S} \subseteq \mathrm{Range}(\boldsymbol{\Sigma}_S^\phi)$:

$$\mathbf{P}_S := (\langle \widetilde{\mathbf{G}}_S \rangle_k^\dagger \mathbf{G}_S)^\dagger (\langle \widetilde{\mathbf{G}}_S \rangle_k^\dagger \mathbf{G}_S) = \mathbf{G}_S^\dagger \langle \widetilde{\mathbf{G}}_S \rangle_k \langle \widetilde{\mathbf{G}}_S \rangle_k^\dagger \mathbf{G}_S, \tag{9}$$

and $\mathbf{P}_S^\perp = \mathbf{I}_r - \mathbf{P}_S$ be its orthogonal complement. Throughout the proof of Theorem C.1, we assume the following:

**Assumption C.1.** *Let* $\min\{N, r\} \gg n > m > k \ge 1.1\bar{r}$ *such that* $\mathrm{rank}(\boldsymbol{\Sigma}_S^\phi) = n$. *We consider a Gaussian embedding (Lemma C.3)* $\boldsymbol{\Gamma} \in \mathbb{R}^{r \times m}$ *with* $m \asymp k$ *such that* $s_k(\widetilde{\boldsymbol{\Sigma}}_S^\phi) \ge \gamma_S$ *for some* $\gamma_S > 0$.

***Proof of Theorems 3.1 and C.1.*** We first recall from Theorem 2.2 that

$$\mathbb{E}\left[\mathrm{ER}\left(\boldsymbol{\theta}_S\right)\right] \le \frac{2\sigma^2}{n} \mathrm{tr}\left(\boldsymbol{\Sigma}^\phi \left(\mathbf{P}_S \boldsymbol{\Sigma}_S^\phi \mathbf{P}_S\right)^\dagger\right) + 2\,\mathrm{tr}\left(\boldsymbol{\Sigma}^\phi \mathbf{P}_S^\perp\right) \|\boldsymbol{\theta}_*\|_2^2.$$

Lemma C.4 suggests that for $m \asymp k/\delta$, with probability at least $1 - \delta/2$,

$$\mathrm{tr}\left(\boldsymbol{\Sigma}^\phi \left(\mathbf{P}_S \boldsymbol{\Sigma}_S^\phi \mathbf{P}_S\right)^\dagger\right) \lesssim \mathrm{tr}\left(\widetilde{\boldsymbol{\Sigma}}^\phi \langle \widetilde{\boldsymbol{\Sigma}}_S^\phi \rangle_k^\dagger\right) + \frac{n}{m\gamma_S} \mathrm{tr}\left(\boldsymbol{\Sigma}^\phi \mathbf{P}_S^\perp\right).$$

Therefore,

$$\mathbb{E}\left[\mathrm{ER}\left(\boldsymbol{\theta}_S\right)\right] \lesssim \frac{\sigma^2}{n} \mathrm{tr}\left(\widetilde{\boldsymbol{\Sigma}}^\phi \langle \widetilde{\boldsymbol{\Sigma}}_S^\phi \rangle_k^\dagger\right) + \left(\frac{\sigma^2}{m\gamma_S} + \|\boldsymbol{\theta}_*\|_2^2\right) \mathrm{tr}\left(\boldsymbol{\Sigma}^\phi \mathbf{P}_S^\perp\right).$$

Then, applying Lemma C.7 with the union bound, we have

$$\mathrm{tr}\left(\boldsymbol{\Sigma}^\phi \mathbf{P}_S^\perp\right) \lesssim \frac{1}{n} \left\|\widetilde{\boldsymbol{\Sigma}}^\phi \langle \widetilde{\boldsymbol{\Sigma}}_S^\phi \rangle_k^\dagger\right\|_2 \mathrm{tr}\left(\boldsymbol{\Sigma}^\phi\right)$$

with probability at least $1 - \delta$. This implies

$$\mathbb{E}\left[\mathrm{ER}\left(\boldsymbol{\theta}_S\right)\right] \lesssim \frac{\sigma^2}{n} \left(\mathrm{tr}\left(\widetilde{\boldsymbol{\Sigma}}^\phi \langle \widetilde{\boldsymbol{\Sigma}}_S^\phi \rangle_k^\dagger\right) + \frac{1}{m\gamma_S} \left\|\widetilde{\boldsymbol{\Sigma}}^\phi \langle \widetilde{\boldsymbol{\Sigma}}_S^\phi \rangle_k^\dagger\right\|_2 \mathrm{tr}\left(\boldsymbol{\Sigma}^\phi\right)\right)$$
$$+ \frac{1}{n} \left\|\widetilde{\boldsymbol{\Sigma}}^\phi \langle \widetilde{\boldsymbol{\Sigma}}_S^\phi \rangle_k^\dagger\right\|_2 \mathrm{tr}\left(\boldsymbol{\Sigma}^\phi\right) \|\boldsymbol{\theta}_*\|_2^2.$$

If $S$ further satisfies $\widetilde{\boldsymbol{\Sigma}}^\phi \preccurlyeq c_S \widetilde{\boldsymbol{\Sigma}}_S^\phi$ for some $c_S \ge \frac{n}{N}$, then we have

$$\mathrm{tr}\left(\widetilde{\boldsymbol{\Sigma}}^\phi \langle \widetilde{\boldsymbol{\Sigma}}_S^\phi \rangle_k^\dagger\right) \le \mathrm{tr}\left(\widetilde{\boldsymbol{\Sigma}}^\phi (\widetilde{\boldsymbol{\Sigma}}_S^\phi)^\dagger\right) \le c_S m, \quad \left\|\widetilde{\boldsymbol{\Sigma}}^\phi \langle \widetilde{\boldsymbol{\Sigma}}_S^\phi \rangle_k^\dagger\right\|_2 \le \left\|\widetilde{\boldsymbol{\Sigma}}^\phi (\widetilde{\boldsymbol{\Sigma}}_S^\phi)^\dagger\right\|_2 \le c_S.$$

Therefore, (7) can be further simplified as

$$\mathbb{E}\left[\mathrm{ER}\left(\boldsymbol{\theta}_S\right)\right] \lesssim \frac{c_S \sigma^2}{n} \left(m + \frac{\mathrm{tr}\left(\boldsymbol{\Sigma}^\phi\right)}{m\gamma_S}\right) + \frac{c_S}{n} \mathrm{tr}\left(\boldsymbol{\Sigma}^\phi\right) \|\boldsymbol{\theta}_*\|_2^2.$$

On the right-hand-side, the first (variance) term is minimized at $m = \sqrt{\mathrm{tr}\left(\boldsymbol{\Sigma}^\phi\right)/\gamma_S}$ where $m + \mathrm{tr}\left(\boldsymbol{\Sigma}^\phi\right)/(m\gamma_S) \le 2\sqrt{\mathrm{tr}\left(\boldsymbol{\Sigma}^\phi\right)/\gamma_S} = 2m$. In addition, incorporting the assumption that $m \asymp k/\delta$ for some $k \ge 1.1\bar{r}$, we take $m = \max\{\sqrt{\mathrm{tr}\left(\boldsymbol{\Sigma}^\phi\right)/\gamma_S}, 1.1\bar{r}/\delta\}$ and get

$$\mathbb{E}\left[\mathrm{ER}\left(\boldsymbol{\theta}_S\right)\right] \lesssim \frac{c_S \sigma^2}{n} m + \frac{c_S}{n} \mathrm{tr}\left(\boldsymbol{\Sigma}^\phi\right) \|\boldsymbol{\theta}_*\|_2^2 = \frac{c_S}{n} \left(\sigma^2 m + \mathrm{tr}\left(\boldsymbol{\Sigma}^\phi\right) \|\boldsymbol{\theta}_*\|_2^2\right).$$

Theorem 3.1 is simplified from Theorem C.1 by taking $k = \lceil 1.1\bar{r} \rceil$ and $\delta = 0.1$. $\qquad\square$

## C.2 Upper Bounding Variance

**Lemma C.4.** *For any $\delta \in (0,1)$, let $\boldsymbol{\Gamma} \in \mathbb{R}^{r \times m}$ be a Gaussian embedding (Lemma C.3) with $m \asymp k/\delta$ columns. Then, with probability at least $1 - \delta$ over $\boldsymbol{\Gamma}$,*

$$\mathrm{tr}\left(\boldsymbol{\Sigma}^\phi \left(\mathbf{P}_\mathcal{S} \boldsymbol{\Sigma}_S^\phi \mathbf{P}_\mathcal{S}\right)^\dagger\right) \lesssim \mathrm{tr}\left(\widetilde{\boldsymbol{\Sigma}}^\phi \langle \widetilde{\boldsymbol{\Sigma}}_S^\phi \rangle_k^\dagger\right) + \frac{n}{m\gamma_S} \mathrm{tr}\left(\boldsymbol{\Sigma}^\phi \mathbf{P}_{\bar{\mathcal{S}}}^\perp\right).$$

*Proof of Lemma C.4.* We first observe that since $\mathrm{rank}(\boldsymbol{\Sigma}_S^\phi) = n$ implies $\mathbf{G}_S \mathbf{G}_S^\dagger = \mathbf{I}_n$,

$$\mathbf{G}_S \mathbf{P}_\mathcal{S} \boldsymbol{\Gamma} = \mathbf{G}_S \mathbf{G}_S^\dagger \langle \widetilde{\mathbf{G}}_S \rangle_k \langle \widetilde{\mathbf{G}}_S \rangle_k^\dagger \mathbf{G}_S \boldsymbol{\Gamma} = \langle \widetilde{\mathbf{G}}_S \rangle_k \langle \widetilde{\mathbf{G}}_S \rangle_k^\dagger \widetilde{\mathbf{G}}_S = \langle \widetilde{\mathbf{G}}_S \rangle_k,$$

and therefore, $\mathbf{G}(\mathbf{G}_S \mathbf{P}_\mathcal{S})^\dagger = \mathrm{argmin}_{\mathbf{Z} \in \mathbb{R}^{N \times n}} \|\mathbf{Z}\|_F^2$ *s.t.* $\mathbf{Z} \in \mathrm{argmin}_\mathbf{Z} \|\mathbf{G} - \mathbf{Z}\mathbf{G}_S \mathbf{P}_\mathcal{S}\|_F^2$ and

$$\widetilde{\mathbf{G}} \langle \widetilde{\mathbf{G}}_S \rangle_k^\dagger = \mathrm{argmin}_{\mathbf{Z} \in \mathbb{R}^{N \times n}} \|\mathbf{Z}\|_F^2 \ \text{s.t.} \ \mathbf{Z} \in \mathrm{argmin}_\mathbf{Z} \|(\mathbf{G} - \mathbf{Z}\mathbf{G}_S \mathbf{P}_\mathcal{S})\boldsymbol{\Gamma}\|_F^2$$

is an approximated solution from a sketched least square problem.

**Accuracy of sketched least square residual.** For $m \asymp k/(\epsilon^2 \delta)$, Lemma C.3 implies that a Gaussian embedding $\boldsymbol{\Gamma}$ is a $(1/2, \delta/2, k)$-JLT (Definition C.1) with $(\Theta(\epsilon/\sqrt{k}), \delta/2)$-JL second moment property (Definition C.2). Then, since $\mathrm{rank}(\mathbf{G}_S \mathbf{P}_\mathcal{S}) = k$, by Lemma C.5, with probability at least $1 - \delta$ over $\boldsymbol{\Gamma}$,

$$\left\|\left(\mathbf{G}(\mathbf{G}_S \mathbf{P}_\mathcal{S})^\dagger - \widetilde{\mathbf{G}} \langle \widetilde{\mathbf{G}}_S \rangle_k^\dagger\right) \mathbf{G}_S \mathbf{P}_\mathcal{S}\right\|_F^2 \le \epsilon^2 \left\|\mathbf{G} - \mathbf{G}(\mathbf{G}_S \mathbf{P}_\mathcal{S})^\dagger (\mathbf{G}_S \mathbf{P}_\mathcal{S})\right\|_2^2.$$

Since $\mathbf{G}_S \mathbf{P}_\mathcal{S} = \mathbf{G}_S \mathbf{G}_S^\dagger \langle \widetilde{\mathbf{G}}_S \rangle_k \langle \widetilde{\mathbf{G}}_S \rangle_k^\dagger \mathbf{G}_S = \langle \widetilde{\mathbf{G}}_S \rangle_k \langle \widetilde{\mathbf{G}}_S \rangle_k^\dagger \mathbf{G}_S$,

$$(\mathbf{G}_S \mathbf{P}_\mathcal{S})^\dagger (\mathbf{G}_S \mathbf{P}_\mathcal{S}) = \mathbf{G}_S^\dagger \langle \widetilde{\mathbf{G}}_S \rangle_k \langle \widetilde{\mathbf{G}}_S \rangle_k^\dagger \mathbf{G}_S = \mathbf{P}_\mathcal{S}.$$

Therefore,

$$\left\|\left(\mathbf{G}(\mathbf{G}_S \mathbf{P}_\mathcal{S})^\dagger - \widetilde{\mathbf{G}} \langle \widetilde{\mathbf{G}}_S \rangle_k^\dagger\right) \mathbf{G}_S \mathbf{P}_\mathcal{S}\right\|_F^2 \le \epsilon^2 \left\|\mathbf{G}\mathbf{P}_{\bar{\mathcal{S}}}^\perp\right\|_F^2. \tag{10}$$

**Accuracy of sketched least square solution.** To upper bound $\left\|\mathbf{G}(\mathbf{G}_S \mathbf{P}_\mathcal{S})^\dagger - \widetilde{\mathbf{G}} \langle \widetilde{\mathbf{G}}_S \rangle_k^\dagger\right\|_F$, we first observe from (10) that

$$\left\|\mathbf{G}(\mathbf{G}_S \mathbf{P}_\mathcal{S})^\dagger - \widetilde{\mathbf{G}} \langle \widetilde{\mathbf{G}}_S \rangle_k^\dagger\right\|_F^2 \le \epsilon^2 \left\|(\mathbf{G}_S \mathbf{P}_\mathcal{S})^\dagger\right\|_2^2 \left\|\mathbf{G}\mathbf{P}_{\bar{\mathcal{S}}}^\perp\right\|_F^2.$$

$\mathbf{G}_S \mathbf{P}_\mathcal{S} \mathbf{G}_S^\top = \langle \widetilde{\mathbf{G}}_S \rangle_k \langle \widetilde{\mathbf{G}}_S \rangle_k^\dagger \mathbf{G}_S \mathbf{G}_S^\top$. Since $\mathrm{rank}(\mathbf{G}_S) = n$, Lemma C.6 implies that for a Gaussian embedding $\boldsymbol{\Gamma}$, $\mathbf{G}_S \mathbf{G}_S^\top \succcurlyeq O\left(\frac{m}{n}\right) \mathbf{G}_S \boldsymbol{\Gamma} \boldsymbol{\Gamma}^\top \mathbf{G}_S^\top$ with high probability. Therefore,

$$\mathbf{G}_S \mathbf{P}_\mathcal{S} \mathbf{G}_S^\top \succcurlyeq O\left(\frac{m}{n}\right) \langle \widetilde{\mathbf{G}}_S \rangle_k \langle \widetilde{\mathbf{G}}_S \rangle_k^\dagger \widetilde{\mathbf{G}}_S \widetilde{\mathbf{G}}_S^\top = O\left(\frac{m}{n}\right) \langle \widetilde{\mathbf{G}}_S \rangle_k \langle \widetilde{\mathbf{G}}_S \rangle_k^\top.$$

Recall that $\langle \widetilde{\boldsymbol{\Sigma}}_S^\phi \rangle_k = \frac{1}{n} \langle \widetilde{\mathbf{G}}_S \rangle_k^\top \langle \widetilde{\mathbf{G}}_S \rangle_k$ and $s_k(\widetilde{\boldsymbol{\Sigma}}_S^\phi) \ge \gamma_S$, we have

$$\left\|(\mathbf{G}_S \mathbf{P}_\mathcal{S})^\dagger\right\|_2^2 = \left\|(\mathbf{G}_S \mathbf{P}_\mathcal{S} \mathbf{G}_S^\top)^\dagger\right\|_2 \le O\left(\frac{n}{m}\right) \left\|\left(\langle \widetilde{\mathbf{G}}_S \rangle_k^\top \langle \widetilde{\mathbf{G}}_S \rangle_k\right)^\dagger\right\|_2$$

$$\le O\left(\frac{n}{m}\right) \frac{1}{n\gamma_S} = O\left(\frac{1}{m\gamma_S}\right).$$

Therefore, applying a union bound gives that with probability at least $1 - \delta$ over $\boldsymbol{\Gamma}$,

$$\left\|\mathbf{G}(\mathbf{G}_S \mathbf{P}_\mathcal{S})^\dagger - \widetilde{\mathbf{G}} \langle \widetilde{\mathbf{G}}_S \rangle_k^\dagger\right\|_F^2 \le O\left(\frac{\epsilon^2}{m\gamma_S}\right) \left\|\mathbf{G}\mathbf{P}_{\bar{\mathcal{S}}}^\perp\right\|_F^2. \tag{11}$$

To upper bound $\left\|\mathbf{G}(\mathbf{G}_S \mathbf{P}_\mathcal{S})^\dagger\right\|_F^2$, we observe that by (11),

$$\left\|\mathbf{G}(\mathbf{G}_S \mathbf{P}_\mathcal{S})^\dagger\right\|_F^2 \le 2\left\|\widetilde{\mathbf{G}} \langle \widetilde{\mathbf{G}}_S \rangle_k^\dagger\right\|_F^2 + 2\left\|\mathbf{G}(\mathbf{G}_S \mathbf{P}_\mathcal{S})^\dagger - \widetilde{\mathbf{G}} \langle \widetilde{\mathbf{G}}_S \rangle_k^\dagger\right\|_F^2$$

$$\lesssim \left\|\widetilde{\mathbf{G}} \langle \widetilde{\mathbf{G}}_S \rangle_k^\dagger\right\|_F^2 + \frac{\epsilon^2}{m\gamma_S} \left\|\mathbf{G}\mathbf{P}_{\bar{\mathcal{S}}}^\perp\right\|_F^2.$$

Finally, normalizing by multiplying $n/N$ on both sides gives

$$\text{tr}\left(\boldsymbol{\Sigma}^{\phi}\left(\mathbf{P}_{\mathcal{S}}\boldsymbol{\Sigma}_{S}^{\phi}\mathbf{P}_{\mathcal{S}}\right)^{\dagger}\right) \lesssim \text{tr}\left(\widetilde{\boldsymbol{\Sigma}}^{\phi}\langle\widetilde{\boldsymbol{\Sigma}}_{S}^{\phi}\rangle_{k}^{\dagger}\right) + \epsilon^{2}\frac{n}{m\gamma_{S}}\text{tr}\left(\boldsymbol{\Sigma}^{\phi}\mathbf{P}_{\mathcal{S}}^{\perp}\right).$$

Taking any small constant $\epsilon > 0$ completes the proof. $\qquad\square$

**Lemma C.5** (Adapting [17, Theorem 23]). *For any $\epsilon > 0$ and $\delta \in (0,1)$, let $\boldsymbol{\Gamma} \in \mathbb{R}^{r\times m}$ b a $(1/2, \delta/2, k)$-JLT (Definition C.1) with $(\Theta(\epsilon/\sqrt{k}), \delta/2)$-JL second moment property (Definition C.2). Given $\mathbf{A} \in \mathbb{R}^{r\times n}$ with $\text{rank}(\mathbf{A}) = k$ and $\mathbf{B} \in \mathbb{R}^{r\times N}$, let*

$$\widehat{\mathbf{W}} = \underset{\mathbf{W}\in\mathbb{R}^{n\times N}}{\text{argmin}} \|\mathbf{W}\|_{F}^{2} \text{ s.t. } \mathbf{W} \in \underset{\mathbf{W}}{\text{argmin}} \left\|\boldsymbol{\Gamma}^{\top}(\mathbf{A}\mathbf{W} - \mathbf{B})\right\|_{F}^{2},$$

$$\mathbf{W}_{*} = \underset{\mathbf{W}\in\mathbb{R}^{n\times N}}{\text{argmin}} \|\mathbf{W}\|_{F}^{2} \text{ s.t. } \mathbf{W} \in \underset{\mathbf{W}}{\text{argmin}} \|\mathbf{A}\mathbf{W} - \mathbf{B}\|_{F}^{2}.$$

*Then, with probability at least $1 - \delta$ over $\boldsymbol{\Gamma}$, $\left\|\mathbf{A}(\widehat{\mathbf{W}} - \mathbf{W}_{*})\right\|_{F} \leq \epsilon \|\mathbf{A}\mathbf{W}_{*} - \mathbf{B}\|_{F}$.*

*Proof of Lemma C.5.* Analogous to the proof of [17, Theorem 23], let $\mathbf{A} = \mathbf{Q}\mathbf{R}$ be a reduced QR decomposition of $\mathbf{A}$ such that $\mathbf{Q} \in \mathbb{R}^{r\times k}$ is an orthonormal basis for $\text{Range}(\mathbf{A})$, and $\mathbf{R} \in \mathbb{R}^{k\times n}$. Reparametrizing $\widehat{\mathbf{Z}} = \mathbf{R}\widehat{\mathbf{W}}$ and $\mathbf{Z}_{*} = \mathbf{R}\mathbf{W}_{*}$, up to constant scaling of $\epsilon$, it is sufficient to show

$$\left\|\mathbf{Q}(\widehat{\mathbf{Z}} - \mathbf{Z}_{*})\right\|_{F} = \left\|\widehat{\mathbf{Z}} - \mathbf{Z}_{*}\right\|_{F} \leq O(\epsilon)\left\|\mathbf{Q}\mathbf{Z}_{*} - \mathbf{B}\right\|_{F}.$$

Since $\boldsymbol{\Gamma} \in \mathbb{R}^{r\times k}$ is an $(1/2, \delta/2, k)$-JLT, we have $\left\|\mathbf{I}_{k} - \mathbf{Q}^{\top}\boldsymbol{\Gamma}\boldsymbol{\Gamma}^{\top}\mathbf{Q}\right\|_{2} \leq 1/2$ with probability at least $1 - \delta/2$ and

$$\begin{aligned}
\left\|\widehat{\mathbf{Z}} - \mathbf{Z}_{*}\right\|_{F} &\leq \left\|\mathbf{Q}^{\top}\boldsymbol{\Gamma}\boldsymbol{\Gamma}^{\top}\mathbf{Q}(\widehat{\mathbf{Z}} - \mathbf{Z}_{*})\right\|_{F} + \left\|\mathbf{Q}^{\top}\boldsymbol{\Gamma}\boldsymbol{\Gamma}^{\top}\mathbf{Q}(\widehat{\mathbf{Z}} - \mathbf{Z}_{*}) - (\widehat{\mathbf{Z}} - \mathbf{Z}_{*})\right\|_{F} \\
&\leq \left\|\mathbf{Q}^{\top}\boldsymbol{\Gamma}\boldsymbol{\Gamma}^{\top}\mathbf{Q}(\widehat{\mathbf{Z}} - \mathbf{Z}_{*})\right\|_{F} + \left\|\mathbf{I}_{k} - \mathbf{Q}^{\top}\boldsymbol{\Gamma}\boldsymbol{\Gamma}^{\top}\mathbf{Q}\right\|_{2}\left\|\widehat{\mathbf{Z}} - \mathbf{Z}_{*}\right\|_{F} \\
&\leq \left\|\mathbf{Q}^{\top}\boldsymbol{\Gamma}\boldsymbol{\Gamma}^{\top}\mathbf{Q}(\widehat{\mathbf{Z}} - \mathbf{Z}_{*})\right\|_{F} + 1/2\left\|\widehat{\mathbf{Z}} - \mathbf{Z}_{*}\right\|_{F}
\end{aligned}$$

which implies $\left\|\widehat{\mathbf{Z}} - \mathbf{Z}_{*}\right\|_{F} \leq 2\left\|\mathbf{Q}^{\top}\boldsymbol{\Gamma}\boldsymbol{\Gamma}^{\top}\mathbf{Q}(\widehat{\mathbf{Z}} - \mathbf{Z}_{*})\right\|_{F}$ with probability at least $1 - \delta/2$.

By the normal equation of the sketched least square problem, $\mathbf{Q}^{\top}\boldsymbol{\Gamma}\boldsymbol{\Gamma}^{\top}\mathbf{Q}\widehat{\mathbf{Z}} = \mathbf{Q}^{\top}\boldsymbol{\Gamma}\boldsymbol{\Gamma}^{\top}\mathbf{B}$. Thus,

$$\left\|\widehat{\mathbf{Z}} - \mathbf{Z}_{*}\right\|_{F} \leq 2\left\|\mathbf{Q}^{\top}\boldsymbol{\Gamma}\boldsymbol{\Gamma}^{\top}(\mathbf{Q}\mathbf{Z}_{*} - \mathbf{B})\right\|_{F}.$$

Since $\mathbf{Q}^{\top}(\mathbf{Q}\mathbf{Z}_{*} - \mathbf{B}) = -\mathbf{Q}^{\top}\left(\mathbf{I}_{r} - \mathbf{Q}\mathbf{Q}^{\top}\right)\mathbf{B} = \mathbf{0}_{k\times N}$ and $\boldsymbol{\Gamma}$ has $(\epsilon/\sqrt{k}, \delta/2)$-JL second moment property, Lemma C.2 implies that with probability at least $1 - \delta/2$,

$$\left\|\mathbf{Q}^{\top}\boldsymbol{\Gamma}\boldsymbol{\Gamma}^{\top}(\mathbf{Q}\mathbf{Z}_{*} - \mathbf{B})\right\|_{F} \leq \Theta\left(\epsilon/\sqrt{k}\right)\|\mathbf{Q}\|_{F}\|\mathbf{Q}\mathbf{Z}_{*} - \mathbf{B}\|_{F} = \Theta(\epsilon)\|\mathbf{Q}\mathbf{Z}_{*} - \mathbf{B}\|_{F}.$$

Then, by the union bound, with probability at least $1 - \delta$ over $\boldsymbol{\Gamma}$, we have

$$\left\|\widehat{\mathbf{Z}} - \mathbf{Z}_{*}\right\|_{F} \leq 2\left\|\mathbf{Q}^{\top}\boldsymbol{\Gamma}\boldsymbol{\Gamma}^{\top}(\mathbf{Q}\mathbf{Z}_{*} - \mathbf{B})\right\|_{F} \leq \epsilon\|\mathbf{Q}\mathbf{Z}_{*} - \mathbf{B}\|_{F}$$

with a suitable choice of the constant for $\Theta(\epsilon)$. $\qquad\square$

**Lemma C.6** ([97]). *For a random matrix $\boldsymbol{\Omega} \in \mathbb{R}^{n\times m}$ ($n > m$) consisting of i.i.d. subgaussian entries with mean zero and variance one, with high probability,*

$$O\left(\left(\sqrt{n} - \sqrt{m}\right)^{2}\right)\mathbf{I}_{n} \preccurlyeq \boldsymbol{\Omega}\boldsymbol{\Omega}^{\top} \preccurlyeq O\left(\left(\sqrt{n} + \sqrt{m}\right)^{2}\right)\mathbf{I}_{n}.$$

## C.3 Upper Bounding Low-rank Approximation Error

**Lemma C.7.** *Under Assumption 2.3, let $\mathbf{\Gamma} \in \mathbb{R}^{r \times m}$ be a Gaussian embedding (Lemma C.3) such that there exists $m > k \geq 1.1\bar{r}$ satisfying $s_k(\widetilde{\mathbf{\Sigma}}_S^\phi) \geq \gamma_S$ for some $\gamma_S > 0$. Then, with probability at least $1 - \exp(-\Omega(\bar{r}))$,*

$$\mathrm{tr}\left(\mathbf{\Sigma}^\phi \mathbf{P}_S^\perp\right) \lesssim \frac{1}{n} \left\| \widetilde{\mathbf{\Sigma}}^\phi \langle \widetilde{\mathbf{\Sigma}}_S^\phi \rangle_k^\dagger \right\|_2 \mathrm{tr}\left(\mathbf{\Sigma}^\phi\right).$$

*Proof of Lemma C.7.* Here we follow a similar proof strategy as [43, Theorem 1]. Let $\mathbf{\Pi}_S := [\mathbf{I}_N]_S \in \mathbb{R}^{n \times N}$ be the selection matrix associated with $S \subseteq [N]$. We introduce the following $N \times N$ oblique projectors:

$$\mathbf{M}_S := \mathbf{G}\mathbf{G}_S^\dagger \langle \widetilde{\mathbf{G}}_S \rangle_k \langle \widetilde{\mathbf{G}}_S \rangle_k^\dagger \mathbf{\Pi}_S, \quad \widetilde{\mathbf{M}}_S := \widetilde{\mathbf{G}} \langle \widetilde{\mathbf{G}}_S \rangle_k^\dagger \mathbf{\Pi}_S.$$

In particular, $\mathbf{M}_S$ and $\widetilde{\mathbf{M}}_S$ are the oblique projectors since with $\mathbf{\Pi}_S \mathbf{G} = \mathbf{G}_S$ and $\mathbf{G}_S \mathbf{G}_S^\dagger = \mathbf{I}_n$,

$$\begin{aligned}\mathbf{M}_S^2 &= \mathbf{G}\mathbf{G}_S^\dagger \langle \widetilde{\mathbf{G}}_S \rangle_k \langle \widetilde{\mathbf{G}}_S \rangle_k^\dagger \mathbf{G}_S \mathbf{G}_S^\dagger \langle \widetilde{\mathbf{G}}_S \rangle_k \langle \widetilde{\mathbf{G}}_S \rangle_k^\dagger \mathbf{\Pi}_S \\ &= \mathbf{G}\mathbf{G}_S^\dagger \langle \widetilde{\mathbf{G}}_S \rangle_k \langle \widetilde{\mathbf{G}}_S \rangle_k^\dagger \mathbf{\Pi}_S = \mathbf{M}_S,\end{aligned}$$

and with $\mathbf{\Pi}_S \widetilde{\mathbf{G}} = \widetilde{\mathbf{G}}_S$,

$$\widetilde{\mathbf{M}}_S^2 = \widetilde{\mathbf{G}} \langle \widetilde{\mathbf{G}}_S \rangle_k^\dagger \widetilde{\mathbf{G}}_S \langle \widetilde{\mathbf{G}}_S \rangle_k^\dagger \mathbf{\Pi}_S = \widetilde{\mathbf{G}} \langle \widetilde{\mathbf{G}}_S \rangle_k^\dagger \mathbf{\Pi}_S = \widetilde{\mathbf{M}}_S.$$

Recalling $\mathbf{P}_S$ from (9), we observe the following identities:

$$\mathbf{M}_S \mathbf{G} = \mathbf{G}\mathbf{G}_S^\dagger \langle \widetilde{\mathbf{G}}_S \rangle_k \langle \widetilde{\mathbf{G}}_S \rangle_k^\dagger \mathbf{G}_S = \mathbf{G}\mathbf{P}_S; \tag{12}$$

since $\mathbf{G}_S \mathbf{G}_S^\dagger = \mathbf{I}_n$,

$$\widetilde{\mathbf{M}}_S \mathbf{M}_S = \widetilde{\mathbf{G}} \langle \widetilde{\mathbf{G}}_S \rangle_k^\dagger \mathbf{G}_S \mathbf{G}_S^\dagger \langle \widetilde{\mathbf{G}}_S \rangle_k \langle \widetilde{\mathbf{G}}_S \rangle_k^\dagger \mathbf{\Pi}_S = \widetilde{\mathbf{G}} \langle \widetilde{\mathbf{G}}_S \rangle_k^\dagger \mathbf{\Pi}_S = \widetilde{\mathbf{M}}_S; \tag{13}$$

and

$$\widetilde{\mathbf{M}}_S \widetilde{\mathbf{G}} \langle \widetilde{\mathbf{G}}_S \rangle_k^\dagger \langle \widetilde{\mathbf{G}}_S \rangle_k = \widetilde{\mathbf{G}} \langle \widetilde{\mathbf{G}}_S \rangle_k^\dagger \widetilde{\mathbf{G}}_S \langle \widetilde{\mathbf{G}}_S \rangle_k^\dagger \langle \widetilde{\mathbf{G}}_S \rangle_k = \widetilde{\mathbf{G}} \langle \widetilde{\mathbf{G}}_S \rangle_k^\dagger \langle \widetilde{\mathbf{G}}_S \rangle_k. \tag{14}$$

Combining (12), (13), and (14), we have

$$\begin{aligned}\mathbf{G}\mathbf{P}_S &= \mathbf{G} - \mathbf{G}\mathbf{P}_S = (\mathbf{I}_N - \mathbf{M}_S)\mathbf{G} \quad \text{(by (12))} \\ &= \left(\mathbf{I}_N - \widetilde{\mathbf{M}}_S\right)(\mathbf{I}_N - \mathbf{M}_S)\mathbf{G} \quad \text{(by (13))} \\ &= \left(\mathbf{I}_N - \widetilde{\mathbf{M}}_S\right)\mathbf{G}\mathbf{P}_S^\perp \quad \text{(by (12))} \\ &= \left(\mathbf{I}_N - \widetilde{\mathbf{M}}_S\right)\left(\mathbf{I}_N - \widetilde{\mathbf{G}} \langle \widetilde{\mathbf{G}}_S \rangle_k^\dagger \langle \widetilde{\mathbf{G}}_S \rangle_k \widetilde{\mathbf{G}}^\dagger\right)\mathbf{G}\mathbf{P}_S^\perp \quad \text{(by (14))}.\end{aligned}$$

Since $\left\|\mathbf{P}_S^\perp\right\|_2^2 = 1$, this implies

$$\begin{aligned}\mathrm{tr}\left(\mathbf{\Sigma}^\phi \mathbf{P}_S^\perp\right) &= \frac{1}{N}\|\mathbf{G}\mathbf{P}_S\|_F^2 = \frac{1}{N}\left\|\left(\mathbf{I}_N - \widetilde{\mathbf{M}}_S\right)\left(\mathbf{I}_N - \widetilde{\mathbf{G}} \langle \widetilde{\mathbf{G}}_S \rangle_k^\dagger \langle \widetilde{\mathbf{G}}_S \rangle_k \widetilde{\mathbf{G}}^\dagger\right)\mathbf{G}\mathbf{P}_S^\dagger\right\|_F^2 \\ &\leq \frac{1}{N}\left\|\mathbf{I}_N - \widetilde{\mathbf{M}}_S\right\|_2^2 \left\|\left(\mathbf{I}_N - \widetilde{\mathbf{G}} \langle \widetilde{\mathbf{G}}_S \rangle_k^\dagger \langle \widetilde{\mathbf{G}}_S \rangle_k \widetilde{\mathbf{G}}^\dagger\right)\mathbf{G}\right\|_F^2.\end{aligned}$$

By the operator norm identity for projectors [98], we have

$$\left\|\mathbf{I}_N - \widetilde{\mathbf{M}}_S\right\|_2^2 = \left\|\widetilde{\mathbf{M}}_S\right\|_2^2 = \left\|\widetilde{\mathbf{G}} \langle \widetilde{\mathbf{G}}_S \rangle_k^\dagger \mathbf{\Pi}_S\right\|_2^2 = \left\|\widetilde{\mathbf{G}} \langle \widetilde{\mathbf{G}}_S \rangle_k^\dagger\right\|_2^2 = \frac{N}{n}\left\|\widetilde{\mathbf{\Sigma}}^\phi \langle \widetilde{\mathbf{\Sigma}}_S^\phi \rangle_k^\dagger\right\|_2,$$

and therefore,

$$\mathrm{tr}\left(\mathbf{\Sigma}^\phi \mathbf{P}_S^\perp\right) \leq \frac{1}{n}\left\|\widetilde{\mathbf{\Sigma}}^\phi \langle \widetilde{\mathbf{\Sigma}}_S^\phi \rangle_k^\dagger\right\|_2 \left\|\left(\mathbf{I}_N - \widetilde{\mathbf{G}} \langle \widetilde{\mathbf{G}}_S \rangle_k^\dagger \langle \widetilde{\mathbf{G}}_S \rangle_k \widetilde{\mathbf{G}}^\dagger\right)\mathbf{G}\right\|_F^2.$$

Since $\widetilde{\mathbf{G}} \langle \widetilde{\mathbf{G}}_S \rangle_k^\dagger \langle \widetilde{\mathbf{G}}_S \rangle_k \widetilde{\mathbf{G}}^\dagger$ is a rank-$k$ orthogonal projector onto

$$\mathrm{Range}\left(\widetilde{\mathbf{G}} \langle \widetilde{\mathbf{G}}_S \rangle_k^\dagger\right) = \mathrm{Range}\left(\mathbf{G}\left(\mathbf{\Gamma} \langle \widetilde{\mathbf{G}}_S \rangle_k^\dagger\right)\right)$$

and Gaussian embeddings are rotationally invariant, $\widetilde{\mathbf{G}}\langle\widetilde{\mathbf{G}}_S\rangle_k^\dagger\langle\widetilde{\mathbf{G}}_S\rangle_k\widetilde{\mathbf{G}}^\dagger$ shares the same distribution as $(\mathbf{G}\mathbf{\Omega})(\mathbf{G}\mathbf{\Omega})^\dagger$ for a $r \times k$ Gaussian embedding $\mathbf{\Omega}$ with $[\mathbf{\Omega}]_{i,j} \sim \mathcal{N}(0, 1/k)$ *i.i.d.*. Then, we observe that $\|(\mathbf{I}_N - \widetilde{\mathbf{G}}\langle\widetilde{\mathbf{G}}_S\rangle_k^\dagger\langle\widetilde{\mathbf{G}}_S\rangle_k\widetilde{\mathbf{G}}^\dagger)\mathbf{G}\|_F^2$ is the rank-$k$ randomized range-finder error of $\mathbf{G}$, which can be controlled according to Lemma C.8: with probability at least $1 - \exp(-\Omega(\bar{r}))$,

$$\mathrm{tr}\left(\mathbf{\Sigma}^\phi \mathbf{P}_{\mathcal{S}}^\perp\right) \lesssim \frac{1}{n} \left\|\widetilde{\mathbf{\Sigma}}^\phi\langle\widetilde{\mathbf{\Sigma}}_S^\phi\rangle_k^\dagger\right\|_2 \|\mathbf{G} - \langle\mathbf{G}\rangle_{\bar{r}}\|_F^2 = \frac{N}{n} \left\|\widetilde{\mathbf{\Sigma}}^\phi\langle\widetilde{\mathbf{\Sigma}}_S^\phi\rangle_k^\dagger\right\|_2 \mathrm{tr}\left(\mathbf{\Sigma}^\phi - \langle\mathbf{\Sigma}^\phi\rangle_{\bar{r}}\right).$$

By the definition of $\bar{r}$ in Assumption 2.3, $\mathrm{tr}\left(\mathbf{\Sigma}^\phi - \langle\mathbf{\Sigma}^\phi\rangle_{\bar{r}}\right) \leq \mathrm{tr}\left(\mathbf{\Sigma}^\phi\right)/N$ and thus,

$$\mathrm{tr}\left(\mathbf{\Sigma}^\phi \mathbf{P}_{\mathcal{S}}^\perp\right) \lesssim \frac{1}{n} \left\|\widetilde{\mathbf{\Sigma}}^\phi\langle\widetilde{\mathbf{\Sigma}}_S^\phi\rangle_k^\dagger\right\|_2 \mathrm{tr}\left(\mathbf{\Sigma}^\phi\right).$$

$\square$

**Lemma C.8** (Randomized range-finder error (simplifying [32, Theorem 10.7])). *Let $\mathbf{\Omega} \in \mathbb{R}^{r\times k}$ be a Gaussian embedding with $[\mathbf{\Omega}]_{i,j} \sim \mathcal{N}(0, 1/k)$ i.i.d.. For any $\mathbf{G} \in \mathbb{R}^{N\times r}$ and $\bar{r} \in \mathbb{N}$ such that $1.1\bar{r} \leq k \ll \min\{N, r\}$, with probability at least $1 - \exp(-\Omega(\bar{r}))$,*

$$\left\|\left(\mathbf{I}_N - (\mathbf{G}\mathbf{\Omega})(\mathbf{G}\mathbf{\Omega})^\dagger\right)\mathbf{G}\right\|_F \lesssim \|\mathbf{G} - \langle\mathbf{G}\rangle_{\bar{r}}\|_F.$$

# D   Experiment Details for Section 4.1

## D.1   Implementation Details

**Synthetic data generation.**   We consider a set of $N = 2000$ samples with high-dimensional pre-trained representations $\phi(\mathbf{X}) \in \mathbb{R}^{N\times r}$ where $r = 2400$, modeled by a Gaussian mixture model (GMM) consisting of $\bar{r} = 8$ well-separated clusters, each with random sizes and variances. Specifically, we generate the GMM dataset as follows:

- Randomly partition the $N$ samples into $\bar{r} = 8$ clusters with sizes $\{N_j \mid j \in [\bar{r}]\}$.

- For each $j \in [\bar{r}]$, generate the cluster mean $\boldsymbol{\mu}_j \in \mathbb{R}^r$ with $\boldsymbol{\mu}_j = (Z_j\bar{r}) \cdot \mathbf{e}_j$ where $Z_j \sim \mathrm{Unif}([\bar{r}])$ and variance $\sigma_j = Z_j' \cdot \sigma_{\max}$ where $Z_j' \sim \mathrm{Unif}([0, 1])$ and $\sigma_{\max} = 0.04$.

- Generate representations $\{\phi(\mathbf{x}_i) \sim \mathcal{N}(\boldsymbol{\mu}_j, \sigma_j^2\mathbf{I}_r) \mid i \in [N_j]\}$ *i.i.d.* for each cluster $j \in [\bar{r}]$.

- Draw a latent label generator $\boldsymbol{\theta}_g \sim \mathcal{N}(\mathbf{0}_r, \mathbf{I}_r)$. For each cluster $j \in [\bar{r}]$, assign the same label $y_i = \boldsymbol{\mu}_j^\top \boldsymbol{\theta}_g$ for all samples $i \in [N_j]$ within the cluster.

**Ridge regression.**   We solve the ridge regression problem over the selected coreset $\mathcal{D}_S$ of $n$ samples and tune the regularization hyperparameter $\alpha$ via grid search over 100 linearly spaced values in $[10^{-2}, 10^2]$ with 2-fold cross-validation.

## D.2   Baselines

We compare SkMM to the following unsupervised data selection methods for regression:

(a) **Uniform sampling** (Uniform) selects $n$ samples uniformly at random from the full dataset $\mathcal{D}$.

(b) **Herding** [51, 52] (Herding) selects data greedily to minimize the distance between the centers of the coreset $\mathcal{D}_S$ and the original dataset $\mathcal{D}$. Notice that although herding aims to reduce the "bias" of the coreset center, it fails to control our notion of bias in the low-rank approximation sense. Given the construction of the GMM dataset, herding has more emphasis on variance reduction, as illustrated in Figure 2.

(c) **K-center greedy** [53] (K-center) provides a greedy heuristic for the minimax facility location problem that aims to minimize the maximum distance between any non-coreset sample and the nearest coreset sample.

(d) **Adaptive sampling** [44, 56] (Adaptive) iteratively samples data based on their squared norms and adaptively updates the distribution by eliminating the spanning subspace of the selected samples from the dataset. It is proved in the recent work [44] that adaptive sampling achieves nearly optimal sample complexity for low-rank approximations, matching that of volume sampling [39, 41] (with the best know theoretical guarantee) up to a logarithmic factor.

In practice, adaptive sampling generally achieves comparable accuracy to volume sampling for low-rank approximations, with considerably better efficiency [44, 45]. Due to the prohibitive cost of volume sampling in high dimensions, we choose adaptive sampling in the comparison.

(e) **Truncated** [18, 72] and **ridge leverage score sampling** [19, 73, 74] (`T/R-leverage`) are the extensions of classical leverage score sampling [54] to high dimensions. In particular, leverage score sampling is originally designed for low-dimensional linear regression, while degenerating to uniform sampling in high dimensions. Consider the high-dimensional representations $\phi(\mathbf{X}) \in \mathbb{R}^{N \times r}$ ($r > N$) in our setting, for each $i \in [N]$,

- leverage score: $l_i := \phi(\mathbf{x}_i)^\top (\phi(\mathbf{X})^\top \phi(\mathbf{X}))^\dagger \phi(\mathbf{x}_i)$,

- truncated leverage score: $l_i^{(m)} := \phi(\mathbf{x}_i)^\top (\langle\phi(\mathbf{X})\rangle_m^\top \langle\phi(\mathbf{X})\rangle_m)^\dagger \phi(\mathbf{x}_i)$ for a given truncation rank $m$, and

- ridge leverage score: $l_i^{(\rho)} := \phi(\mathbf{x}_i)^\top (\phi(\mathbf{X})^\top \phi(\mathbf{X}) + \rho\mathbf{I}_r)^\dagger \phi(\mathbf{x}_i)$ for a given regularization parameter $\rho > 0$. Larger $\rho$ brings ridge leverage score sampling closer to uniform sampling.

Therefore, both truncated and ridge leverage score sampling balance the variance-bias tradeoff by adjusting the truncation rank $m$ and regularization parameter $\rho$, respectively.

**Baseline details.** For `Herding` and `K-center`, we adopt the DeepCore implementation [1]. Notice that `Herding` is a deterministic algorithm. For `Adaptive`, we use the implementation from [45]. For `T-leverage`, we use a rank-$m$ truncated SVD to compute the leverage scores, with $m = 4\bar{r} = 32$ as in SkMM (*i.e.*, providing both methods approximately the same amount of information and compute). For `R-leverage`, we choose $\rho = 10^3$.

## E   Additional Experiments and Details for Section 4.3

Table 4: Accuracy and F1 score (%) of FT over (the last two layers of) ResNet18 on StanfordCars

| | $n$ | 2000 | 2500 | 3000 | 3500 | 4000 |
|---|---|---|---|---|---|---|
| Uniform Sampling | Acc | $29.19 \pm 0.37$ | $32.83 \pm 0.19$ | $35.69 \pm 0.35$ | $38.31 \pm 0.16$ | $40.35 \pm 0.26$ |
| | F1 | $26.14 \pm 0.39$ | $29.91 \pm 0.16$ | $32.80 \pm 0.37$ | $35.38 \pm 0.19$ | $37.51 \pm 0.23$ |
| Herding [51] | Acc | $29.19 \pm 0.21$ | $32.42 \pm 0.16$ | $35.83 \pm 0.24$ | $38.30 \pm 0.19$ | $40.51 \pm 0.19$ |
| | F1 | $25.90 \pm 0.24$ | $29.48 \pm 0.23$ | $32.89 \pm 0.27$ | $35.50 \pm 0.22$ | $37.56 \pm 0.21$ |
| Contextual Diversity [78] | Acc | $28.50 \pm 0.34$ | $32.66 \pm 0.27$ | $35.67 \pm 0.32$ | $38.31 \pm 0.15$ | $40.53 \pm 0.18$ |
| | F1 | $25.65 \pm 0.40$ | $29.79 \pm 0.29$ | $32.86 \pm 0.31$ | $35.55 \pm 0.14$ | $37.81 \pm 0.23$ |
| Glister [79] | Acc | $29.16 \pm 0.26$ | $32.91 \pm 0.19$ | $36.03 \pm 0.20$ | $38.16 \pm 0.12$ | $40.47 \pm 0.16$ |
| | F1 | $26.33 \pm 0.19$ | $30.05 \pm 0.28$ | $\mathbf{33.26 \pm 0.18}$ | $35.41 \pm 0.14$ | $37.63 \pm 0.17$ |
| GraNd [66] | Acc | $28.59 \pm 0.17$ | $32.67 \pm 0.20$ | $35.83 \pm 0.16$ | $38.58 \pm 0.15$ | $40.70 \pm 0.11$ |
| | F1 | $25.66 \pm 0.15$ | $29.70 \pm 0.22$ | $32.76 \pm 0.16$ | $35.72 \pm 0.15$ | $37.83 \pm 0.11$ |
| Forgetting [80] | Acc | $28.61 \pm 0.31$ | $32.48 \pm 0.28$ | $35.18 \pm 0.24$ | $37.78 \pm 0.22$ | $40.24 \pm 0.13$ |
| | F1 | $25.64 \pm 0.25$ | $29.58 \pm 0.30$ | $32.38 \pm 0.20$ | $35.16 \pm 0.18$ | $37.41 \pm 0.14$ |
| DeepFool [81] | Acc | $24.97 \pm 0.20$ | $29.02 \pm 0.17$ | $32.60 \pm 0.18$ | $35.59 \pm 0.24$ | $38.20 \pm 0.22$ |
| | F1 | $22.11 \pm 0.11$ | $26.08 \pm 0.29$ | $29.83 \pm 0.27$ | $32.92 \pm 0.33$ | $35.47 \pm 0.22$ |
| Entropy [82] | Acc | $28.87 \pm 0.13$ | $32.84 \pm 0.20$ | $35.64 \pm 0.20$ | $37.96 \pm 0.11$ | $40.29 \pm 0.27$ |
| | F1 | $25.95 \pm 0.17$ | $30.03 \pm 0.17$ | $32.85 \pm 0.23$ | $35.19 \pm 0.12$ | $37.33 \pm 0.34$ |
| Margin [82] | Acc | $29.18 \pm 0.12$ | $32.73 \pm 0.15$ | $35.67 \pm 0.30$ | $38.27 \pm 0.20$ | $40.58 \pm 0.06$ |
| | F1 | $26.15 \pm 0.12$ | $29.66 \pm 0.05$ | $32.86 \pm 0.30$ | $35.61 \pm 0.17$ | $37.77 \pm 0.07$ |
| Least Confidence [82] | Acc | $29.05 \pm 0.07$ | $32.88 \pm 0.13$ | $35.66 \pm 0.18$ | $38.25 \pm 0.20$ | $39.91 \pm 0.09$ |
| | F1 | $26.18 \pm 0.04$ | $30.03 \pm 0.14$ | $32.79 \pm 0.15$ | $35.42 \pm 0.16$ | $37.14 \pm 0.12$ |
| SkMM-FT | Acc | $\mathbf{29.44 \pm 0.09}$ | $\mathbf{33.48 \pm 0.04}$ | $\mathbf{36.11 \pm 0.12}$ | $\mathbf{39.18 \pm 0.03}$ | $\mathbf{41.77 \pm 0.07}$ |
| | F1 | $\mathbf{26.71 \pm 0.10}$ | $\mathbf{30.75 \pm 0.05}$ | $33.24 \pm 0.05$ | $\mathbf{36.38 \pm 0.05}$ | $\mathbf{39.07 \pm 0.10}$ |

**Implementation details.** For CLIP standard transform, we transform the image size to 224, with normalization mean (0.48145466, 0.4578275, 0.40821073) and std (0.26862954, 0.26130258, 0.27577711).

**Parameter Count.** We show the parameter sizes for the two-layer fine-tuning experiments in **??**. The representation dimension $d$ is 512 for ResNet18, the number of classes $K$ is 10 for CIFAR-10 and 196 for Stanford Cars. The last layer parameter size is 5130 for CIFAR-10 and 100548 for Stanford Cars. The second but last layer parameter size is 2364426 for CIFAR-10. When we do

Table 5: Classification accuracy (%) of LP over CLIP on CIFAR-10.

| $n$ | 1000 | 2000 | 3000 | 4000 |
|---|---|---|---|---|
| Uniform Sampling | $91.68 \pm 0.45$ | $92.22 \pm 0.25$ | $92.60 \pm 0.14$ | $92.79 \pm 0.12$ |
| Herding [51] | $91.68 \pm 0.27$ | $92.20 \pm 0.17$ | $92.72 \pm 0.51$ | $93.00 \pm 0.45$ |
| Contextual Diversity [78] | $91.98 \pm 0.11$ | $92.53 \pm 0.05$ | $92.81 \pm 0.25$ | $92.99 \pm 0.24$ |
| Glister [79] | $91.09 \pm 0.08$ | $91.60 \pm 0.23$ | $91.83 \pm 0.08$ | $91.87 \pm 0.09$ |
| GraNd [66] | $88.48 \pm 0.90$ | $88.89 \pm 0.53$ | $89.04 \pm 0.24$ | $89.54 \pm 0.46$ |
| Forgetting [80] | $91.51 \pm 0.01$ | $92.15 \pm 0.02$ | $92.61 \pm 0.01$ | $92.81 \pm 0.01$ |
| DeepFool [81] | $91.68 \pm 0.32$ | $91.78 \pm 0.79$ | $91.93 \pm 0.89$ | $92.18 \pm 0.70$ |
| Entropy [82] | $88.66 \pm 1.01$ | $89.96 \pm 0.42$ | $90.27 \pm 1.11$ | $91.06 \pm 0.75$ |
| Margin [82] | $91.22 \pm 0.64$ | $91.60 \pm 0.92$ | $91.94 \pm 0.83$ | $92.09 \pm 0.81$ |
| Least Confidence [82] | $89.38 \pm 1.73$ | $90.49 \pm 1.47$ | $90.83 \pm 1.43$ | $91.26 \pm 1.30$ |
| SkMM-LP | $\mathbf{92.96 \pm 0.07}$ | $\mathbf{93.38 \pm 0.01}$ | $\mathbf{93.67 \pm 0.01}$ | $\mathbf{93.78 \pm 0.04}$ |

Table 6: Accuracy of FT over ResNet18 on CIFAR-10.

| $n$ | 2500 | 3000 | 3500 | 4000 |
|---|---|---|---|---|
| Uniform Sampling | $77.55 \pm 0.16$ | $78.04 \pm 0.18$ | $78.46 \pm 0.09$ | $78.83 \pm 0.15$ |
| Herding [51] | $77.58 \pm 0.17$ | $77.74 \pm 0.19$ | $78.37 \pm 0.14$ | $78.39 \pm 0.11$ |
| Contextual Diversity [78] | $77.24 \pm 0.08$ | $77.65 \pm 0.10$ | $78.17 \pm 0.07$ | $78.22 \pm 0.11$ |
| Glister [79] | $77.46 \pm 0.13$ | $77.95 \pm 0.15$ | $78.19 \pm 0.10$ | $78.54 \pm 0.08$ |
| GraNd [66] | $77.22 \pm 0.10$ | $77.74 \pm 0.07$ | $78.31 \pm 0.11$ | $78.49 \pm 0.10$ |
| Forgetting [80] | $77.32 \pm 0.20$ | $77.87 \pm 0.21$ | $78.05 \pm 0.04$ | $78.53 \pm 0.09$ |
| DeepFool [81] | $77.25 \pm 0.09$ | $77.70 \pm 0.21$ | $78.04 \pm 0.12$ | $78.46 \pm 0.13$ |
| Entropy [82] | $77.55 \pm 0.21$ | $77.74 \pm 0.12$ | $78.23 \pm 0.20$ | $78.41 \pm 0.08$ |
| Margin [82] | $77.24 \pm 0.15$ | $77.81 \pm 0.23$ | $78.32 \pm 0.17$ | $78.52 \pm 0.15$ |
| LeastConfidence [82] | $77.46 \pm 0.23$ | $77.93 \pm 0.14$ | $78.21 \pm 0.17$ | $78.60 \pm 0.10$ |
| SkMM-FT | $\mathbf{77.75 \pm 0.08}$ | $\mathbf{78.12 \pm 0.04}$ | $\mathbf{78.66 \pm 0.06}$ | $\mathbf{79.11 \pm 0.02}$ |

the last two layers fine-tuning, the total parameter size is 8398858 for CIFAR-10 and 2459844 for Stanford Cars.

**StanfordCars Baselines.** For DeepCore baselines, there are some methods that require warmup training before the finetuning (e.g. Uncertainty-Entropy), we use one-layer training and two-layer training (freezing other layers) in the warmup training for linear probing selection and two-layer finetuning selection. The warmup training is done with Adam optimizer with learning rate 0.01 for 10 epochs.

**Finetuning Details** We finetune the model for 50 epochs for both linear probing and finetuning using Adam optimizer with a learning rate 0.01.

