# OpenReview forum: "Sketchy Moment Matching: Toward Fast and Provable Data Selection for Finetuning"
_NeurIPS.cc/2024/Conference — NeurIPS 2024 poster_

### Official Review · Reviewer_dwk6 · 2024-07-12

**Soundness:** 3
**Presentation:** 3
**Contribution:** 3
**Rating:** 6
**Confidence:** 3

**Summary:**

This paper addresses the problem of data selection for finetuning large pre-trained models. The key contributions are:

1. A theoretical analysis of data selection for finetuning that reveals a variance-bias tradeoff in high dimensions.
2. A provable result showing that gradient sketching can efficiently find a low-dimensional subspace that preserves fast-rate generalization.
3. A practical two-stage algorithm called Sketchy Moment Matching (SkMM) that uses gradient sketching to explore the parameter space and moment matching to exploit the low-dimensional structure.
4. Empirical validation on synthetic and real datasets demonstrating the effectiveness of the approach.

**Strengths:**

1. The paper provides a rigorous generalization analysis for data selection in both low and high-dimensional settings. The proofs are detailed and appear sound.
2. The proposed SkMM method is simple to implement and scalable to large models/datasets. Experiments on both synthetic and real data demonstrate the effectiveness of the approach.

**Weaknesses:**

1. Some of the theoretical results rely on assumptions (e.g., low intrinsic dimensionality) that may not always hold in practice. More discussion of the implications when these assumptions are violated would be valuable.
2. The method introduces new hyperparameters (e.g., sketching dimension, moment matching strength) without much guidance on how to set them optimally.

**Questions:**

1. Does the approach extend naturally to other finetuning scenarios beyond linear probing (e.g., adapters, full finetuning)?
2. How does the computational cost of SkMM compare to other data selection methods as the dataset/model size increases?

**Limitations:**

yes

---

> ### Author Rebuttal · Authors · 2024-08-06
>
> We would like to thank the reviewer for their insightful questions and suggestions. We are glad that they found our theory solid and our method effective. On the questions raised in the review:
> 1. __Low intrinsic dimensions of fine-tuning__:
>     Recalling the references [2,72] from the introduction, we highlight that the __low intrinsic dimension of fine-tuning is an extensively observed phenomenon in practice with theoretical rationale__. In particular, [2] demonstrates the surprisingly low intrinsic dimensions of fine-tuning language models in practice; while [72] provides a theoretical justification for the ubiquity of low-rank structures in natural data. One of the main goals of this work is to leverage such low intrinsic dimensions via data selection and enable learning with a sample complexity independent of the high parameter dimension $r$.
>
>     As explained in footnote 7 (page 5), such __a low intrinsic dimension is necessary for sample-efficient learning__. Intuitively, if all the $r$ directions in the high-dimensional parameter space are equally important, when the coreset size $n < r$, the learned parameter $\theta_S$ must fail to capture the orthogonal complement of the space spanned by the coreset and lead to $\mathbb{E}[ER(\theta_S)] \gtrsim r-n$. We will further elaborate on these assumptions in the revision.
>
> 2. __Choices of hyperparameters__:
>     The two hyperparameters in SkMM (Algorithm 3.1) are the sketching dimension $m$ and the constant $c_S$ that controls the moment matching strength.
>     * __Theorem 3.1 provides theoretical guidance for the choice of sketching dimension__: $m = O(\overline{r})$ where $\overline{r}$ is the low intrinsic dimension. In practice, choosing $m$ to be a small constant multiple of $\overline{r}$ (eg, $m = 2\overline{r}$) is generally sufficient. (Notice that such pessimistic constants in theory compared to practice are ubiquitous in sketching, cf. [30,49,81] in submission.)
>     * On the choice of $c_S$, we first __recall from Remark 3.3 that the lower $c_S$ corresponds to the stronger moment matching, resulting in harder optimization of (5), but up on convergence, leading to a better generalization guarantee__. In practice, we start by choosing $c_S$ close to one to ensure the solvability of (5) and then gradually decrease $c_S$ until the optimization of (5) fails or takes too long to converge. Specifically, we fix $c_S = 0.999$ in the synthetic experiments as reported in Sec 4.1 (line 264); while for experiments on real data, we explore $c_S = 0.99, 0.9, 0.8, 0.7, 0.6$ in the respective order for the minimum feasible $c_S$.
>
>     We will include the above discussion on hyperparameter tuning (especially for $c_S$) in the revision.
>
> 3. __Extension beyond linear probing__:
>     Thanks for the constructive question. In the revision, we will extend the empirical evaluation beyond linear probing. In particular, we demonstrate the effectiveness of SkMM in data selection for __finetuning the last two layers__ of an ImageNet-pre-trained ResNet18 on two datasets: CIFAR-10 with 10 well-balanced classes and StanfordCars with 196 imbalanced classes. Please refer to General Response 3 for the results and details of the additional experiments.
>
> 4. __Computational efficiency of SkMM__:
>     SkMM is efficient in both memory and computation while scaling well with the model and data sizes. Please refer to General Response 1 for a detailed discussion on the computational efficiency of SkMM.
>
> We are happy to answer any further questions you may have. Thanks again for the helpful feedback.

---

> > ### Comment · Reviewer_dwk6 · 2024-08-14
> >
> > Thank you for your response on the questions! After reading the other reviews, I would like to keep my current score of 6.

---

### Official Review · Reviewer_gV74 · 2024-07-14

**Soundness:** 2
**Presentation:** 3
**Contribution:** 3
**Rating:** 5
**Confidence:** 3

**Summary:**

The authors study the task of data selection. They extend the classical variance reduction to the high dimensional case and provide a variance-bias tradeoff analysis. Based on the theoretical results, they propose sketchy moment matching, which first utilizes gradient sketchy to form a low-dimensional space and then uses moment matching to reduce the variance.

**Strengths:**

The proposal is a reasonable improvement over the baselines which often only consider bias or variance reduction. The theoretical analysis is also a decent contribution of the paper.

**Weaknesses:**

The experiment focuses on linear probing, which already limits the scope of the evaluation. Furthermore, even under this limited scope, the setting does not seem to be challenging. For the synthetic setup, the sample count is 2000 while the rank is 2500, so it seems not to be a very high-dimension setup (the rank is not so much larger than the sample count). Also, the cluster count seems to be low for both tasks, 8 for synthetic, while the number of class is 10 for Cifar-10.

**Questions:**

For the convenience of the readers, could you list the number of parameter fine-tuned for the Cifar-10 linear-probing task and the number of samples of Cifar-10? (I know this is something that can be looked up online, but these seems to be important numbers to show that the experiment setting is high dimensional)

Also, could you scale up the synthetic experiment, like increasing sample count/rank/number of clusters? Testing on Cifar-100 is also another way to evaluate the performance on a larger number of clusters.

**Limitations:**

Yes, the authors discuss the limitations.

---

> ### Author Rebuttal · Authors · 2024-08-06
>
> We would like to thank the reviewer for their helpful questions and suggestions, and we are glad that they found this work well-presented and theoretically sound. Nevertheless, we believe there have been misconceptions regarding some key notions and the focus of this work. Before diving into the specific questions, we want to emphasize the following two main theoretical/algorithmic contributions of this work:
> * a generalization bound of data selection for high-dimensional fine-tuning with a low intrinsic dimension that unveils the possibility of learning a fine-tuning model with a sample complexity proportional to the low intrinsic dimension (instead of the high parameter dimension), and
> * SkMM, a practical data selection framework inspired by the analysis that finds such coresets efficiently via gradient sketching + moment matching.
>
> Meanwhile, we recall the key notion of "high-dimensional finetuning" as explained in the introduction (footnote 1): the over-parametrized setting where the number of finetuning parameters $r$ is larger than the selected downstream sample size $n$. That is, __“high-dimensional” refers to the relative magnitude $r>n$, instead of the absolute data/model size__.
>
> On the questions raised in the review:
> 1. __Scaling up linear probing on synthetic data does not change the takeaway information__:
>     While we agree with the reviewer that more experiments on larger scales would intuitively be preferred, scaling up the synthetic data experiments is not a reasonable choice because (i) our current synthetic data experiments readily lie in the “high-dimensional”/over-parametrized regime with $n < r$ even though the absolute sample/model sizes are not huge; and (ii) scaling up linear probing experiments on synthetic data generally does not change the takeaway information, as shown below.
>
>     Consider a set of $N=2000$ samples in a higher dimension $r=4000$ generated from a GMM with $\overline{r}=12$ clusters where ridge regression over the full dataset achieves $L(\theta_{[N]})=1.18e-2$. We show the results in the table below, which are consistent with the synthetic data experiments in the submission. The main takeaways remain unchanged: SkMM achieves among the best empirical risks across different coreset sizes $n$, especially for small $n$; while methods that facilitate variance-bias balance (SkMM and T/R-leverage) tend to outperform the rest baselines. (Notice that for $n \ge 120$, data selection can bring even lower empirical risk than learning over the full dataset, which coincides with the recent theory and observation in [41] in submission.)
>
>     | n | 48 | 64 | 80 | 120 | 400 | 800 | 1600 |
>     |---|---|---|---|---|---|---|---|
>     | Herding | 4.04e+03 | 4.04e+03 | 4.04e+03 | 3.87e+03 | 3.74e+03 | 3.67e+03 | 1.76e+03
>     | Uniform | (1.24 $\pm$ 1.19)e2 | (0.81 $\pm$ 1.18)e2 | (0.76 $\pm$ 1.11)e2 | (0.74 $\pm$ 1.11)e2 | (9.92 $\pm$ 0.18)e-3 | (9.56 $\pm$ 0.03)e-3 | (1.64 $\pm$ 0.39)e-2 |
>     | K-center | (6.64 $\pm$ 0.22)e-1 | (4.72 $\pm$ 2.79)e-1 | (3.98 $\pm$ 0.81)e-2 | (1.71 $\pm$ 2.62)e-1 | (3.12 $\pm$ 0.67)e-2 | (1.29 $\pm$ 2.29)e-1 | (2.04 $\pm$ 0.62)e-1 |
>     | Adaptive | (2.19 $\pm$ 1.27)e-2 | (2.78 $\pm$ 3.34)e-2 | (2.51 $\pm$ 2.02)e-2 | (1.88 $\pm$ 1.63)e-2 | (2.39 $\pm$ 4.25)e-2 | (9.52 $\pm$ 0.05)e-3 | (1.71 $\pm$ 2.72)e-1 |
>     | T-leverage | (1.06 $\pm$ 8.15)e2 | (7.21 $\pm$ 8.25)e1 | (5.44 $\pm$ 7.70)e2 | (5.44 $\pm$ 7.70)e2 | __(8.27 $\pm$ 1.62)e-3__ | __(8.66 $\pm$ 1.79)e-3__ | (1.09 $\pm$ 0.37)e-2 |
>     | R-leverage | (3.52 $\pm$ 7.63)e1 | (1.92 $\pm$ 1.77)e-2 | (1.19 $\pm$ 2.18)e-2 | __(1.03 $\pm$ 0.05)e-2__ | (9.48 $\pm$ 0.50)e-3 | (9.54 $\pm$ 0.02)e-5 | (1.17 $\pm$ 0.47)e-2 |
>     | SkMM | __(2.13 $\pm$ 1.28)e-2__ | __(1.65 $\pm$ 1.30)e-2__ | __(1.11 $\pm$ 0.06)e-2__ | __(1.03 $\pm$ 0.03)e-2__ | (9.34 $\pm$ 0.13)e-3 | (9.41 $\pm$ 0.11)e-3 | __(1.08 $\pm$ 0.42)e-2__ |
>
>     To further investigate the effect of problem size on the data selection performance, instead of scaling the synthetic data experiments, we will provide more extensive empirical evaluations of our method on real data, as elaborated below and in General Response 3.
>
> 2. __More comprehensive experiments on real vision tasks__:
>    Thanks to the constructive suggestion, in the revision, we will extend the empirical evaluation to
>    * real vision tasks with a larger number of classes (ie, StanfordCars with 16,185 images in 196 imbalanced classes of cars, in addition to CIFAR-10 with 60,000 images in 10 balanced classes) and
>    * fine-tuning settings beyond linear probing with parameters of much higher dimension (ie, finetuning the last two layers of an ImageNet-pre-trained ResNet18 with $r=2,364,426$ parameters, versus linear probing on CLIP with $r=5,130$ parameters, vide Table 5 in the attached PDF for detailed configurations). We will make sure to include these experimental details in the revision.
>
>    Please refer to General Response 3 for the results and more details of the additional experiments.
>
> We are happy to answer any further questions you may have. If our responses above help address your concerns, we would truly appreciate a re-evaluation accordingly.

---

> > ### Author Response · Authors · 2024-08-11
> >
> > As the deadline for the discussion phase gets close, we would greatly appreciate it if the reviewer could provide some feedback on our responses, which we believe have addressed all the concerns raised in the review regarding the limited scope of the experiments.
> >
> > In particular, we kindly re-highlight the additional experiments in General Response 3 and the attached PDF where we provided stronger empirical evidence on (i) real data with more classes and class imbalance, as well as (ii) fine-tuning settings (beyond linear probing) with much higher dimensions (i.e., the highly over-parametrized settings). Meanwhile, we improved the comprehensiveness of our experimental details thanks to the suggestions from the reviewers (vide General Response 3).
> >
> > We value the opportunity to improve our work based on these constructive suggestions and are always happy to provide further clarifications if needed. Thanks again for your time.

---

> > > ### Comment · Reviewer_gV74 · 2024-08-11
> > >
> > > Thanks for the additional experiments. I am raising my score to 5.

---

> > > > ### Author Response · Authors · 2024-08-13
> > > >
> > > > Thanks a lot for your response, as well as for all the helpful feedback.

---

### Official Review · Reviewer_iMuZ · 2024-07-15

**Soundness:** 2
**Presentation:** 3
**Contribution:** 2
**Rating:** 5
**Confidence:** 3

**Summary:**

This paper concerns the data selection problem: given a collection of $N$ embeddings of dimension $r$ for $r\gg N$, the goal is to pick a subset $S$ of points of size $n$ so that one could run any downstream algorithm on $S$ with a regularization term, so that the empirical risk is small even on the entire finetuning set. Assuming the model is $y=\phi(X) \theta_*+z$ where $\phi: \mathbb{R}^d\rightarrow \mathbb{R}^r$ and $z$ is an i.i.d. noise vector with zero mean and bounded variance, then there exists a subspace that one could project onto and decompose the empirical risk as a bias and a variance term. Further, under the assumption that the second moment matrix has low intrinsic dimension, then one could find a good subspace via gradient sketching: draw a JL matrix $\Gamma\in \mathbb{R}^{r\times m}$ for $m\ll r$, then as long as one has $\Gamma^\top \Sigma^{\phi} \Gamma \preceq c_S \cdot \Gamma^\top \Sigma^{\phi}_S \Gamma$, then the error could be decomposed into a bias, variance and a sketching error term. A sketching gradient, moment-matching algorithm is proposed, involves applying sketching to the gradient, form the Jacobian and solve a quadratic relaxation. Experiments are performed on both synthetic datasets and CIFAR10.

**Strengths:**

The main theoretical contribution is that for over-parametrized setting where $r\gg n$, one could provably show the existence of a subspace that one could project onto and perform data selection on that subspace. Moreover, if the second moment in addition has low intrinsic dimension, then one could use standard dimensionality reduction techniques (in $\ell_2$ norm) to sketch the high-dimensional gradient. In the sketchy moment-matching algorithm proposed in the paper, the authors first sketch the gradient then use uniform sampling  to construct $S$.

**Weaknesses:**

The core results of this paper are not technically very novel and surprising, the algorithm could be interpreted as a generalization of the leverage score sampling via JL trick due to Spielman and Srivastava, STOC'08. The analysis largely draws inspirations from the over-parametrization literature, which makes sense as finetuning is essentially training in an over-parametrized setting. Another point that is a bit unsatisfactory is the sketchy moment-matching algorithm utilizes quadratic relaxation to solve the program efficiently with projected gradient descent, but all analysis is based upon *not solving the quadratic programs*. The authors should try to provide some theoretical justifications of sketchy moment-matching, as that's one of the key contributions of this paper.

**Questions:**

What is the runtime efficiency of your proposed method? It seems the performance is slightly better than ridge leverage score sampling, but ridge leverage score sampling could be implemented in input sparsity time, see the algorithm due to Cohen, Musco and Musco, SODA'17. Their algorithm is based on recursive uniform sampling, so could be implemented efficiently in practice.

**Limitations:**

Yes.

---

> ### Author Rebuttal · Authors · 2024-08-06
>
> We appreciate the insightful questions and suggestions from the reviewer. However, we believe there have been misunderstandings regarding the focus and contribution of this work. We hope that the following responses will help clarify these confusions.
> 1. __Our theoretical contributions are explanatory instead of instrumental__:
>     As discussed in the related works, prior theoretical studies on data selection are conducted either in the low-dimensional linear regression setting or in the asymptotic regime, neither of which aligns with the fine-tuning in practice that manages to learn high-dimensional parameters with much fewer samples. (Although we analyze fine-tuning in the kernel regime following [83], the over-parametrization literature is generally out of the data selection setting.) To fill the gap, we provided
>     * __a generalization bound of data selection for high-dimensional fine-tuning with low intrinsic dimension__ in the non-asymptotic regime that unveils the possibility of learning a fine-tuning model with a sample complexity proportional to the low intrinsic dimension (instead of the high parameter dimension), and
>     * __a practical data selection framework inspired by the analysis__, SkMM, that finds such coresets efficiently via gradient sketching + moment matching.
>
>     Instead of proposing an instrumental theory, we use the existing theoretical tools from sketching and statistical learning to provide __an explanatory theory for the sample efficiency of data selection for fine-tuning that generally admits low intrinsic dimensions in practice__ and __an efficient method for selecting such data__.
>
>     While we agree that gradient sketching shares a similar high-level idea as the JL trick in Spielman and Srivastava, STOC'08 (which is ubiquitous in the sketching literature, cf. [30,81] in submission), leveraging such a fundamental theoretical framework should not compromise the novelty of an explanatory theory.
>
> 2. __The quadratic relaxation is a practical realization of SkMM instead of the only solution__:
>     As elaborated in General Response 2, after gradient sketching, moment matching in the resulting low-dimensional subspace can be realized via various methods, including leverage score sampling and existing polynomial-time heuristics of discrete optimization for experimental design:
>     * Our main motivation for exploring beyond leverage score sampling is the empirical observation that (truncated/sketched) leverage score sampling can perform poorly, especially in the low data regime (cf. Table 1). This can be explained by the dependence of the moment matching guarantee of leverage score sampling on the matrix coherence (vide General Response 2).
>     * Discrete optimization for the V-optimality is an alternative to leverage score sampling. However, such optimization is known to be hard: the best available polynomial-time heuristic with a theoretical guarantee ([4] in submission) involves matrix inversion in each optimization step, bringing instability issues and compromising the practical application.
>
>     Our proposal of SkMM can be interpreted as a quadratic relaxation of moment matching $\widetilde{\Sigma} \preccurlyeq c_S \widetilde{\Sigma}_S$ by assuming that $\widetilde{\Sigma}, \widetilde{\Sigma}_S$ commute (i.e., are simultaneously diagonalizable). This relaxation improves computational efficiency and avoids numerical instability by eliminating the need for matrix inversion. Although the assumption that the two matrices commute does not hold in general, it is a valuable heuristic for designing efficient algorithms, as demonstrated empirically across various domains. For example, in optimization, this approach is used to simplify certain SDP relaxations and approximate solutions effectively (Lu, Monteiro, 2005). In handling distributional shifts, it is used to design optimization formulations to obtain minimax risks efficiently (Blaker, 2000; Lei et al., 2021). We will further clarify this in the revision.
>
> 4. __Efficiency of SkMM__:
>     As ridge leverage score approximation via recursive sampling, gradient sketching in SkMM can be computed efficiently in input sparsity time, and the following moment matching stage takes place in the low dimension $m$ (vide General Response 1).
>     As discussed in General Response 2, while (ridge) leverage score sampling also facilitates moment matching as SkMM and enjoys a slightly better runtime in the moment matching stage (in a lower-order term), with experiments (cf. Table 1), we show that SkMM tends to provide better performance as it is tailored for optimizing moment matching.
>
> We are happy to answer any further questions you may have. If our responses above help address your concerns, we would truly appreciate a re-evaluation accordingly.
>
> References:
> * Lu, Z., Monteiro, R. D. C. (2005). "Primal-dual interior-point methods for semidefinite programming using inexact step directions." Mathematical Programming, 103(2), 453-485.
> * Blaker, Minimax estimation in linear regression under restrictions, annals of statistics 2000
> * Lei et al, Near-optimal linear regression under distribution shift. ICML 2021

---

> > ### Comment · Reviewer_iMuZ · 2024-08-08
> >
> > I thank authors for the comments. I'll keep my score as I believe the paper will benefit from a significant amount of revisions by incorporating all reviewers' comments.

---

> > > ### Author Response · Authors · 2024-08-08
> > >
> > > Many thanks for the timely response. We kindly refer the reviewer to the general responses for the list of important revisions we made, which we believe have accommodated all the suggestions and questions from the reviewers. If the reviewer found any unaddressed concerns, we would greatly appreciate it if they could specify them during the discussion phase. We are always happy to provide further clarifications and improve our work based on the constructive feedback from the reviewers.

---

> > > > ### Comment · Reviewer_iMuZ · 2024-08-13
> > > >
> > > > I've raised my score to 5 in light of authors' responses. However, I do believe the paper would benefit a lot by one more careful iteration based on the revisions proposed by authors.

---

> > > > > ### Author Response · Authors · 2024-08-13
> > > > >
> > > > > Many thanks for the reply and the valuable suggestions. We will make sure to incorporate all the modifications described in the responses carefully in the revision.

---

### Official Review · Reviewer_NGFs · 2024-07-15

**Soundness:** 2
**Presentation:** 3
**Contribution:** 3
**Rating:** 6
**Confidence:** 3

**Summary:**

This paper studies the problem of data selection in the over-parametrized fine-tuning regime, i.e. when the number of fine-tuning parameters $r$ is larger than the amount $N$ of available examples. We want to subsample $n\ll N$ examples that form a representative set to train on, and hopefully achieve quality as close as possible to fine-tuning on the whole set.

The idea is to compute the gradients $G\in \mathbb{R}^{N\times r}$ of all examples wrt the fine-tuning params and then select a subsample $S\subseteq [N]$ such that the Gram matrix of the gradients is approximated: $c\cdot \Sigma_S := c \cdot G^\top I_S G \approx G^\top G := \Sigma$. However, this is not possible to achieve since the model is over-parameterized. Fortunately, if the spectral approximation holds on a low-dimensional subspace of the parameter space, this is good enough, so the authors project the gradients on a random low-dimensional space. The proof goes through under the assumption that the singular values of the gradient matrix are well-concentrated on a small enough (<10%) support.

The experimental results include fine-tuning on a synthetic linear task, as well as fine-tuning a vision transformer on CIFAR-10 image classification.

**Strengths:**

- The authors study the data selection for fine-tuning problem from first principles
- The writing is overall good and math looks sound, even though I didn't check details.
- The experimental results look promising since SkMM beats a variety of algorithms including leverage scores.
- The idea of spectral approximation on a subspace of the parameter space is interesting.

**Weaknesses:**

- Important details on the experimental setup are missing or unclear. Specifically, what is the optimization process after the data is subsampled? For the image classification experiments, what is being fine-tuned, is it all the ViT parameters? For how many epochs?
- The algorithm requires computing the gradients of all samples, which can be computationally expensive. Besides, if we are computing all gradients, why can't we just train one epoch on all datapoints? Why is data selection useful in this case?
- The literature review could be expanded, including relevant papers such as BADGE [1], Coreset-based sensitivity sampling [2].
- In the experimental results, the authors should also compare with margin sampling (in addition to entropy sampling), as well as uniform sampling for the image classification task.
- Computing the moment-matching subset in Algorithm 3.1 seems overly complicated, see questions

[1]: Deep Batch Active Learning by Diverse, Uncertain Gradient Lower Bounds

[2]: Data-Efficient Learning via Clustering-Based Sensitivity Sampling: Foundation Models and Beyond

**Questions:**

In Remark 3.2 the authors write that their goal is to achieve the constraint $\tilde{G}^\top \tilde{G} \preceq c_S \cdot \tilde{G}^\top I_S \tilde{G}$ (1). They subsequently relax this problem and solve the resulting constrained convex optimization problem using projected gradient descent. However, it seems to me that (1) might be equivalent to $U^\top I_S U \succeq c_S^{-1} \cdot I$, where $U\Lambda^{1/2} V^\top$ is the SVD of $\tilde{G}$. Here $U\in {N\times \bar{r}}$ is a tall and thin matrix. This is a spectral sparsification task which could be solved using leverage score sampling on the rows of $U$. Furthermore, this is the same as sampling examples proportional to the squared $\ell_2$ norms of the rows of $U$. Maybe, I'm missing something, so please correct me if I'm wrong.

---

> ### Author Rebuttal · Authors · 2024-08-06
>
> We would like to thank the reviewer for their constructive questions and suggestions, and we are glad that they found this work interesting and well-presented. On the questions raised in the review:
> 1. __Cost of gradient computation and SkMM:__
>     First, we kindly emphasize the __ubiquitous role of gradients in most data selection methods__. For linear probing, the gradients (of the finetuning model instead of the loss) are essentially the last-layer features from the pre-trained model, which can be computed without backward propagation. Such information is necessary for all unsupervised data selection methods discussed in "1.1 Related Works" including leverage score sampling and influence function. Beyond linear probing, gradient computation can bring asymptotically similar costs as *one epoch* of training on the full data set. Notice that this is more efficient compared to methods with *several epochs* of warmup training (eg, [1,15,38,52,56,69,83] in submission, including all methods we compared against in the real-data experiments except uniform sampling and herding). Such cost of gradient computation or warmup training is acceptable because the selected data have their own importance (eg, finetuning similar models with limited memory and computation). Moreover, we refer to General Response 1 that SkMM based on gradient sketching is highly efficient in both memory and computation. As mentioned in Algorithm 3.1, the gradients can be computed and compressed (via sketching) in parallel and on the fly without storage, while all the subsequent steps for moment matching are conducted in the lower dimension $m \ll r$. We will make sure to clarify these points further in the revision.
>
>     Second, we highlight that __training for one epoch on the full dataset is generally insufficient for finetuning__. For example, when finetuning CLIP via linear probing on CIFAR-10 (Table 2), training for one full epoch leads to test accuracy of only $91.94 \pm 0.17$ (cf. $92.96 \pm 0.07$ learned with as few as $1000$ selected data via SkMM).
>
> 2. __Sketchy moment matching v.s. sketchy leverage score sampling__:
>     Up to proper scaling $U^\top diag(\frac{N}{n} s) U \succcurlyeq c_S^{-1}$ (where $s \in \{0,1\}^N$, $\|s\|_0 = n$), we agree with the reviewer that leverage score sampling based on the sketched gradients $\widetilde{G}$ can also provide good control over $c_S$ with high probability. However, such control comes with nuance dependence on matrix coherence and can render the upper bound vacuous in the worst case (vide General Response 2).
>
>     This is exactly the major motivation for the comparison with the truncated and ridge leverage score sampling (T/R-leverage) in the synthetic data experiments. In particular, leverage score sampling based on the sketched gradients is effectively equivalent to T-leverage (ie, when the original gradient dimension $r$ is too high for the exact computation of SVD for $G \in \mathbb{R}^{N \times r}$, randomized SVD based on sketching is generally used as an approximation, in which case leverage scores of $\widetilde{G}$ is exactly T-leverage). In experiments, we also confirmed the indistinguishable behaviors between the two, and therefore only reported one of them. As discussed in Sec 4.1, __while SkMM and T/R-leverage both encourage variance-bias balance__ (by controlling $c_S$) and therefore provide among-the-best risks in Table 1, __SkMM shows better performance on average as it is tailored for optimizing__ $c_S$. Nevertheless, we agree that with very limited computation (ie, even optimizing (5) in low dimension $m$ is infeasible), sketchy (ridge) leverage score sampling would be a more affordable alternative to SkMM.
>
>     Motivated by this insightful question, we will extend Sec 3.2 based on the above discussions on the connection between SkMM and leverage score sampling in the revision.
>
> 3. __Improvements on related works, experiment setup, and more comprehensive experiments__:
>     Many thanks for the constructive suggestions on related works and experiments. We will include discussions on the suggested literature in the next version. In addition, we will provide more experiments on real data with more comprehensive baselines (including margin sampling) in the revision. (Please refer to General Response 3 for the results and details of the additional experiments.) We will also provide more detailed experimental setups in the revision as follows.
>     * After data selection, we fine-tune the model in two different settings: (a) linear probing over CLIP-pre-trained ViT (vide Tables 1 and 3 in the attached PDF) and (b) non-linear finetuning of the last two layers in an ImageNet-pre-trained ResNet18 (vide Tables 2 and 4). Please refer to Table 5 in the attached PDF for detailed configurations like parameter dimensions.
>     * The fine-tuning parameters are optimized via Adam with a learning rate $10^{-2}$, for 200 epochs over CIFAR10 (following [29] in submission, Tables 1 and 2) and 50 epochs over StanfordCars (Tables 3 and 4).
>
> We are happy to answer any further questions you may have. If our responses above help address your concerns, we would truly appreciate a re-evaluation accordingly.

---

> > ### Comment · Reviewer_NGFs · 2024-08-12
> >
> > I would like to thank the authors for their efforts to answer all of my questions. I have increased my score.

---

> > > ### Author Response · Authors · 2024-08-13
> > >
> > > We are happy that our responses have addressed your concerns. Many thanks for the reply and all the constructive suggestions.

---

> ### Author Response · Authors · 2024-08-12
>
> As the deadline for the discussion phase approaches, we would greatly appreciate it if the reviewer could provide some feedback on our responses, which we believe have addressed all the questions in the review. As a summary of important updates we made thanks to the valuable suggestions from the reviewer:
> * In Response 1 and General Response 1, we provided further clarification on the computational efficiency of SkMM and comparison with other data selection methods.
> * In Response 2 and General Response 2, we discussed the connection and difference between SkMM and leverage score sampling in terms of moment matching, where such difference comes from, and why we may want an alternative to leverage score sampling.
> * In Response 3, General Response 3, and the attached PDF, we provided stronger empirical evidence on (i) more real vision tasks and (ii) fine-tuning settings (beyond linear probing) with much higher dimensions, with (iii) more comprehensive baselines (including margin sampling) and (iv) more detailed experimental setup.
>
> We value the opportunity to improve our work based on these constructive suggestions and are always happy to provide further clarifications if needed. Thanks again for your time.

---

### Author Rebuttal · Authors · 2024-08-06

First, we would like to thank all the reviewers for their time, efforts, and valuable suggestions. In the general response, we address some common questions raised in the reviews and summarize important revisions we made
1. __Computational efficiency of SkMM__:
    SkMM is efficient in both memory and computation. Consider the two stages in SkMM:
    * The dimensionality reduction via __gradient sketching__ can be computed __in parallel with input-sparsity time__ and __on the fly without storing the (potentially) high-dimensional gradients__. In particular, let $nnz(G) \le Nr$ be the number of nonzero entries in the (high-dimensional) gradients $G \in \mathbb{R}^{N \times r}$, as discussed in Remark C.1, sketching via a (sub)Gaussian embedding $\Gamma \in \mathbb{R}^{r \times m}$ can be computed in $O(nnz(G) m)$ time, while a sparse embedding (Remark C.1 (b)) can further accelerate the sketching process to $O(nnz(G))$ time. Only the sketched gradients $\widetilde{G} = G \Gamma$ needs to be stored, requiring just $O(Nm)$ memory (recall that $m \ll r$ is a small number proportional to the low intrinsic dimension). For linear probing, $G$ is simply the last-layer feature of the pre-trained model. Beyond linear probing, computing the gradients costs no more than one full training epoch, much cheaper than many data selection methods based on a few epochs of warmup training (eg, [1,15,38,52,56,69,83] in submission, including all methods we compared against in the real-data experiments except uniform sampling and herding).
    * After gradient sketching, __variance reduction via moment matching happens in the low dimension__ $m$, with a low memory footprint $O(Nm)$, taking $O(m^3)$ for the spectral decomposition and $O(Nm)$ per iteration for optimizing the moment matching objective (5).

    We will highlight the computational efficiency of SkMM further in the main text of the revision.

2. __Different methods for moment matching__:
    As mentioned in the limitations and future directions, after gradient sketching, __variance reduction in the resulting low-dimensional subspace via moment matching can be realized using various methods__, among which leverage score sampling based on the sketched gradients $\widetilde{G}$ is arguably the most intuitive approach. Alternatively, we proposed an optimization-based method in Algorithm 3.1 that solves a quadratic relaxation (Remark 3.2) as a practical heuristic. In the synthetic data experiments, we show that __SkMM provides a better variance-bias balance and hence better performance than leverage score sampling__ (T-leverage).

   To better clarify the connection with leverage score sampling, in the revision, we will add __a brief review of the theoretical guarantee for leverage score sampling and explain why an alternative could be desired__. In particular, Theorem 17 of [81] (in submission) implies that leverage score sampling on $\widetilde{G}$ provides $c_S \le \frac{(1+\epsilon) m}{\tau_S N}$ with probability at least $1-\delta$ for a coreset size $n = O(m \log(m/\delta) / \epsilon^2)$, where $\tau_S \in [0,1]$ is the minimum leverage score of $\widetilde{G}$ over the coreset $S$ ($\tau_S$ appears because samples in $S$ are equally weighted in the data selection setting). __Such dependence on matrix coherence can render the upper bound vacuous in the worst case__. Nevertheless, leverage score sampling based on the sketched gradients can be computed more efficiently than SkMM in $O(Nm^2)$ time and can provide good control over $c_S$ when $\tau_S$ is reasonably large. Overall, as pointed out in Sec 4.1, both SkMM and leverage score sampling facilitate variance-bias balance in data selection, while __SkMM is tailored for optimizing moment matching, providing better empirical performance at a slightly higher cost in the low intrinsic dimension__ $m$.

3. __Additional experiments__:
    Thanks to the reviewers' suggestions, we will extend the empirical evaluation on real vision tasks in the revision from the following two aspects:
    * We explore two real vision tasks with different scales and structures: (a) CIFAR-10 with 60,000 images in 10 balanced classes (vide Tables 1 and 2 in the attached PDF) and (b) StanfordCars with 16,185 images in 196 imbalanced classes (vide Tables 3 and 4). (Notice that we expanded the original CLIP linear probing experiments on CIFAR-10 with more comprehensive baselines.)
    * We investigate different fine-tuning settings including (a) linear probing over CLIP-pre-trained ViT (vide Tables 1 and 3) and (b) non-linear finetuning of the last two layers in an ImageNet-pre-trained ResNet18 (vide Tables 2 and 4). We summarize the detailed configurations like the parameter counts in Table 5. (Notice that finetuning the last few layers of strong pre-trained models like CLIP can distort the features and hurt the performance significantly, as studied in (Kumar et al., 2022). Therefore, we choose a weaker pre-trained model, ResNet18, for finetuning beyond linear probing.)

    For both fine-tuning settings on the two datasets, SkMM demonstrates appealing generalization across different coreset sizes. In particular, SkMM tends to outperform the other baselines by a larger margin on the imbalanced StanfordCars dataset compared to the well-balanced CIFAR-10, manifesting the effectiveness of SkMM in balancing the variance-bias trade-off in data selection as suggested by the theory and in the synthetic data experiments.


References:
* Kumar, Ananya, et al. "Fine-tuning can distort pretrained features and underperform out-of-distribution." arXiv preprint arXiv:2202.10054 (2022).

---

### Decision · Program_Chairs · 2024-09-25

**Decision:**

Accept (poster)

**Comment:**

This paper studies the problem of selecting a small subset of a given dataset whilst preserving enough information to perform a downstream fine tuning task. The strategy consists in drawing gaussian projections of the gradients of the datapoints w.r.t. the parameters and matching the dataset variance (cf. Eq. (5)). It is shown that if the data has low intrinsic dimension, the algorithm enjoys a favorable excess risk bound. Many experiments are run on both synthetic and real world data which demonstrate the efficiency of the proposed method.

After the author-reviewer discussion, **the reviewers unanimously agree on acceptance**. However, **reviewer iMuZ** did have some **legitimate concerns** regarding the presentation and the perceived similarity of the method with JL trick in classical spectral sparsification literature the similarity of the proof techniques with  the overparametrization literature. I have followed up with the reviewer in further discussion regarding the proofs, and it seems like the concerns are not that bad: the proofs merely use standard techniques but do not borrow large chunks from any single source. Whilst I agree with the reviewer that the method is similar to the JL trick in the sparsification literature, applying it to a new problem is a **valid contribution**. Thus, overall, it seems that the level of novelty is not as high as it looks at first glance, but is still at a reasonable level.
Still, in the camera ready version the authors should better explain the relationship between their method and the *sparsification literature* [1] and better explain the  *quadratic relaxation*, which  used in the experiments.  The authors promised many such changes to reviewer iMuZ and they should be sure to incorporate all of them. I do have reasonable confidence in the authors' sincerity, since the **rebuttal** was very **thorough** and they have added many **new experiments** at the rebuttal stage, further strengthening the contribution.



**Reference**

[1] Daniel A. Spielman, Nikhil Srivastava, "Graph Sparsification by Effective Resistances". STOC 2008.